# PROCEEDINGS A

Evidence synthesis 

civil engineering

COVID-19, Infection control, SARS-CoV-2, Building ventilation

**Author for correspondence:**
Henry C. Burridge
e-mail: h.burridge@imperial.ac.uk

# The ventilation of buildings and other mitigating measures for COVID-19: a focus on wintertime

Henry C. Burridge[1], Rajesh K. Bhagat[2], Marc E. J. Stettler[1], Prashant Kumar[3], Ishanki De Mel[4], Panagiotis Demis[4], Allen Hart[5], Yyanis Johnson-Llambias[5], Marco-Felipe King[6], Oleksiy Klymenko[4], Alison McMillan[7], Piotr Morawiecki[5], Thomas Pennington[5], Michael Short[4], David Sykes[8], Philippe H. Trinh[5], Stephen K. Wilson[9], Clint Wong[10], Hayley Wragg[5], Megan S. Davies Wykes[11], Chris Iddon[12], Andrew W. Woods[13], Nicola Mingotti[13], Neeraja Bhamidipati[13], Huw Woodward[14], Clive Beggs[15], Hywel Davies[12], Shaun Fitzgerald[11], Christopher Pain[16] and P. F. Linden[2]

[1]Department of Civil and Environmental Engineering, Imperial College London, Skempton Building, South Kensington Campus, London SW7 2AZ, UK
[2]Department of Applied Mathematics and Theoretical Physics, Centre for Mathematical Sciences, University of Cambridge, Wilberforce Road, Cambridge CB3 0WA, UK
[3]Global Centre for Clean Air Research (GCARE), Department of Civil and Environmental Engineering, and [4]Department of Chemical and Process Engineering, University of Surrey, Stag Hill, Guildford GU2 7XH, UK
[5]Department of Mathematical Sciences, University of Bath, Claverton Down, Bath BA2 7AY, UK
[6]School of Civil Engineering, University of Leeds, Leeds LS2 9JT, UK

[7]Prifysgol Glyndŵr Wrecsam, Ffordd yr Wyddgrug, Wrecsam LL11 2AW: Wrexham Glyndŵr University, Mold Road, Wrexham LL11 2AW, UK

[8]AEROS Consultancy, 35 Nairn St, Glasgow G3 8SE, UK

[9]Department of Mathematics and Statistics, University of Strathclyde, Livingstone Tower, 26 Richmond Street, Glasgow G1 1XH, UK

[10]Mathematical Institute, University of Oxford, Andrew Wiles Building, Radcliffe Observatory Quarter, Woodstock Road, Oxford OX2 6GG, UK

[11]Department of Engineering, University of Cambridge, Trumpington Street, Cambridge CB2 1PZ, UK

[12]Chartered Institution of Building Services Engineers, 222 Balham High Road, London SW12 9BS, UK

[13]BP Institute for Multiphase Flow, University of Cambridge, Madingley Rd, Cambridge CB3 0EZ, UK

[14]Centre for Environmental Policy, Imperial College London, London SW7 2AZ, UK

[15]Carnegie School of Sport, Headingley Campus, Leeds Beckett University, Leeds LS6 3QT, UK

[16]Department of Earth Science and Engineering, Imperial College London, Royal School of Mines, South Kensington Campus, London SW7 2AZ, UK

HCB, 0000-0002-0719-355X; RKB, 0000-0002-8928-4534; PK, 0000-0002-2462-4411; M-FK, 0000-0001-7010-476X; PHT, 0000-0003-3227-1844; SKW, 0000-0001-7841-9643; HW, 0000-0001-8994-8057

The year 2020 has seen the emergence of a global pandemic as a result of the disease COVID-19. This report reviews knowledge of the transmission of COVID-19 indoors, examines the evidence for mitigating measures, and considers the implications for wintertime with a focus on ventilation.

## 1. Executive summary

Winter 2020–2021 presented significant risks in managing the ventilation of buildings and maintaining a healthy indoor environment. The extent of the COVID-19 pandemic, both in terms of its stage of development and the controlling measures in-place, varies widely across the globe both inter- and intra-country. Within the United Kingdom of Great Britain and Northern Ireland (UK), the disease seems to be at a dangerous juncture with the reproduction number (i.e. the average number of infections arising from a single infectious case) currently above one for COVID-19 (e.g. the R-number is being reported, as of 11 December 2020, as 0.9–1.0, [1]). This is at a time when cooler weather is approaching where people typically spend longer indoors in the company of others and the supply of outdoor ventilation air is reduced. Simultaneously, the UK and other governments have been trying to maintain their economies, which is accompanied by increased social interactions (e.g. schools and universities remained open throughout autumn 2020 across the UK). More latterly, the programme of mass vaccination is ongoing but concerns about more transmissible variants may make efforts to suppress the spread of COVID-19 while enabling increased levels of social freedoms relevant for the foreseeable future. This report is intended to review much of the knowledge surrounding the indoor spread of COVID-19, and present new results which can inform guidance for mitigating the impact of the disease—our focus has been the UK but our findings will be more widely useful.

It is the premise of this report that COVID-19 may be spread via three main routes (droplet, contact and airborne) all of which we assume to be potentially significant. By consideration of the indoor environment and our behaviour within it, we discuss potential mitigating strategies for all three routes. Appropriate social distancing (§2b) and/or the wearing of face coverings appear to be effective measures to help mitigate the risk of transmission via the droplet route. Increased hand hygiene and the cleaning of surfaces, particularly high-touch public surfaces (i.e. those frequently touched by more than one individual), will reduce transmission via the contact

route. Cleaning of surfaces using disinfectants based on alcohols and reduced contact with these surfaces appear to be effective measures to reduce transmission (§2a(ii)). Indoor spaces that bring individuals together over long periods, e.g. open-plan offices, school classrooms and the like, or those that lead to increased respiratory activity, e.g. gyms, choral halls, etc., are expected to make the airborne spread of the virus an important consideration. Hence, our primary focus is the airborne/aerosol route, since mitigating the spread via this route is the most challenging. The evidence suggests that adequate supply of outside (or at least uncontaminated) air is crucial in helping ensure the reproduction number of a particular indoor space is minimized and ideally remains below one.

Assessment of the ventilation provision, or where practical the monitoring of $CO_2$ levels to indicate ventilation provision (§3), can help manage the risk of COVID-19 transmission via the airborne route—these measures are especially pertinent for wintertime. Most documented cases of transmission which are believed to have arisen from the airborne route have been in environments where the outdoor air supply would not have complied with current UK design guidance. It is inferred from this, and the documented modelling, that provision of outdoor air in-line with existing design guidance will help reduce the risk of transmission by the airborne route. The rate of provision of outdoor air can be inferred by monitoring $CO_2$ levels in occupied spaces, maintaining these below about 1000 ppm being indicative of adequate ventilation in many indoor environments, including offices (with design guidance for some indoor spaces permitting 1500 ppm see [2] and §3b for a fuller discussion). However, higher ventilation rates may be needed wherever activity levels increase beyond desk-based work. Risks can also be reduced by reducing occupancy (while ensuring full outdoor air supply rates are maintained), staggering occupancy (via appropriate timetabling) and by the purging of indoor spaces between events (§2d). The present assessment is based on data derived from the SARS-CoV-2 virus strains that were prevalent during 2020. However, as the virus continues to mutate new, more transmissible, variants have become increasingly prevalent. Should these new strains become dominant then current efforts to suppress the spread of COVID-19 may need to be reviewed. Numerous additional engineering strategies are available to help further reduce the risk of transmission and we review these in detail (see §4).

The review of current knowledge has identified the following key research questions.

— What are the SARS-CoV-2 viral load distributions and respiratory droplet size distributions emitted by an individual carrying out activities such as sitting, walking, talking, singing, sneezing, coughing? How might these vary with an individual's size, age, etc.?
— What is the trajectory of droplets and aerosols containing viral particles from exhalation to removal under different ventilation modes, occupancy levels, occupant behaviour/movement, and environmental conditions (e.g. temperature, humidity, etc.)?
— How can risk and severity of infection of an individual be determined from the nature of viral exposure (i.e. how and where droplets or viral particles are deposited in the respiratory system of an individual, their frequency, the peak/cumulative dose, etc.)?
— What processes determine the timescales for purging a space or determining the frequency of cleaning and which environmental conditions affect these?
— What are the quantitative impacts of wearing face coverings?
— How effective are localized outdoor air supplies and/or purification methods, and what factors affect their results?
— How can existing knowledge, and perhaps answers to the above questions, be deployed to better understand and predict the spread of COVID-19, for which high-spreading statistical outlier events (so-called 'superspreaders') appear to be significant?

## 2. The indoor spread of COVID-19 and wintertime

Florence Nightingale is often credited with having promoted the idea that the indoor environment plays a critical role in determining health outcomes. Her pioneering work on hospital ward design

is still highly relevant with the guiding principles of high ceilings, adequate natural lighting, and sufficient ventilation proving sound design for any indoor environment. However, modern architectural approaches, which often rely on mechanical means to condition the environment, may not follow these principles. In the present work, we focus on two indoor spaces, namely, open-plan offices and school classrooms, because these constitute spaces which are recurrently attended by the same group of people, are occupied for significant portions of each weekday, are frequently populated at moderate to high occupancy density and do not often adhere to Nightingale's design principles. In short, these settings contain very common spaces in which a significant proportion of the population may potentially be at moderate to high risk of exposure to the COVID-19 corona virus. However, the findings and guidance reported within this document apply generically to almost all indoor spaces and we urge that the guiding principles be widely applied. We note that the findings are based on data concerning the SARS-CoV-2 variants dominant in 2020, the increasing spread of more transmissible SARS-CoV-2 variants (e.g. the 'UK' or 'Kent' variant B1.1.7 and the 'South Africa' variant 501Y.V2) may warrant additional caution as new data becomes available.

## (a) COVID-19 transmission indoors

The novel coronavirus disease (COVID-19), which causes respiratory and other symptoms, was declared a pandemic by the World Health Organization (WHO) in March 2020. Transmission of such respiratory infections occurs via virions encapsulated in particles of respiratory secretions (in this case the virus SARS-CoV-2) formed in the respiratory tract of an infected person and spread to other humans via three routes: the droplet route, the contact route and the airborne route [3].

The droplet route involves the transfer of respiratory droplets from an infected person to the mucous membranes of a subject, i.e. respiratory droplets land in the mouth, nose or eyes of others. The contact route takes place when respiratory droplets are deposited onto surfaces that are then touched by other people who go on to touch their mouth, nose or eyes before washing or disinfecting their hands. The airborne route (also called aerosol transmission) occurs when exhaled respiratory droplets are small enough to remain suspended in air such that they can be inhaled into the respiratory system of other people.

At the beginning of the COVID-19 pandemic, a lack of direct empirical evidence on airborne transmission of SARS-CoV-2 highly influenced health policy decisions which were intended to control the pandemic and the public response to it. However, an increasing body of evidence (particularly from poorly ventilated indoor environments), a better understanding of the disease progression, and information on the asymptomatic and pre-symptomatic transmission of the virus strongly support the case for airborne transmission of SARS-CoV-2 virus (see [4,5], for discussion and references therein).

In an indoor environment, the ventilation flow modulates the transport and advection of any aerosols (including bio-aerosols), pollutants and $CO_2$ produced by indoor-sources/occupants and further determines their subsequent removal from within the indoor environment. Traditionally, building ventilation has been studied in the context of thermal comfort and in the last few decades energy efficiency. The focus on energy efficiency, often imposed by changes in construction standards, in combination with space restrictions arising from higher population density, have led to more tightly constructed, and less spacious, buildings. These tightly constructed structures can, without careful design, maintenance and operation, result in inadequate ventilation provision. However, there has been a timely shift of focus and, in addition to energy efficiency and thermal comfort, indoor air quality (and implicitly the removal of any indoor airborne pollutants produced by the occupants) has become a core focus [6]. In the present work, we review the current knowledge to offer advice to mitigate COVID-19 spread via the airborne route. However, it is essential not to do so at the expense of considering droplet transmission and contact transmission, and measures to minimize the risk of transmission via these routes must be given as much priority as consideration of the airborne route. As such, within this section we review transmission via the droplet and aerosols route (§2a(i)) and contact routes (§2a(ii)), we then

consider the role that social distancing (§2b), face coverings (§2c), and occupancy behaviour (§2e) can play in affecting the various modes of transmission. In §3, we address the role of ventilation in influencing the airborne transmission route. We go on to discuss the suitability of other measures in mitigating the spread (§4), and we present four appendices within the electronic supplementary material which cover: recommendations for natural ventilation in winter (appendix A), factors affecting and modelling considerations for, surface transmission (appendix B), details of the potential for ultraviolet germicidal irradiation (UVGI) air disinfection to mitigate COVID-19 transmission (appendix C), and aerosols in the context of singing and musical instruments (appendix D).

### (i) Transmission via droplets and aerosols

Respiratory diseases are transmitted by exposure to pathogen-laden droplets produced by expiratory events such as breathing, coughing, sneezing, speaking, singing and laughing [7,8]. The expiratory droplets range between 0.01 and 1000 µm [9], and conventionally they are classified in two categories; droplets smaller than 5–10 µm are referred to as droplet nuclei or aerosols, whereas droplets larger than 5–10 µm in diameter are classified as respiratory droplets [10,11]. This somewhat arbitrary size classification implicitly refers to the transmission modes/mechanisms, namely droplet, and airborne transmission. However, the distinction between droplet transmission and airborne transmission determined by a simple cut-off in droplet size neglects a multitude of physical processes crucial to the droplet evolution within an indoor environment. For example, droplets that are larger than a selected cut-off size at the source may shrink due to evaporation, becoming sufficiently small before they settle, such that they then contribute to airborne transmission.

The distinction between droplet transmission and airborne transmission is explained by the route of infection. Droplet transmission occurs at short-range when a subject is exposed to large pathogen-laden droplets expelled by an infected person that impact upon their mucous membranes. Droplet transmission usually occurs in close proximity (see §2b). While droplets may fall quickly onto a surface close to the source, aerosols are expected to remain airborne for longer periods and can be advected away from the source with ventilation flows leading to what we term 'the airborne transmission route'. Therefore, an important aspect of understanding droplet and airborne transmission is the size distribution of the expiratory droplets containing the virus, which are composed of water, salts and organic material [12]. Droplets and aerosols produced by violent expiratory events such as coughing and sneezing have been investigated and reviewed by several authors, including Yang *et al.* [13], Bourouiba *et al.* [14], Bourouiba [15] and Mittal *et al.* [3]. However, under normal circumstances, especially for asymptomatic or presymptomatic carriers (who generally do not cough), the cumulative amount of expiratory fluid and consequently the droplets and aerosols produced by low-frequency intermittent events such as coughing and sneezing are likely to be less than that of high-frequency events such as breathing and talking. de Oliveira *et al.* [16] modelled the evolution of expiratory spray and aerosol. They showed how the size of exhaled particles changes due to evaporative drying, with particles with original diameter less than 10 µm reaching an equilibrium size that is dependent on the droplet composition in the timescale of approximately 1 s. Based on their estimates, they reported that 30 s of continued speech - compared to a short cough–might release an order of magnitude higher viable viral dose into the surroundings.

The studies conducted on disease progression suggest that infectivity of COVID-19 peaks before the onset of symptoms and, consequently, preventing pre-symptomatic and asymptomatic transmission is key to containing the spread of the disease [17]. At the early stage of SARS-CoV-2 infection, upper respiratory tract symptoms and the presence of high concentrations of SARS-CoV-2 virus in oral fluids are common [18] which support the recent findings identifying speech droplets to be a potential cause of transmission [7,19,20].

Conversational speech produces a wide range of droplet sizes (sub-micrometre up to the order of 100 µm) which are exhaled at speeds of the order of 3.5–4 m s$^{-1}$. The reported size

distributions of speech droplets show a large variation, due to different measurement techniques, different vocalizations, evaporation of droplets prior to measurement, and natural variation among different people [21]. Aerosol measurements capable of measuring particles in the range 0.5–20 µm indicate that speech droplets form across the measurement range, with geometric mean diameter of approximately 1 µm, droplet number concentrations in exhaled breath of the order of 0.1–1 cm$^{-3}$, and exhaled particle emissions rates of the order or 1–10 s$^{-1}$ [22,23]. Asadi et al. [22] showed that speaking louder is correlated with higher particle emissions and found that a small fraction of people are 'super-emitters', who consistently release an order of magnitude more particles than others. Chao et al. [24] measured the droplet size distribution of cough and speech droplets at mouth opening and found the geometric mean diameter of cough droplets was 13.5 µm. By contrast, speech droplets were 16 µm, but had a reported maximum diameter of up to 1000 µm. Xie et al. [21] reported the average speech droplet diameter to be between 50 and 100 µm. Interestingly, this study also showed that both the number and the droplet size increased significantly when the subjects swallowed food dye solution (with or without sugar) before the experiment, indicating that eating may promote the release of higher numbers and larger sizes of expiratory droplets. Although light scattering measures only larger droplets and consequently provides a conservative estimate of total droplet count, Stadnytskyi et al. [20] measured high droplet release rates relative to other studies when using this technique. Both Anfinrud et al. [19] and Stadnytskyi et al. [20] showed that speech droplets of size 10–100 µm can remain suspended for up to 30 s. Therefore, it is imperative to appreciate that speech droplets can potentially transmit respiratory diseases by both the droplet and airborne transmission routes. By contrast, it has consistently been shown that the majority of aerosol particles in exhaled breath are less than 5 µm [25], which diminishes the possibility of droplet transmission when breathing; however, airborne transmission cannot be ruled out.

When droplets are exhaled, they evaporate at a rate that depends on droplet size and composition, and the relative humidity and temperature of the air. Redrow et al. [26] compared the evaporation time and resulting nuclei sizes of model sputum, saline solution and water droplets. They showed that sputum droplets containing protein, lipid, carbohydrate, salt and water leave larger nuclei than salt solution. They calculated the time scales of evaporation of water droplets at room temperature, for relative humidities between 0 and 80%, to be 0.1–1 s for droplets less than 10 µm and 7–40 s for 100 µm droplets. Therefore, it is expected that droplets larger than 100 µm settle on the floor or other nearby surfaces [27], while droplets smaller than about 10 µm tend to form nuclei and are transported as passive scalars, i.e. they were transported by airflows without the dynamics of the airflow being significantly affected [28].

The final size of exhalation droplets depends upon many factors including the initial size, non-volatile content, relative humidity, temperature, ventilation flow and the residence time of the droplet. Marr et al. [29] gave the equilibrium size for 10 µm-sized model respiratory droplets containing 9 mg ml$^{-1}$ NaCl, 3 mg ml$^{-1}$ protein, and 0.5 mg ml$^{-1}$ surfactant to be 2.8 µm and 1.9 µm at relative humidities of 90% and less than 64%, respectively.

A significant uncertainty in our ability to quantify the relative importance of airborne transmission is the viral load associated with different aerosol sizes for different expiratory events, at different stages of infection and the potential for natural variation among people. This information is currently unknown, leading to large uncertainty bounds.

### (ii) Transmission via surface contacts and fomites

Fomites are inanimate objects that have become carriers of virus particles and these have been shown to play a role in the spread of viruses. Little is known of the true risk of becoming infected by SARS-CoV-2 through this pathway as it is difficult to isolate from the droplet and airborne transmission routes, the dose required to become infected has not been determined and the majority of studies that have looked at its survival on surfaces have used far greater viral loads than would be deposited naturally [30]. Knowledge of SARS-CoV-2 survival on surfaces and interactions between the fomite and droplet/airborne routes through deposition and re-suspension of viral particles is nevertheless important in minimizing the risk of infection.

Fomite transmission occurs predominantly from human behaviour through non-infected individuals making contact with or handling infected objects, which may be infected via deposition of large droplets from infected individuals or via direct contact by individuals with viral particles on their hands. The non-infected individual then transfers viral particles from their hands to mucous membranes by touching their eyes, nose or mouth. Advice given by various groups to avoid unnecessary contact between hands and objects in public environments, as well as advice to avoid touching one's face, is sound and should be encouraged. In China, it was found that the majority of surfaces within hospital wards that had infected individuals, were found to have traces of the virus [31], and therefore cleaning of surfaces is an important mitigation strategy to avoid infection.

This section briefly summarizes what is known about SARS-CoV-2 and its survival and transmission via surfaces. Note that a more thorough and detailed literature review is found in the electronic supplementary material, appendix B for this document, which details the sources upon which this advice is based.

*Factors affecting the survival of SARS-CoV-2 on surfaces*. A number of environmental factors affect the survival of SARS-CoV-2 on surfaces.

— **Temperature** effects have been reported to be significant, with higher temperatures decreasing survival times [32]. Comfortable indoor temperatures should be maintained and the use of air conditioning should be minimized wherever practical with the appropriate supply of outdoor air remaining a priority.
— **Humidity** has been shown to also have an effect on the virus, with drier conditions being more suitable for virus survival [33]. While higher humidity is preferable to reduce viral infection, there are numerous health issues related to high humidity and promotion of mould growth. We therefore advise that in cold weather the relative humidity should be maintained at between 40 and 50%, rather than below 30%, which is typical of many indoor environments in winter [32].
— **Light** is also demonstrated as an effective method for SARS-CoV-2 deactivation with 90% of the virus inactivated every 6.8 to 14.3 min depending on the intensity of simulated natural light [34]. UV-C light has been shown to deactivate other strains of coronavirus [35]. While the use of artificial light cleaning technologies is not suggested as a replacement for disinfectant cleaning practices, well-lit rooms, particularly via natural lighting is preferred based on evidence from SARS-CoV-2 and other viruses [32,34].
— It is now well known that SARS-CoV-2 has different survival times on different surfaces, with laboratory inoculations of SARS-CoV-2 survival rates varying from 3 h for paper and tissue to up to 72 h (3 days) on hard, smooth surfaces such as plastic and stainless steel (and also on surgical masks) [36]. Glass and bank notes have survival times in the region of 3 days, with cloth and wood reported at 2 days. More recent results suggest even longer survival times such as at least 28 days at 20°C and 50% relative humidity when dried onto non-porous surfaces at a starting viral load typically excreted by infected patients [37]. While likely viral loads on contaminated objects are highly uncertain, and hence the risk associated with the fomite pathway relative to that for droplet and aerosol transmission cannot currently be quantified, we believe it is important to regularly clean often-handled objects and surfaces in public spaces.

*Cleaning recommendations*. The above leads to the recommendation that it is important to frequently clean high-touch surfaces such as door handles, classroom and meeting room desks, tap handles, swing door handles, ticket machines, pin code keypads, communal office kitchens, etc. Providing point-of-contact public alcohol-based disinfectant, as is now common on university campuses, shops and public transport, is an effective mitigation strategy to ensure that public spaces are less likely to become contaminated. We conjecture that an effective mitigation strategy for certain public spaces that involve fomites, such as public computer laboratories found in universities or libraries, would be to encourage users to clean the workspace (keyboards, mouse,

desk and hands) before and after use. Further research is needed to establish the efficacy of such interventions depending on the frequency of cleaning and in comparison (or in combination) with other mitigation approaches such as handwashing. The wearing of face coverings will also reduce droplet deposition on the workspace and should therefore be encouraged.

Disinfectants based on alcohols (ethanol, 2-propanol, 2-propanol with 1-propanol) as well as other common disinfectants (glutardialdehyde, formaldehyde, and povidone iodine (0.23–7.5%)) are very effective against coronaviruses in general, and come highly recommended [38]. Substances such as 70% ethanol, 70% isopropanol, 0.1% hydrogen peroxide and 0.1% soldium laureth sulfate have been tested specifically on SARS-CoV-2 on various surfaces, and can effectively deactivate the virus within 1 min [39]. UK Government advice to use soap and water to clean surfaces may not be the most effective, since soap and water alone was not shown to deactivate the virus after 5 min [40]. However, if accompanied by scrubbing this may be more effective at its physical removal [38]. Based on the above experimental findings, confirming that sodium laureth sulfate is present as an ingredient in the soaps used for disinfection [39] can be useful, noting that this substance may cause irritation and is not used in products for sensitive skin and other skin conditions. Sodium hypochlorite, which has not been tested specifically on SARS-CoV-2, requires a concentration of at least 0.2% to be effective [38]. In addition, we refer readers to the guidance on the appropriate and safe use of cleaning substances [41,42].

## (b) Social distancing indoors

Social distancing describes the effort to ensure that individuals remain separated by a particular distance and is often recommended primarily to reduce the transmission of disease via the droplet route. The quantification of an appropriate social separation/distance to avoid droplet transmission is often based on research by Wells [43]. His model for disease transmission considered droplets produced by sneezing or coughing to behave ballistically, with no interaction between them; i.e. droplets would fall from the height they were produced to the floor (approx. 2 m vertically), while simultaneously evaporating. In spite of the evaporation process, the so-called 'large' droplets would reach the floor; meanwhile the so-called 'small' droplets would evaporate quickly leaving relatively 'dry' aerosol particles known as droplet nuclei. Wells proposed two mechanisms for infection: droplet transmission due to large droplets and airborne transmission due to small droplets that evaporate sufficiently to become suspended in the air for long times. Wells calculated a cut-off between small and large droplets as 100 $\mu$m (not 5 $\mu$m, as is often cited).

Wells' falling-evaporation curve has been used to propose a social distancing rule by considering how far large droplets travel horizontally as they fall. The total distance travelled by a droplet is determined by its initial horizontal velocity, and also whether it is contained in a coherent flow structure caused, for example, by coughing or sneezing [15]. Small droplets fall more slowly than large droplets, so travel further, but they take less time to evaporate. The size of the largest droplet that totally evaporates before falling 2 m is identified, then the horizontal distance this droplet travels is calculated and used to define a social distancing rule. For example, for droplets expelled at 10 m s$^{-1}$ (typical for coughing) this distance is 2 m, while for droplets expelled at 1 m s$^{-1}$ (breathing) this distance is less than 1 m [28].

In addition to the work of Wells, the experiments by Jennison [44] have also been used as evidence for social distancing of 1–2 m. Jennison used high-speed photography to examine the fall of droplets produced by talking, coughing and sneezing, concluding that the majority of the droplets fell to the ground within 1 m (the field of view for the experiments). However, no details were provided about how this conclusion was reached and it was acknowledged that the experimental method was not sensitive enough to capture all the droplets, tending to select for larger droplets [45].

The Wells model and many of its extensions assume that droplets behave independently of each other, travelling ballistically. However, more recent research has shown that this is often not a good assumption. Experimental studies of coughing and sneezing show that exhalation results

in a puff of warm, humid air that influences the distance travelled by groups of droplets [45]. Experimental images show that the turbulent gas cloud emitted from a human sneeze can travel 8 m, carrying particles along with it [15]. These results suggest that for coughing and sneezing, the exhalation or puff needs to be taken into account in calculating the maximum distance travelled by droplets. Moreover, consideration of these images highlights that violent respiratory events have significant directionality and in contexts where the direction of these events can be inferred then this should influence the layout of desks, etc., within classrooms and offices.

Current social distancing advice aims to reduce droplet transmission, however social distancing can also reduce transmission by small droplets, as aerosols are diluted with distance from the source [46]. The suggested distance of 2 m is based on a model that assumes ballistic droplets, rather than particles that travel within an exhaled puff that dilutes with distance. Further research is needed to identify an appropriate distance at which sufficient dilution has occurred. This distance will be influenced many factors, including the nature of the exhalation (breathing, talking, coughing, etc.), the properties of the air (humidity, temperature), the droplet size distribution, the virus concentration with droplet size, and the size of the infectious dose.

There is increasing evidence that airborne transmission of SARS-CoV-2 is significant (e.g. [5,25]). Outdoors, in well-ventilated indoor spaces, and for short interaction times, social distancing will reduce airborne transmission and will therefore reduce infection risk when combined with other measures, such as face masks and hand washing. However, for longer exposure times and/or in poorly ventilated spaces, social distancing is unlikely to be sufficient as the dilution at room scale will not reduce the aerosol concentration enough to avoid an infectious dose. This emphasizes the importance of additional measures, such as good ventilation and face coverings.

## (c) Guidance on face masks and coverings

There is substantial evidence that face masks and coverings can lessen the spread of COVID-19 by reducing the emissions of virus-laden particles, both droplets and aerosols [47] from the wearer. Moreover, it has also been shown that face masks and coverings may also be protective by reducing the dose of SARS-CoV-2 received by the wearer. Face shields might prevent the escape of droplets from the user, as well as similarly protecting the user and providing some limited protection to the eyes, but they are unlikely to stop the transmission of aerosols at short or long range (the airborne route) unless used in conjunction with a face mask or covering. The use of face masks and coverings is especially important in settings where there is decreased physical distancing such as shops, public transport or in work environments. However, the impact of wearing face masks or coverings on the intended interactions and activities should be carefully considered, with members of disadvantaged or vulnerable groups given due attention.

A systematic review and meta-analysis of distancing, face masks and eye wear (up to May 2020) concluded that 'face mask use could result in a large reduction in risk of infection, with stronger associations with N95 or similar respirators compared with disposable surgical masks or similar' [48]. They also noted that 'transmission of viruses was lower with physical distancing of 1 m or more' and 'eye protection also was associated with less infection'. This review was in both healthcare and non-healthcare settings. Another meta-analysis concluded 'that face masks protect populations from infections and do not pose a significant risk to users' [49].

Masks often refer to surgical or respiratory masks (respirators) that medical staff use, whereas face coverings encompass broader types and materials such as homemade cloth masks, but may include just a simple scarf [50]. Leung *et al.* [47] stated that surgical masks have been demonstrated 'to reduce coronavirus detection and viral copies in large respiratory droplets and in aerosols'. Their results suggest that these masks could prevent transmission of viruses from symptomatic (and for COVID-19 pre-symptomatic/asymptomatic) individuals.

There is evidence that face masks and coverings may be effective at reducing COVID-19 cases across the world. Mitze *et al.* [51] noted infections dropped to near zero after face masks were introduced on 6 April 2020 in Jena, Germany and concludes 'that 20 d after becoming mandatory

face masks have reduced the number of new infections by around 45%. As economic costs are close to zero compared to other public health measures, masks seem to be a cost-effective means to combat COVID-19'. Similarly, 'from epidemiological data, places that have been most effective in reducing the spread of COVID-19 have implemented universal masking, including Taiwan, Japan, Hong Kong, Singapore, and South Korea' [52]. Zhang *et al.* [53] conclude 'that wearing of face masks (or coverings) in public corresponds to the most effective means to prevent inter-human transmission, and this inexpensive practice, in conjunction with extensive testing, quarantine, and contact tracking, poses the most probable fighting opportunity to stop the COVID-19 pandemic, prior to the development of a vaccine'.

'The benefits that face masks could offer as a non-pharmaceutical intervention were investigated using mathematical models and show that face mask use by the public could make a major contribution to reducing the impact of the COVID-19 pandemic' [54]. They demonstrated that mask (and face covering) wearing can reduce transmission, with high rates of adoption likely to reduce the reproduction number to below one.

As of the 5 June 2020 the World Health Organization [55] has recommended the wearing of face masks and coverings for communities and circumstances, where there is risk of transmission and in areas where physical distancing cannot be achieved, for the 'general public should be encouraged to use medical and non-medical masks in areas with known or suspected community transmission' of COVID-19 virus. More recently (in August 2020) the WHO have released a video to advise the wearing of face masks or coverings. The Royal Society Data Evaluation and Learning for Viral Epidemics, or 'DELVE', Initiative [56] reported that 'asymptomatic (including pre-symptomatic) infected individuals are infectious' and 'respiratory droplets from infected individuals are a major mode of transmission'. Reporting that masks reduced droplet dispersal and 'cloth-based face masks reduce emission of particles by variable amounts', similar to the percentage reductions reported (of viral, bacterial and dust particles) by surgical masks. In a more recent study, key findings were that cloth face coverings are effective in protecting the wearer and those around them and that face masks and coverings are part of 'policy packages' that need to be seen together with other measures such as social distancing and hand hygiene [50].

The physics of particle capture by the materials of face masks is complex and it is also recognized that these mechanisms will be at play in face coverings. Beyond those normally associated with filtration, they reduce the forward momentum of the exhaled fluid, effectively limiting the spread of air and aerosol from the wearer. Face coverings and masks keep the fluid close to the body plume lowering the chances of direct exposure of potential contagion-laden droplets, and by preventing the release of the non-gaseous constituents of the expiratory fluids into the environment [57–59].

The study of efficacy of face coverings and masks in regulating emissions during expiratory activities was conducted by Asadi *et al.* [60] they noted 'both surgical masks and unvented KN95 respirators, even without fit-testing, reduce the outward particle emission rates by 90 and 74% on average during speaking and coughing, respectively, compared to wearing no mask, corroborating their effectiveness at reducing outward emission'. Their results imply cloth face covering and other masks would 'reduce emission of virus-laden aerosols and droplets associated with expiratory activities.' Including many of the particles emitted in the aerosol range (less than $5\,\mu m$), and inertial impaction will be more prevalent as particle sizes increased and is likely to lead to fewer droplet size particles being emitted (greater than $5\,\mu m$).

A similar experiment also looked at the efficacy of masks, coverings and face shields and their ability to reduce transmission against simulated cough-generated, small aerosol particles from 0 to $7\,\mu m$. Lindsley *et al.* [61] results show N95 respirators are 99% effective at blocking transmission, surgical masks 59%, 3-ply cotton cloth face masks 51% and a polyester neck gaiter 47% and folded double 60%, whereas a face shield blocked only 2% of the cough aerosol. The masks, coverings and face shield became progressively more efficient at blocking the cough aerosol, as the aerosol particle size fraction increased from less than $0.6\,\mu m$ to $7\,\mu m$. They stated 'These results suggest that cloth face coverings would be effective as source control devices

against the large respiratory aerosols that are thought to play an important role in SARS-CoV-2 transmission'.

Face masks and coverings may also protect the wearer, to different degrees of effectiveness. To quote van der Sande et al. [62] 'Any type of general mask use is likely to decrease viral exposure and infection risk on a population level, in spite of imperfect fit and imperfect adherence, personal respirators providing most protection'. They found that FFP2 (N95) masks protected the wearer significantly better than surgical masks or a simple cloth. Also, similarly outward protection (emissions) was correlated with protection, with FFP2 masks being the best of those tested.

Thus the material and the make-up of the face mask or covering is important for the filtration efficacy. Wilson et al. [63] modelled the risk of transmission based on published data on the effectiveness of various material against a viral challenge; these included FFP2 (N95) respirator material (at 95% efficiency), with surgical mask material slightly less efficient. A vacuum cleaner bag (83%) was found to be the most efficient household material and a scarf (44%) the least efficient. In between these two materials were a tea towel, cotton mix, linen, a pillowcase, silk and 100% cotton T-shirt.

User protection is discussed by Gandhi et al. [64] who hypothesize that 'universal masking reduces the 'inoculum' or dose of the virus for the mask-wearer, leading to more mild and asymptomatic infection manifestations'. They state numerous cases where universal masking led to fewer cases, or more asymptomatic cases as opposed to comparative examples, whether this was in animals, on cruise ships, meat factories or regions of universal mask wearing. 'Countries accustomed to masking since the 2003 SARS-CoV pandemic, including Japan, Hong Kong, Taiwan, Thailand, South Korea, and Singapore, and those who newly embraced masking early on in the COVID-19 pandemic, such as the Czech Republic, have fared well in terms of rates of severe illness and death' [64].

The use of face shields as an alternative to face masks and coverings within the service industry in the UK is popular, it is also of great benefit to those who are hard of hearing (HoH). 'Face shields can substantially reduce the short-term exposure of health care workers to large infectious aerosol particles, but smaller particles can remain airborne longer and flow around the face shield more easily to be inhaled' [65]. Thus their use might prevent the escape of droplets from the user, as well as similarly protecting the user and providing some limited protection to the eyes. They are however, unlikely to stop the transmission of aerosols (the airborne route), without being used in conjunction with a face mask or covering. Verma et al. [66] visualized the flow around a face shield (and respirators with valves) and noted 'that although face shields block the initial forward motion of the jet, the expelled droplets can move around the visor with relative ease and spread out over a large area'. Similar results were found for respirators with exhale valves, with aerosols escaping through the valve.

Any covering of the face negatively impacts the HoH, with close to 11 million people in the UK who are HoH (around one in six people). Where possible, it would be prudent to adapt the guidance on face masks to accommodate them, as the usage of face masks may hinder the ability to listen and lip-read. An effective strategy to accommodate those HoH is to use clear face masks [67], or the use of novel technologies such as captioning apps.

Finally, the application of the precautionary principle suggests that people should be encouraged to wear face masks and coverings on the grounds that, in many circumstances, we have little to lose and potentially something to gain from this measure. Greenhalgh et al. [68] said that 'masks are simple, cheap, and potentially effective … and outside the home in situations where meeting others is likely (for example, shopping, public transport), they could have a substantial impact on transmission with a relatively small impact on social and economic life'.

## (d) Source reduction through timetabling and purging between events

Airborne infection risk is reduced when the ventilation provision of outdoor air is maximized. Operating the existing indoor environment conditioning and controlling equipment in a manner that fixes the outdoor air supply rate to be maximal (with due consideration to the practical limits

for a comfortable indoor environment), the airborne risk can be greatly reduced by lowering occupancy in a given indoor space. For example, should the ventilation plant be kept running at the same level (i.e. unchanged absolute outdoor air supply rate) and the occupancy halved (e.g. through week in—week out working) in an indoor space then the chances that infection occurs within is approximately halved.

Where reductions to the absolute levels of occupancy are impractical then some change of timetabling should be considered. Where possible, attendance should be extended by some occupants arriving and leaving earlier than usual with other occupants arriving and leaving later than usual. Moreover, consideration should be given to purging rooms between meetings, classes and events. This would require the room to be unoccupied between consecutive events during which period all possible efforts are made to increase the outdoor air supply rate (whether by opening windows, doors and ventilation systems). At the end of the purging period it is best if the room is cleaned in the manner detailed in §2a(ii) to minimize the chance of spread to the next occupants. It is highly likely that the greatest rates of decay in the concentration of virus-laden particles will occur at the start of the purging periods. So any purging duration is better than none as long as the increased ventilation flows are given time to establish themselves. That said, the longer the purging duration then the lower will be the resulting concentration levels. To establish how effective these purging strategies may be, a simple 'room-change' time scale can be calculated by dividing the volume of the space by an estimate of the rate of outdoor air supply during purging; note that the room-change time scale, in hours, is simply the inverse of the air changes per hour. Melikov et al. [69] report the intake fraction (the proportion of air exhaled by the infected person that is then inhaled by another occupant) for various purge scenarios. They quantified the reduction in intake fraction for ventilated cases with purging periods (of between 15 and 30 min) which were in the range 0.2–1.8 room-change time scales. They went to consider cases of increased ventilation rates which led to purging times (15 min) being around 5.5 room-change time scales. Their conclusions included the suggestion that periods of constant occupation should be short with appropriately long breaks being recommended; break durations of 10–20 min were recommended for occupancy durations of 30–45 min for classrooms, meeting rooms, conference rooms, etc.

In summary, wherever possible occupancy should be reduced (by remote working and/or reduced occupancy), additionally unoccupied periods should be introduced at regular intervals throughout the day during which the space should be purged and after which the space should be cleaned. Risk-based calculations are needed to determine the optimal purging times.

## (e) The influence of occupancy behaviour on indoor air movement: implications for the spread of COVID-19

The movement of people within enclosed spaces leads to considerable disturbance of the air and any airborne aerosols. Although aerosols greater than about $10\,\mu m$ might, in some settings, settle from the space relatively rapidly and so they may be dispersed by people movement, smaller aerosols (e.g. less than $5\,\mu m$) which can remain airborne for several tens of minutes may be much more strongly affected by such dispersal [70]. This dispersal may in fact dominate other dispersal processes if there is a sufficient frequency of people passing through a space. People have widths typically in the range 0.3–0.5 m, and move at speeds of $1–2\,m\,s^{-1}$ even when walking, and this leads to a highly turbulent wake with Reynolds number of about $10^5$. The mixing associated with this wake in corridors, supermarket aisles, meeting rooms, school classrooms or other spaces with relatively high people density and movement (even with 2 m spacing) may be key for quantifying the aerosol dispersion prior to it being ventilated or settling out from the space [70]. Indeed, with ventilation timescales of 10–20 min in typical buildings, with 5–10 air changes per hour, and the settling time of small aerosols (less than $10\,\mu m$) being of comparable duration, the aerosols may be mixed by the wakes of many people. For example, in a supermarket, with one person moving down an aisle every 10–30 s, a cloud of infected aerosol may be mixed by the wakes of between

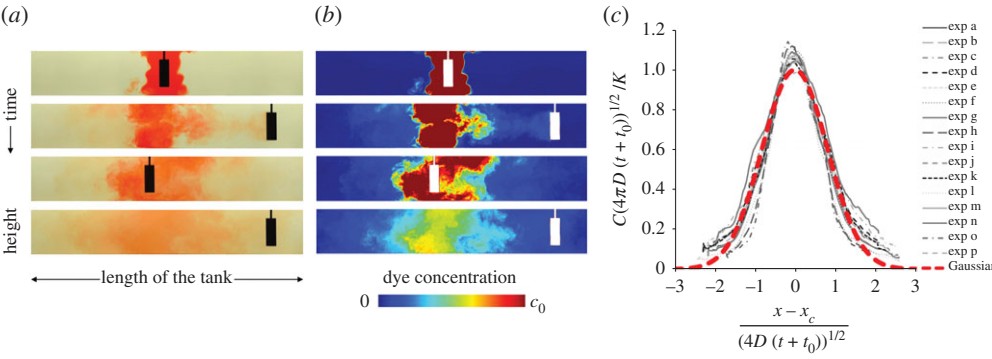

**Figure 1.** (*a*,*b*) Images show the depth-averaged concentration of a cloud of dye in a laboratory tank. This evolves in time owing to the mixing produced by the repeated motion of a cylinder, representing movement of people in a corridor. In (*a*), pictures of the tank are shown as captured during an experiment. In (*b*), false colours are used to represent the dye concentration field in each of these pictures, with red being the maximum concentration and blue showing absence of dye. The white rectangle on each image in (*b*) illustrates the position of the cylinder at that time. (*c*) Collapse of the experimental data of the depth-averaged dye concentration to a continuum model of the concentration of the dye along the channel. The *y*-axis shows the dimensionless concentration, at each time scaled relative to the theoretical value in the centre of the channel at that time, and the *x*-axis shows the dimensionless distance along the channel, scaled with the predicted diffusive spreading along the channel at each time (after [70]). (Online version in colour.)

20 and 120 people. This mixing leads to a more uniform, but smaller concentration of aerosol in space, thereby increasing the risk of exposure to some aerosol for the subsequent people that pass by the aisle, although the amount of aerosol may be smaller; the associated risk of infection from any virus in these aerosols depends on dose and hence the amount of aerosol to which they are exposed.

Laboratory simulations of the dispersal of both clouds of dye and suspended particles have been carried out in a fluid-filled channel of size $1.04 \times 0.10 \times 0.20$ m, as a model of a corridor. To model the movement of people, cylinders of radius 0.015–0.050 m were moved back and forth along the channel, with speeds of 0.1–0.2 m s$^{-1}$, thereby providing a dynamically similar flow regime for the full-scale flow of people's wakes. Data show that the motion of the cylinder leads to an effective dispersion coefficient for the along-channel mixing, which provides the basis for a theoretical model [70]. In experiments with a background ventilation along the corridor, in addition to the people-driven mixing, a dilution wave migrates along the corridor after the release of infected aerosol along the corridor, but the dispersion associated with people movement causes the aerosol to mix back upstream into the dilution wave, delaying the effectiveness of the ventilation. In figure 1*a*,*b*, images at successive times from an experiment illustrate the dispersal of a cloud of dye along the channel. In figure 1*c*, the concentration data from a number of experiments collapse to a simple model for the dispersion.

Scaling up to a building, we find that the typical mixing rates associated with people in a corridor/supermarket aisle depend on the frequency of passage of people. With a person walking along the corridor/aisle every 10–40 s [71,72], this corresponds to a dispersion coefficient in the range 0.05–0.2 m$^2$ s$^{-1}$ [70], so that over a period of 600–1200 s, airborne aerosol may spread about 5–15 m along the corridor.

Calculations by Bhamidipati and Woods based on [70] developed a simple computational model for the dispersal of aerosols in a building by a stream of individual people moving through a building, modelling the detailed mixing produced by each person. This has led to a series of simulations of the dispersal of clouds of aerosol as people pass along a corridor. Figure 2 shows the concentration of aerosol with time, over a period of 15 s as an infected person walks down a corridor breathing out. A series of local clouds of aerosol-laden air are breathed out by the infected

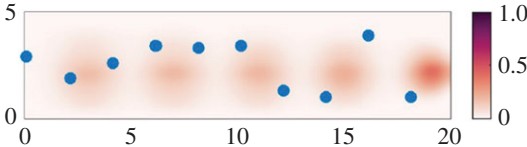

**Figure 2.** Image showing the mixing of individual clouds of infected aerosol, dyed different shades of red and normalized relative to the initial concentration as seen on the legend. The corridor is 5 m wide ($y$-axis) and 20 m long ($x$-axis), with people (blue dots) moving from left to right along the corridor. In this simulation, the along-corridor people spacing is 2 m and they move with speed 1.5 m s$^{-1}$, while the clouds of aerosol are produced by one person moving down the corridor, so the older cloud at the left-hand end of the corridor is more dispersed than that on the right (calculations by Bhamidipati and Woods based on [70]). (Online version in colour.)

person, and these are then dispersed owing to the mixing produced by the continuing stream of people following in the wake of the infected person.

Models of different building types are under continuing development, but the key result is the efficacy of mixing produced by the movement of people, which can lead to widespread dispersal of small aerosol produced by an infected person. This has leading-order implications for the occupancy levels in shops and in the corridors of classrooms and offices in terms of the risks of exposure to the small aerosols due to the repeated passage of people. The combination of aerosol settling and ventilation of the air from the space typically leads to a residence time of the small aerosols of several tens of minutes; if these small aerosols are present in sufficient numbers to play a role in infection transmission, which depends on the source strength (i.e. the number of infectious people present), and the residence time (i.e. as regulated by the ventilation rate), then the continued presence of the infectious and healthy people may provide a pathway for transmission.

Further experiments have been carried out in several operational buildings in which localized dilute clouds of $CO_2$ are released at a point in a room or corridor, and the subsequent spreading of this cloud is then measured over time; the $CO_2$ acts as an analogue for the small aerosols in that it moves with the air flow in the space [73]. Comparisons have been made between the rate of dilution and flushing of the $CO_2$ from the space in the case with people moving and with no people moving in the space. In the case of a ventilated corridor, for example, the impact of the people moving along the corridor is to drive additional mixing of the cloud of $CO_2$ into the new ventilation air, thus delaying the flushing of the $CO_2$ from the space. Figure 3$a$ illustrates the results of an experiment, which was carried out in a ventilated corridor in the cystic fibrosis ward of the new Royal Papworth hospital in Cambridge [73]. A small pulse of $CO_2$ gas was released at the end of the corridor, and the evolving concentration of the $CO_2$ was measured at three locations along the corridor using an array of sensors. It is seen that when the corridor was empty (black lines in figure 3$a$), the concentration of $CO_2$ initially increased as the gas was transported by the ventilation flow past the sensors; the concentration then gradually decayed owing to the continuing ventilation. The experiment was then repeated, with one person continuously walking back and forth along the corridor (red lines in figure 3$a$). Now, the mixing driven by the movement of the person resulted in additional dispersal and mixing of the $CO_2$ tracer, as illustrated diagrammatically in figure 3$b$. As a result, the residence time of the $CO_2$ in the corridor increased by about five times [73]. This illustrates how the internal mixing processes of the air in buildings, and especially people movement, can lead to greater residence times for small aerosols. To mitigate the risks of these small aerosols, potential solutions include increasing ventilation rates, reducing the duration of exposure, and reducing the source of the aerosols by reducing occupancy levels and through the use of face masks.

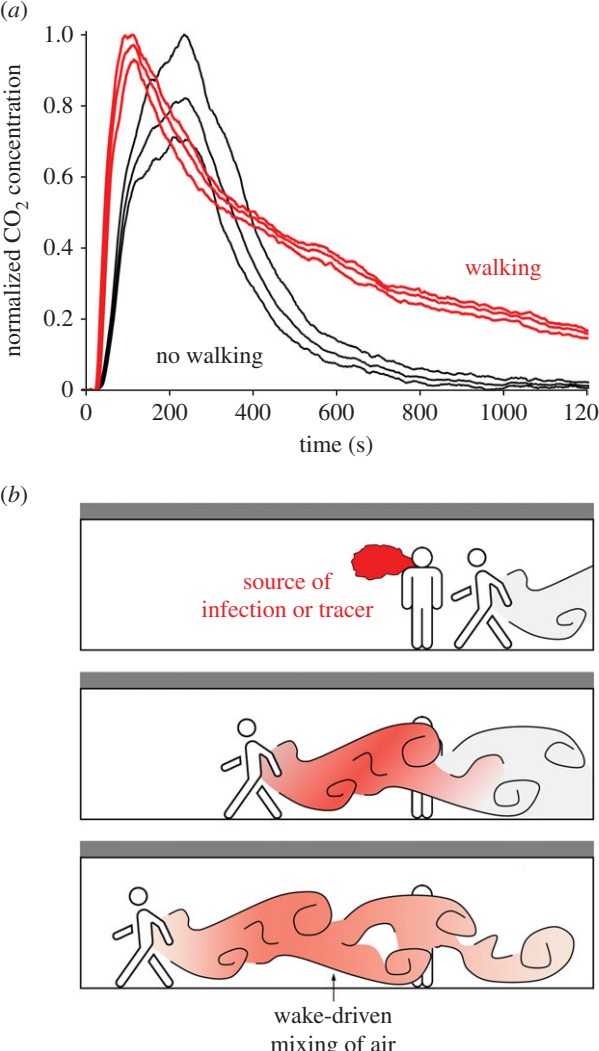

**Figure 3.** (a) Time series of the concentration of $CO_2$ as recorded by three sensors located in a ventilated but empty corridor in the cystic fibrosis ward of the new Royal Papworth Hospital in Cambridge (UK), after the local release of a pulse of $CO_2$ at one end of the corridor (black lines). The experiment was repeated with one person continuously walking back and forth along the corridor, and the red lines show the $CO_2$ concentration obtained in this second experiment [73]. (b) Schematic illustrating the effect of the mixing associated with the wake of a person walking along the corridor. (Online version in colour.)

## 3. Ventilation and the airborne transmission route

The adequate ventilation of a building space should be regarded as the primary mitigating measure against the spread of airborne diseases. There is strong evidence that COVID-19 can be spread via the airborne route (e.g. [74]) and, now, wider spread acceptance that airborne transmission is significant [5,75,76]. In temperate climates, this leads to the simple advice that all ventilation (by which we refer exclusively to the supply of outdoor, or suitably sterilized or filtered, air) systems be operated to maximize supply and ventilation openings (e.g. windows, vents, louvres, doors, etc.) be opened to the extent permitted by design. However, in colder periods (like the British heating season) there is a conflict between reducing airborne infection risk, by increasing the outdoor air supply, and maintaining occupants' thermal comfort and

reducing energy consumption and the associated costs. In this section, we seek to provide guidance to resolve this conflict.

In order to quantify precisely the risk of airborne infection occurring it is necessary to determine occupants' exposure to airborne virus particles. To do so rigorously is challenging (often to the point of impracticality see §3a) and thus it is wise to focus efforts on estimating the likelihood, perhaps better the change in likelihood, that airborne infection may occur within a space under a number of scenarios under the assumptions made. The modelling of airborne infection is a rich field in its own right, but broadly speaking two main approaches are typically adopted; namely, the 'dose-response' and 'Wells-Riley' approaches. Sze To & Chao [77] provide a detailed review of both approaches. However, since dose-response models typically require more detailed parameterization of the immune response to the pathogen, to date, most studies of the airborne infection for COVID-19 have generated estimates of the risk using models based on the Wells-Riley [78,79] approach.

## (a) Towards an understanding of ventilation and COVID-19

Ventilation, i.e. the supply of outdoor air, dilutes any pollutants produced indoors—including viable airborne virus particles. This dilution is dominated either: by the incoming outdoor air mixing with the existing indoor air, with then further mixing occurring as the air is transported through the building space before the ultimate evacuation outdoors; or by seeking to introduce the outdoor air into the occupied zone of the building space in a relatively unmixed state and then 'displace' any air polluted by airborne virus particles back outdoors (e.g. [57,80–84]). Irrespective of the intended strategy, mixing will occur inhomogeneously within the building space which results in unpredictable distributions of virus particles. Moreover, differences in temperature between indoor and outdoor air and the production of heat within the building space (every occupant outputting approximately 100 W of heat) exacerbate the complexity of indoor air flows and increase the unpredictability of the distribution of airborne virus particles. Finally, in all indoor spaces (perhaps excluding COVID-19 hospital wards) it is unknown if there are any infectious occupants (the sources) and where they might be located. As such, trying to predict the distribution of viral contamination, for example, using normal exhaust ventilation design techniques (which require knowledge of the source location and rate of contamination) is likely to be futile. Therefore estimating the risk of infection via modelled data typically requires the assumption that the distribution of airborne virus laden particles is relatively uniform irrespective of reality. We note that in the case that 'dead' or 'stagnant' zones can be expected or evidenced within an indoor space then further considerations are required see, for example §4 for further discussion.

As such, when the likelihood of an infector being present is approximately proportional to the number of occupants, the critical input pertaining to the control of the environmental quality for any model estimating airborne infection risk within an indoor space is the bulk supply of outdoor air, or ventilation rate, per person which we denote $Q$. However, outdoor air may enter an indoor space via a ventilation system, windows, doors, vents, cracks in the building fabric or, indeed, though the very fabric itself (i.e. many building materials, e.g. bricks, are porous). As such, there is significant scope for both intentional and unintentional supply of outdoor air. Directly measuring the air flow through all of the potential pathways for any given indoor space is challenging. Pressure testing can be used to measure infiltration rates but cannot assess the ventilation rates in operational settings and does not help with individual spaces or zones. For purely mechanically ventilated, well-sealed, buildings it may be tempting to assume the intended outdoor air supply is actually achieved; however, such an assumption is not without risk. Measurements of the actual air flows could be made in the duct work but these are not without their own challenges (the velocity profiles of air flows in these typically tortuous ducts is a challenging area of fluid mechanics in its own right).

In most indoor environments, the dominant source of $CO_2$ is human activity and the level of $CO_2$ in outdoor air remains broadly constant (at about 400 ppm). As such, it may be tempting

to try to infer outdoor air supply (ventilation) rates by monitoring $CO_2$. However, while $CO_2$ is an excellent proxy by which to determine indicative levels of ventilation, rigorously determining the precise (typically transient) outdoor air supply rate is non-trivial (e.g. see the appendices to Burridge et al. [85]).

Crucially, Rudnick & Milton [86] established a methodology, based on the Wells–Riley model [78,79], which takes monitored $CO_2$ data (the equipment to do so costing less than a couple of hundred pounds per sensor) and directly infers the risk of airborne infection without the need to assess/assume the ventilation rate (nor does it require the space to be in steady-state). Their model, assuming the presence of an infector, can be parametrized for any airborne disease to provide both the likelihood of infection and the expected number of secondary infections arising within an indoor spaces over time periods selected such that the space remains constantly occupied.

In addition to the monitored $CO_2$ the risk reported by Rudnick & Milton [86] depends strongly on the time period of assessment, the occupancy level, the nature of the virus and occupancy activity—the latter two aspects being parameterized via the rate of generation of infectious quanta $q$ per person (usually expressed in quanta per hour). Wells [79] conceived the idea of a quantum (or infectious dose) in an effort to describe the stochastic behaviour of airborne infection, and values for the quanta generation rate have been derived for SARS-CoV-2 (e.g. [87]). Just as for most other airborne diseases, there is wide variation in the values relevant for an infectious individual depending on (a) their activity level, (b) the viral load in their sputum and (c) the ratio between one infectious quantum and the amount of infectious RNA $\mathrm{ml}^{-1}$. However, with such a novel disease uncertainties are compounded by our relative lack of experience and data concerning COVID-19. For Wells–Riley-based models, when the environment is in steady state and the infection risk is low then a reasonable approximation for the likelihood of airborne infection is obtained from $P = 1 - \exp\left[-(I/n)(p/Q)\, qT\right] \approx (I/n)(p/Q)\, qT$, where $I/n$ is the fraction of occupants infected, $p/Q$ is the ratio of the breathing (pulmonary ventilation) rate and the outdoor air supply rate per person, and $qT$ is the number of quanta emitted during the time period of assessment (Rudnick & Milton [86] express the ratio $p/Q$ via the ratio of the excess $CO_2$ within the space and the $CO_2$ added to exhaled breath). This approximation illustrates that when the risk is low then the likelihood of airborne infection increases linearly with the time period of assessment and the quanta generation rate (for which there is great uncertainty regarding appropriate values) but the likelihood also inversely proportional to the total ventilation rate $nQ$, making this a key environmental parameter. At moderate and high levels of infection risk, this crude linear approximation over estimates the likelihood of airborne infection significantly. Buonanno et al. [87] report that in typical scenarios with low activity levels, $q \approx 1\,\mathrm{h}^{-1}$ may be appropriate for COVID-19. Buonanno et al. [87] further suggest that should an individual be vocalizing (in a manner not dissimilar to talking) while carrying out light exercise (e.g. walking) then values as high as $q \approx 100\,\mathrm{h}^{-1}$ can be inferred—for offices occupied and ventilated at appropriate levels (e.g. approx. $10\,\mathrm{m}^2$ per occupant and the outdoor air supply per person is $Q \approx 10\,\mathrm{l\,s}^{-1}\,\mathrm{p}^{-1}$ [88]) then for occupancy periods of constant occupancy of, say, 4 h the expected number of secondary infections (secondary infection numbers, $S_1$) within the office (assuming a floor-to-ceiling height of between 3 and 4 m) would be around 4, i.e. from a single infector then four new COVID-19 infections would result. For more typical behaviour in an office, i.e. most occupants carrying out desk work with a few talking relatively quietly for which $q \approx 1\,\mathrm{h}^{-1}$, then even assuming the office remains constantly occupied for a full $9\,\mathrm{h\,d}^{-1}$, secondary infection numbers of around 0.1 are obtained (these rise to around 0.2 if the office is poorly ventilated, i.e. $Q = 4\,\mathrm{l\,s}^{-1}\,\mathrm{p}^{-1}$, and drop to around 0.06 if the ventilation is doubled).

The results of Buonanno et al. [87] take a value of $q = 142\,\mathrm{h}^{-1}$ and show the airborne infection risk for various public indoor spaces (namely, shops and restaurants), reporting the expected number of secondary infection numbers for differing exposure scenarios (changing outdoor air supply and occupancy scenarios) which they describe as before and after 'lockdown' (Note that they do not report values for restaurants after lockdown). They select modelled durations of

**Table 1.** The expected number of secondary COVID-19 infections arising, $S_I$, calculated over the period (5 working days) that a pre/asymptomatic person remains attending work in an open-place office (floor plan of 400 $m^2$ and floor-to-ceiling height of 3.5 m) occupied by 40 people for 8 h each day [85].

| secondary infections, $S_I$ | $Q = 4\,l\,s^{-1}\,p^{-1}$ | $Q = 10\,l\,s^{-1}\,p^{-1}$ | $Q = 20\,l\,s^{-1}\,p^{-1}$ |
|---|---|---|---|
| $q = 0.3\,h^{-1}$ | 0.25 | 0.13 | 0.07 |
| $q = 1\,h^{-1}$ | 0.84 | 0.42 | 0.24 |
| $q = 5\,h^{-1}$ | 4.0 | 2.1 | 1.2 |
| $q = 20\,h^{-1}$ | 14 | 7.6 | 4.4 |
| $q = 100\,h^{-1}$ | 35 | 26 | 18 |

approximately 3 h. For poorly ventilated spaces, Buonanno *et al.* [87] report secondary infection numbers ranging from $2 \leq S_1 \leq 50$ before lockdown ($1.7 \geq Q \geq 0.2\,l\,s^{-1}\,p^{-1}$) and after lockdown $0.1 \leq S_1 \leq 0.8$ after ($10.5 \geq Q \geq 5.2\,l\,s^{-1}\,p^{-1}$). For better ventilated spaces then $1 \leq S_1 \leq 6$ ($10 \geq Q \geq 4.5\,l\,s^{-1}\,p^{-1}$) before lockdown and afterwards $0.1 \leq S_1 \leq 0.4$ ($55 \geq Q \geq 22\,l\,s^{-1}\,p^{-1}$).

Since an occupant can become infected at any point on any given day then taking any particular duration seems an arbitrary choice which, for the most part, is hard to justify. Burridge *et al.* [85] point out that for indoor spaces which are regularly/consistently attended by the same/similar group of people (e.g. open-plan offices or some school classrooms, and herein a 'regularly attended space') one should consider the likelihood of infection over a period during which an infectious person may remain pre/asymptomatic. For COVID-19, this period is currently estimated to be between 5 and 7 days. They developed simple extensions to the work of [86] which enabled variations in occupancy behaviour and activity to be accounted for and the likelihood of infection to be assessed from monitored $CO_2$. In doing so, Rudnick & Milton [85] calculate the likelihood that an indoor space contributes to the spread of COVID-19 by assuming that a single pre/asymptomatic infector regularly attends the space and ceases to do so once they show symptoms reporting the expected number of secondary infections arising, $S_I$, for modelled and monitored spaces.

As shown in table 1 for typical behaviour in a typical office Burridge *et al.* [85] report $S_I \approx 0.42$ which rises to $S_I \approx 0.84$ for a poorly ventilated space ($Q = 0.4\,l\,s^{-1}\,p^{-1}$) and falls to $S_I \approx 0.24$ if the ventilation is doubled—which are reassuringly below one, thereby suggesting that the return to desk-based office work is unlikely to contribute significantly to the COVID-19 spread. However, even if the office workers remain sedentary but start vocalizing (akin to a call-centre, or similar) then $S_I \approx 2$—highlighting the importance of occupancy behaviour in determining whether or not a particular indoor space contributes to the spread of COVID-19 or not. Not unsurprisingly, there is still great uncertainty regarding the nature of transmission and infection for the disease COVID-19, not least due to the apparent propensity for mutations to give rise to new variants which are more transmissible. For example, data for the B1.1.7 variant suggests it is approximately 70% more transmissible than pre-existing strains [89,90]. Should this increases apply equally across all transmission routes then the secondary infection numbers reported in the first three lines of table 1 would increase by around 70%. Difficulties in robustly parametrizing and modelling infection via the airborne route does not escape these uncertainties. For example, within the literature discussed above quanta generation rates differing by at least two orders of magnitude are taken to evaluate the risk. Efforts to reduce these uncertainties are ongoing; however, while the absolute numbers vary with such modelling choices the predominant conclusions that fewer people in well ventilated spaces for shorter durations will reduce transmission by the airborne route do not change. Burridge *et al.* [85] also present results for the airborne infection risk from $CO_2$ data monitored within an office with uncontrolled natural ventilation. The particular office is not of a modern well-sealed design. The risk levels within that monitored naturally ventilated office remain comparable with the modelled office, intended as being 'typical', therein. However, the airborne infection risk was calculated for periods when the windows were opened and for those

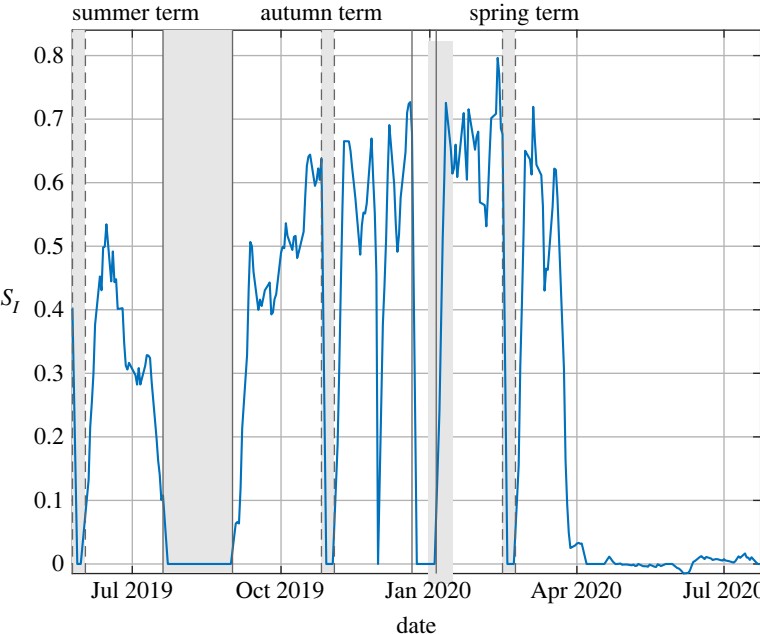

**Figure 4.** The variation in the absolute expected number of secondary infection, $S_I$, during the period 25 May 2019 to 20 July 2020 determined from monitored $CO_2$ and the school's occupancy timetable for a typical classroom within a relatively modern build school (rebuilt in September 2016). The shaded regions correspond to holiday periods. For data recorded during the summer term of 2019 the average was $S_I = 0.35$, for the autumn term of 2020 the average was $S_I = 0.53$, and for data recorded during the spring term of 2020 the average was $S_I = 0.63$ until the lockdown of 20 March.

when the windows remained shut—the risk of infection was approximately doubled when the windows remained shut. This provides quantitative support for the guidance that where practical ventilation openings (like windows, doors, etc.) should be opened, but doing so is not always without conflict, see the electronic supplementary material, appendix A for further discussion. This is particularly pertinent during wintertime. As shown in figure 4, analysis of the $CO_2$ levels monitored in schools indicated that just due to changes in the supply of outdoor air the airborne infection risk in winter is approximately double that in summer.

Such findings beg the question: how might the infection risk vary between seasons as weather changes from being relatively temperate and what might this mean for the spread of COVID-19 in winter 2020/21? Vouriot *et al.* [91] examined the expected number of secondary infections within various spaces inside schools. Their most meaningful results are for classrooms, which can be regarded as regularly attended spaces, where they reassuringly find secondary infection numbers below one if pupils are assumed to be carrying out desk-based learning in a relatively calm/quiet environment and quanta generation of $q \approx 1$ are appropriate. The secondary infection numbers rise to $1.5 \leq S_I \leq 3.8$ if one assumes values of $q \approx 5$ which are more appropriate if the class is carrying out more vocal activities, and they rise further still if one assumes pupils are actively moving around the classroom. Crucially, in all cases, the greatest secondary infection numbers are obtained during colder winter months (for example, November to February)—being typically around 80% greater than those estimated for more temperate months (i.e. May to September). We see no reason not to expect similar trends in the secondary infection number of indoor spaces beyond schools and such convictions underpin our guidance to assess and ideally monitor ventilation provision in order to understand the needs for modification and the implementation of other measures.

## (b) Ventilation guidance for wintertime

A concern for wintertime is that buildings become less well ventilated with a lower supply of outside air in order to maintain warm conditions indoors. The spread of COVID-19 has been prevalent in domestic settings, which frequently have no mechanical ventilation provision, and we recommend that keeping windows/vents even slightly open (or where available using, so called, 'trickle vents') may help in reducing the spread, especially within shared bedrooms.

Ventilation systems that recirculate a proportion of the indoor air, primarily to temper the temperature of the outdoor air without increasing energy consumption, are common but it is crucial to regard only the flow of outdoor air as contributing to the ventilation rate (see the section entitled 'In-duct UVGI air disinfection' within the electronic supplementary material, appendix B for more detailed discussion). We recommend evaluating the benefits of increased monitoring of the indoor environment, especially indicative outdoor air supply rates via $CO_2$ levels. Much work has assessed the effects of $CO_2$ on human physiology with no negative consequences being associated with low $CO_2$ levels and some degradation of performance at high levels (e.g. [92,93]). However, simply recommending $CO_2$ levels be kept as low as possible is impractical due to fixed heating capacities and other constraints. $CO_2$ levels not exceeding 1000 ppm within an indoor space broadly indicate that the outdoor air supply is likely to be adequate for offices and mechanically ventilated classrooms (i.e. taking a person's $CO_2$ production rate to be approximately $6\,\mathrm{ml\,s^{-1}\,p^{-1}}$ and $CO_2$ levels in the ventilating outdoor air as 400 ppm gives $Q \approx 10\,\mathrm{l\,s^{-1}\,p^{-1}}$), with the equivalent level being 1500 ppm in naturally ventilated school classrooms (i.e. $Q \approx 5\,\mathrm{l\,s^{-1}\,p^{-1}}$, see Annex A of [2], for a full discussion). Doing so will assist in: (a) highlighting high-risk spaces for which mitigation measures need to be considered (e.g. enhanced ventilation provision to manage certain activities or some of those measures detailed in §4), and (b) helping obtain the quantitative evidence to reassure occupants as to their relative safety. Where monitoring of $CO_2$ is undertaken, consideration of the build-up and decay of $CO_2$, and the variation in $CO_2$ levels within the space can provide additional insights; where experience or skills are lacking, engagement with a professional building services engineer would be beneficial. We note that the design of school classrooms will, in some cases, have been carried out allowing $CO_2$ levels to reach up to 2000 ppm for brief periods; as Vouriot *et al.* [91] show, this does not necessarily indicate unacceptable risk levels for COVID-19. Moreover, active management of classroom windows and other ventilation openings may enable these high peaks to be avoided or minimized; appendix A within the electronic supplementary material provides discussion of strategies to do so in winter without excessively impacting occupant thermal comfort.

Where people are brought together in moderate to high densities for significant portions of the day and no monitoring is intended, we recommend that the design provision for ventilation of the indoor space be investigated to determine whether adequate ventilation is likely being achieved. Consideration should be given to installation and the subsequent maintenance to estimate whether or not the design provision might realistically be attained. Where the design provision is unavailable, efforts should be made to 'reverse engineer' an understanding of the ventilation. In the absence of knowledge as to the number of infected occupants, ventilation provision should always be considered per capita based on the design occupancy—we ventilate buildings for the sake of the occupants. Decreasing occupancy density should be considered both to minimize transmission via the droplet route, the fomite route (see §2b) and, crucially, the airborne route as the *per capita* ventilation provision from fixed systems/plant can generally not be drastically increased without new investment (see §2d). Where ventilation rates meet existing design guidance the expected risk of airborne infection might be regarded as low. In the absence of monitoring, consideration should be given to potential stagnant zones within indoor space (e.g. a sheltered reading corners or break-out spaces); where suspected these should be addressed (see §4).

Where $CO_2$ monitoring is carried out, sensors should be placed at heights and locations within the space representative of the breathing zone. Where practical, occupancy and excess $CO_2$ (i.e. $CO_2$ above outdoor levels) should be recorded and simple calculations undertaken to estimate the

level of risk to occupants (e.g. [85])—we hope that, in the most part, these calculations will prove reassuring. Where no calculations are desired, then for given activity within a space, it is worth noting that the airborne infection risk depends directly on the excess $CO_2$ relative to the number of persons responsible for producing it—this latter quantity being impossible to evaluate without monitoring (in which case the simple calculations of Burridge *et al.* [85] may prove more useful). In all cases, we suggest that consideration not only of risk but also the rate at which the risk is increasing with time is prudent. The relative rate of increase in airborne infection risk is directly proportional to the ratio of the instantaneous value of excess $CO_2$ and the current occupancy [85]. These values are easily obtained and indicate the level of risk that will be realized should all else remain equal.

We note, however, that ventilation flows are complex and may involve air flow from one space in a building to another. They are also strongly influenced by the locations of openings and the strength of internal heat gains. Further work on the implications of these effects is needed.

Finally, we conclude that the risk of COVID-19 being spread by the airborne route is not insignificant, varies widely with activity level and environmental conditions (which are predominantly determined by the bulk supply of outdoor air), and is expected to increase in winter relative to summer. These conclusions are based on data relating to the SARS-CoV-2 strains that were dominant throughout 2020. However, throughout the later part of 2020 and into 2021 multiple mutations to the SARS-CoV-2 virus (see e.g. [94]) have conspired to give rise to new variants which are significantly more transmissible (for example variants B1.1.7 and 501Y.V2). Should new variants become prevalent then the ventilation provision in many indoor spaces may need evaluation in light of the new data.

## 4. Other mitigating measures

This section explores some of the additional measures available to mitigate risk of airborne transmission of COVID-19 in indoor environments. Through the present work, we use 'airborne' transmission to refer to transmission via smaller particles which can be suspended in the air for a considerable time. By 'droplet' transmission, we mean short-range transmission via larger droplets which fall to a surface within seconds and within a few metres of the infected person.

The risk of airborne transmission indoors can be mitigated through dilution of the indoor air by clean outdoor air, as discussed in §3. This requires a ventilation system for which the air intake can be increased, or installing secondary ventilation systems. Substantially increasing the ventilation in a space is often impractical. Further, in order to reduce the risk of infection by a factor, the ventilation rate must be increased by the equivalent factor. Increasing the volume of outdoor air becomes particularly challenging in winter without compromising the thermal comfort of occupants or energy use (due to an increased heating load). These factors limit the viability of reducing risk by increasing ventilation.

The mechanisms for the transmission of COVID-19 are not yet well understood. Reducing the risk of infection to zero is not possible (short of global eradication), therefore, where practical to do so, all available measures should be taken in order to gain the biggest reduction in risk. In the absence of or to complement increased ventilation, alternative engineering control measures can be used. Some of the measures available are discussed here.

### (a) Air filters or cleaners

Filters and air cleaners, such as UV cleaners or photocatalytic oxidation, can either be installed directly within the existing ventilation system, removing virus-laden particles from recirculated air, or as independent units within a room to supplement the existing ventilation.

In their latest guidance document in response to the pandemic, The Federation of European Heating, Ventilation and Air Conditioning associations (REHVA) recommend avoiding the use of centralized recirculation as typical local air filters within these systems are not effective at filtering out viral material which tends to be too small for the filter [95]. Installing high-efficiency

particulate air (HEPA) filters would allow some virus-laden particles to be removed, however these are not easily installed in existing systems and further system modifications are required in order to provide a higher pressure drop to maintain the same airflow rate.

Standalone air filters can be effective at removing virus-laden particles provided they target the appropriate range of particle size. However, regardless of the efficacy of the filter itself, the supply of clean air is limited by the flow rate of air passing through the filter. Despite this, the supply of clean air in a space can be up to $1000\,m^3\,h^{-1}$ [96]. However, filters require regular servicing to maintain their effectiveness—clogged filters lead to a build up of viral particles and can act as a source of viral matter rather than a sink [97].

Air cleaners such as UV air cleaners are also limited by the volume of air that can be passed through the device. UV light has been shown to be effective in deactivating various viruses under laboratory conditions, including coronaviruses [5,98]. While the evidence that UV is effective against SARS-CoV-2 specifically is currently limited, it seems highly likely to be the case [98]. The use of UV lamps can lead to the generation of ozone and other harmful by-products [99,100]. The potential impact of ozone generation on occupant health should be fully assessed in advance of installing any air cleaner which uses UV lamps.

The efficacy of air filters and cleaners is likely to be highly sensitive to their location within a room. Depending on the size, shape and airflow patterns within a room, the air within some areas of the room may never reach the device. In the worst case scenario, the device simply recirculates the same small volume of air within a much larger room. Therefore while these devices tend to promise certain air changes per hour equivalent of clean air, in reality this is only true if the air pulled in by the device has not already been cleaned. In smaller, poorly ventilated rooms such devices are likely to provide significant benefit, however for larger spaces an understanding of the airflow patterns within the room is required to ensure that the device is effective.

Care should be taken when considering which air cleaner/filter to use as many devices have been found to have a limited effect [96,99]. HEPA filters are often recommended as the most effective technology currently available (e.g. [96]).

## (b) Personalized ventilation

Personalized ventilation (PV) supplies clean air directly to the breathing zone of an occupant of a room via a device installed at their workstation. A certain minimum velocity is required for the supplied air in order to penetrate the convective flow driven by body heat [101]. Further, a large target area is desired in order to account for occupants' movement. The required rate of clean air supply can therefore be high. While studies have shown that PV can be effective at reducing risk for occupants while at their workstations [101,102], protection is not provided to occupants when away from their workstations. PV may in fact facilitate the transport of exhaled pathogens to other occupants [103]. Installing a personalized device to deliver the required air supply to each occupant in an office is also likely to be expensive and impractical.

Alsaad & Voelker [104] consider the use of a ductless PV system in conjunction with displacement ventilation. The ductless PV is a stand-alone system, independent of the building ventilation, which transports cool air from the lower part of the room and delivers it to the breathing zone of the occupant. This provides a much cheaper system which is much easier to install. However, the CFD study presented by Alsaad & Voelker [104] suggests that the system is only effective when the PV is not used by the infected occupant. Further, a system which transports air from the lower part of the room to the breathing zone risks transporting virus-laden droplets which may otherwise deposit on the floor.

PV may be most viable in scenarios where the occupant is required to remain at their workstation for the duration of their shift.

## (c) Desk and ceiling fans

While desk or ceiling fans do not enhance the bulk supply of outdoor air, they can be used to increase air mixing within a room, which may lead to a more homogeneous distribution of virus-laden particles. Where areas of stagnant air are identified (or are suspected), a fan can be used to increase mixing with the wider space, therefore potentially reducing the risk of the accumulation of virus-laden particles within the stagnant zone.

The localized (relatively) high velocity air flows produced by desk fans may result in increased re-suspension of virus-laden particles and this risk should be considered. However, studies have found that using either desk or ceiling fans to increase air mixing within a room can lead to increased rates of deposition [105–107]. These experiments use oil droplets [106], cigarette smoke particles [107] and a combination of oil droplets, salt and incense [105] under controlled laboratory conditions. However, deposition rates can vary by orders of magnitude depending on the particle size, room surface-to-volume ratio (which was varied predominantly via the inclusion of furniture) and airflow speeds and turbulence within the space. Generally, the removal of airborne particles through deposition is likely to be much lower than that through ventilation, however for poorly ventilated rooms these rates can be comparable, particularly for larger particles (greater than $1\,\mu m$). It is unclear whether the results of these studies would translate to increased deposition rates of COVID-19-laden particles under real-world conditions. The impact of changing deposition rates on the risk of surface transmission via fomites is also unclear.

Owing to this uncertainty and the potential risk of increasing the re-suspension of virus-laden particles within the space, the use of desk fans is generally not recommended. However, when a stagnant zone is evident, either via intuition (e.g. sheltered spaces like reading corners within classrooms and breakout spaces within open-plan offices) or through monitoring $CO_2$ concentrations, the benefit of using a desk fan to increase mixing with the wider space may outweigh the potential drawbacks. Where fans are used in this capacity they should be orientated such that increased mixing is achieved between the stagnant zone and the surrounding space (into which the outdoor air provision should be checked).

Ceiling fans promote vertical mixing of air within rooms and, in addition to increased bulk mixing, help reduce temperature stratifications from forming within the room. As such where the ventilation strategy relies on stratification within the room, e.g. where a displacement ventilation strategy can be successfully evidenced, the use of ceiling fans is not recommended. Otherwise, temperature stratification within a room could significantly inhibit the vertical mixing, and dilution, of virus-laden particles from within the breathing zone and therefore the use of a ceiling fan is likely to be beneficial. Moreover, in the heating season the downwards mixing of warmer air from the ceiling can enable the increased supply of outdoor air without compromising thermal comfort (nor increasing the energy consumption associated with heating). Further, their use in conjunction with upper room UV (see §4f) has been found to greatly increase the exposure of virus-laden particles to the upper region of the room [5,108].

## (d) Air ionization

Air ionization is a relatively new technology and involves the production of ions such as the hydroxyl radical ($OH^-$) from a corona discharge between two high potential electrodes. These ions have been shown to have germicidal properties as they react with the surface structure of the pathogen [103]. The technology was shown to be effective against certain bacterial pathogens when installed in a hospital intensive care unit, but with no effect on others [109]. Ionizers are easy to deploy and have high energy efficiency but have not been evidenced as effective devices against viruses including SARS-CoV-2. Some devices are known to produce by-products such as ozone so their effect on air quality should also be considered [99,110]. A recent study concluded that associated by-products other than ozone can also adversely affect the respiratory and cardiovascular systems [111].

## (e) The use of screens

Using screens to provide a physical barrier between the occupants of a space is a simple and easily applied measure to mitigate the risk of transmission. The use of screens is widespread in hospitals and the commercial sector; in supermarkets they are used to provide a barrier between the shop assistant and the customer.

However, this application is targeted at reducing the risk of infection via larger droplets. There are very few examples of the use of screens to mitigate airborne transmission (i.e. smaller droplets or droplet nuclei), either by providing a barrier directly between occupants, or in an attempt to favourably manipulate the airflow patterns within a space. Noakes *et al.* [112] use CFD simulations to investigate the effect of partitioning a hospital ward room on the risk of patient-to-patient and patient-to-visitor transmission. They conclude that, combined with carefully considered changes to the positioning and number of the ventilation inlets and outlets, the risk of infection can be reduced significantly. However, in some instances the risk of transmission from patient to visitor was increased by the presence of the partition due to reduced airflow in certain areas of the room.

This highlights the main problem with the use of screens indoors; the impact of the screen on the airflow patterns within a space is very difficult to predict. While the exchange of air between two areas of a room may be reduced, the presence of a screen can lead to areas of stagnant or recirculating flow where the virus could accumulate. Further to this, installing screens which affect airflow would require an evaluation of the impact on any environmental monitoring undertaken, and crucially on fire safety (e.g. as they may prevent smoke from reaching the smoke detector or have other fire safety implications).

## (f) Upper room ultra-violet germicidal irradiation

Upper room UV provides a way to use the deactivating properties of UV light without the limitation of the airflow rate through a cleaning unit. While real-world studies of its application are limited (e.g. [113]), laboratory and modelling studies suggest that, provided sufficient efficacy of UV in deactivating the virus, the method can be effective at mitigating airborne transmission (e.g. [114–116]). Upper room UV is likely to be at its most effective in poorly ventilated spaces [116] and can be installed reasonably easily at a low cost. However sizing a system is not always straightforward, and it is essential that systems are installed by professionals who also ensure that occupied zone UV-C irradiation levels are safe.

The method depends on the virus reaching the upper section of the room which is exposed to UV light. Therefore, appropriate internal flow patterns are required in order to ensure that the virus is transported through the unit and deactivated. The process is complicated by the large range in particle sizes (see §2a(i)) produced by an infected person, where larger particles require higher vertical velocities to be carried to the upper section. Therefore, while a large range of particle sizes may well be deactivated, there is a danger that a certain fraction remains.

Care must be taken when installing upper room UV as prolonged exposure to UV light can lead to damage to the skin and eyes [117,118]. As is the case when used in-duct or in standalone air cleaners, UV lamps installed in the upper room can also lead to the generation of ozone and other harmful by-products [100,113]. However, studies such as Nardell *et al.* [118] have demonstrated that the method can be implemented with minimal risk to occupants due to UV exposure. Further, 'ozone-free' UV lamps can be used to avoid adverse effects on indoor air quality due to the production of ozone. Only real-world trials will prove the efficacy of the method (see appendix B within the electronic supplementary material) and costs (of purchase, installation and maintenance) should always be compared to the equivalent costs of appropriate upgrades to the ventilation provision to the indoor space.

## (g) Summary

Each of the engineering control measures considered here have their advantages and limitations, and none are able to entirely eradicate the risk of transmission. Upper room UV holds promise as a method to significantly reduce the risk of transmission, particularly in poorly ventilated spaces. However, further effort is required to demonstrate its effectiveness against SARS-CoV-2 in real world scenarios. Air ionization is an emerging technology, but there are unresolved concerns regarding its impact on indoor air quality, primarily due to the production of ozone, and there is little evidence that it is effective against viruses. The addition of air filters that are effective against viral transmission to the existing ventilation system are likely to require complementary modifications to the ventilation system to, at the very least, account for the changes in pressure drops across the system. Independent filter units can be used, however in order to maximize their impact in larger spaces an understanding of the airflow patterns within the space is useful. Filters are an established technology which are likely to reduce risk without any detrimental impact to occupants if appropriate maintenance can be ascertained and achieved. Using desk fans to increase the mixing of air within a room is easily applied and may be a very affordable measure, however their merits are questionable in the absence of evidence of stagnant zones. Ceiling fans provide the benefit of increased vertical mixing within the room. However in the case of both desk and ceiling fans, there are uncertainties as to their impact on re-suspension and deposition of virus-laden particles and the associated impact on transmission risk. The use of screens can be effective to mitigate spread via the droplet route, but their use is problematic as a measure against airborne transmission since impacts on the circulation of air and the local ventilation provision is unpredictable without detailed bespoke study. Moreover, an assessment of a screen's impact on fire safety would be prudent ahead of any installation.

The primary control measure for airborne transmission indoors is ventilation and achieving adequate outdoor air supply rates to ensure an acceptable level of risk should be the priority. However, where additional risk reduction is required or desired other measures can be implemented to help mitigate the spread of COVID-19. Where the intention of these measures is to reduce the risk of airborne transmission their costs should always be compared to the cost of upgrading the existing ventilation provision. Moreover, these measures should always be implemented in addition to any existing measures to reduce the spread of infection via all routes of transmission, e.g. adequate hand hygiene and cleaning, the wearing of face coverings or other personal protection equipment and social distancing.

Data accessibility. This article does not contain any additional data.

Authors' contributions. This document results from the work and discussions led by P.F.L. and C.P. under Task 7 (Environmental and aerosols transmission) within the Royal Society's 'Rapid Modelling of the Pandemic project' (RAMP). H.C.B. led the compilation and editing of this document. M.S.D.W., H.D. and S.F., and P.F.L. each acted in the capacity of supporting editors; all members of RAMP Task 7 contributed directly or indirectly to the production and editing of this body of work. Further notable contributions include— production of executive summary: H.C.B.; production of text concerning COVID-19 transmission indoors and transmission via droplets and aerosols: R.K.B., M.E.J.S. and P.K.; production of text concerning transmission via surface contacts and fomites, and the appendix 'SARS-CoV-2 on hard surfaces': I.D.M., P.D., A.H., Y.J.-L., M.-F.K., O.K., A.M., P.M., T.P., M.S., D.S., P.H.T., S.K.W., C.W. and H.W.; production of text concerning social distancing indoors: M.S.D.W.; production of text concerning guidance on face masks and coverings: D.S. and C.I.; production of text concerning source reduction through timetabling, and the purging between events: H.B.; production of text concerning the impact of occupancy behaviour on indoor air movement: A.W.W., N.M. and N.B.; production of text concerning ventilation and the airborne transmission route: H.C.B.; production of text concerning other mitigating measures: H.W.; production of the appendix 'Natural inputs to ventilation provision': C.I.; Production of the appendix 'UVGI and COVID-19': C.B.; production of the appendix 'Aerosols in the context of singing, and woodwind and brass musical instruments': A.M.

Competing interests. We declare we have no competing interests.

Funding. This work was undertaken as a contribution to the Rapid Assistance in Modelling the Pandemic (RAMP) initiative, coordinated by the Royal Society.

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
