## [Peer Review File · Proceedings. Mathematical, Physical, and Engineering Sciences]

Review History

RSPA-2020-0855.R0 (Original submission)

Review form: Referee 1

Is the manuscript an original and important contribution to its field?

Excellent

Is the paper of sufficient general interest?

Good

Is the overall quality of the paper suitable?

Excellent

Can the paper be shortened without overall detriment to the main message?

Yes

Do you think some of the material would be more appropriate as an electronic appendix?

Yes

Do you have any ethical concerns with this paper?

No

Recommendation?

Accept with minor revision (please list in comments)

Comments to the Author(s)

An excellent overview - just missing some important additional references - and a suggestion:

- can the authors create a practical, one-page guide to what, say, a class teacher can do immediately or over a longer term - to enhance ventilation in a school - or a team leader for their office workers, e.g.

Option 1: Open the windows if available (quick, free, minimal disruption)

Option 2: If windows cannot open, work outside (quick, free, some disruption)

Option 3: If it is too cold to work outside and windows cannot open, increase the HVAC mechanical ventilation (quick, \$, minimal disruption - if possible, else see below)

Option 4: Consider periodic venting of the classroom/office if the windows can open but it is very cold outside (quick, free, some disruption)

Option 5: install CO2 monitors to optimise the timing of venting for a given occupancy in an office or classroom environment (needs installation, \$\$, some disruption when CO2 threshold exceeded indicating venting required)

Option 6: use portable HEPA-filtered air purifiers (needs ordering, \$\$, minimal disruption when in operation - some noise disturbance, drafts)

Option 7: install UVGI (needs installation, \$\$\$, minimal disruption after installation)

Option 8: upgrade HVAC system (needs serious infrastructure work, \$\$\$\$, severe disruption during installation, minimal afterwards)

The authors can think of additional features. Think of it as a one-page hand-out to various firms, companies, institutions - if you are offering your services to do this - as this may well be the case - as this virus will be with us for a while to come.

- there are few references cited for the mask section - some additional studies suggested below - to both contain exhaled aerosols as well as protecting against incoming aerosols:

<https://journals.plos.org/plospathogens/article?id=10.1371/journal.ppat.1003205>

Health and Safety Executive UK. RR619 Evaluating the protection afforded by surgical masks against influenza bioaerosols. 2008. <https://www.hse.gov.uk/research/htm/rr619.htm> (Accessed 4 Nov 2020)

Makison Booth C, Clayton M, Crook B, Gawn JM. Effectiveness of surgical masks against influenza bioaerosols. *J Hosp Infect* 2013; 84(1): 22-6.

Weber A, Willeke K, Marchioni R, Myojo T, McKay R, Donnelly J, et al. Aerosol penetration and leakage characteristics of masks used in the health care industry. *Am J Infect Control* 1993; 21(4): 167-73.

van der Sande M, Teunis P, Sabel R. Professional and home-made face masks reduce exposure to respiratory infections among the general population. *PLoS One* 2008; 3(7): e2618.

Davies A, Thompson KA, Giri K, Kafatos G, Walker J, Bennett A. Testing the efficacy of homemade masks: would they protect in an influenza pandemic? *Disaster Med Public Health*

Prep 2013; 7(4): 413-8.

Also, can the authors insert a section on the 'short-range aerosol' transmission route. So this is like short-range droplets - but the transmission is mainly via aerosols produced during talking, breathing (rather than coughing, sneezing - which we don't do very often and we normally cover our mouths/noses when we do), i.e. the garlic breath phenomenon over about 1 m typical conversational distance, as we think that this is how SARS-CoV-2 is mainly transmitted now.

These additional reviews may help:

Recognition of aerosol transmission of infectious agents: a commentary

Raymond Tellier, Yuguo Li, Benjamin J. Cowling, Julian W. Tang

BMC Infect Dis. 2019; 19: 101. Published online 2019 Jan 31. doi: 10.1186/s12879-019-3707-y

Airborne spread of infectious agents in the indoor environment

Jianjian Wei, Yuguo Li

Am J Infect Control. 2016 Sep 2; 44(9): S102-S108. Published online 2016 Aug 30. doi:

10.1016/j.ajic.2016.06.003

For their section on contact/ fomite transmission - following on from the above - the UK SAGE/NERVTAG has also found that contact/fomite transmission only accounts for up to about 20% of all respiratory virus transmission - likely including SARS-COV-2 also in this mini-review that they did a while ago. Can the authors incorporate this or at least some of the studies within this review:

https://assets.publishing.service.gov.uk/government/uploads/system/uploads/attachment_data/file/897598/S0574_NERVTAG-EMG_paper_-_hand_hygiene_010720_Redacted.pdf

Review form: Referee 2

Is the manuscript an original and important contribution to its field?

Good

Is the paper of sufficient general interest?

Excellent

Is the overall quality of the paper suitable?

Good

Can the paper be shortened without overall detriment to the main message?

No

Do you think some of the material would be more appropriate as an electronic appendix?

Yes

Do you have any ethical concerns with this paper?

No

Recommendation?

Accept with minor revision (please list in comments)

Comments to the Author(s)

Overall, this manuscript is well-written, with comprehensive background information that is grounded in the literature combined with evidence-based recommendations for practical engineering solutions. This is a timely and useful overview. I recommend only minor changes, easily achieved by small wording changes. The one section I encourage the authors to consider cutting down or completely moving to the Appendix is Section 3.1. It contains a very different level of detail from the rest of the manuscript, and appears to be written for a different audience from the rest of the manuscript.

Specific Comments.

1. In the discussion of indoor transmission (S2.1), the authors allude to the recent research and the focus on thermal comfort and energy efficiency. What isn't clear from their wording is the fact that strides in energy efficiency have generally come at the expense of ventilation (by making buildings less 'leaky'), and thus indoor air quality. Reducing transmission of airborne pathogens requires more ventilation - the exact opposite of what energy efficiency has required. I recommend slightly rewording the text to make this dichotomy clear.

2. Section 2 includes two separate ideas: transmission (airborne versus fomite), and then some recommendations on social distancing. The literature review of these transmission sections is strong, and the authors made the (in my opinion, correct) choice to combine aerosol and droplet transmission in one section. I deeply appreciate the authors' efforts to make the combined importance of these components clear, particularly later in the text when they discuss the fact that plexiglass barriers and the like do not prevent aerosol transmission. However, my concern/comment with this section is that the authors don't provide an estimate of the relative importance of these two modes (airborne versus surface) of transmission. From my reading of the literature, the bulk of case studies (e.g. the cruise ship study and the Skagit choir study) have shown that airborne transmission dominates. This seems like a useful point to make in terms of providing useful recommendations: vast sums of money are spent on surface decontamination and chemicals, without strong evidence that this is a substantial mode of transmission. I encourage the authors to provide some wording on their / the literature's perspective on relative importance of transmission. The importance of airborne transmission is alluded to later in the section, but not considered relative to fomite transmission.

3. In the discussion of surface cleaning, I encourage the authors to note the potential dangers of exposure to certain cleaning materials (namely sodium hypochlorite), which can produce toxic byproducts - potentially dangerous to janitorial staff, particularly if improperly used.

4. In the Surface cleaning section, the authors refer to recommendations of different approaches - rightly noting that both duration of cleaning and chemical concentration are important considerations. The authors refer to the Kampf studies. However, it would be useful to note whether these testing studies are done on true SARS-CoV-2 viruses, or if they are conducted on proxies: there is recent evidence (though unpublished, I suspect - I've heard in seminars, but not seen a peer-reviewed paper) emerging that SARS-CoV-2 may not respond to disinfectants (namely hydrogen peroxide) in the same way as its proxies.

5. In section 2.3.1, the authors discuss the importance of face coverings / masks. There is substantial research showing that different masks (materials / designs) have different efficacies as a function of aerosol size, and it might be useful for the authors to allude to that fact (i.e. some masks are better than others, but any mask is probably better than nothing).

6. Section 3.1 is far more detailed than the previous sections, and while all important information, it is much harder to follow and would be a useful place to shorten to key points and move the remaining information to the manuscript's Supplemental Information for ease of reading. It also seems out of place as social distancing and ventilation are discussed in the previous section, and this part goes into more detail on that. Instead, focusing on the recommendations for Winter 2020

would be useful.

7. In section 4.2 the authors discuss UV and personal filters. I think they must comment on the downside of improper use of UV (potential generation of high ozone levels; damage to eyes/skin if improperly used/installed). In Section 4.4 the authors allude to ionizers. Again, there are potential issues of O₃ generation, as well as other byproducts:

Zhang, Y., Mo, J., Li, Y., Sundell, J., Wargocki, P., Zhang, J., Little, J.C., Corsi, R., Deng, Q., Leung, M.H.K., Fang, L., Chen, W., Li, J., Sun, Y., 2011. Can commonly-used fan-driven air cleaning technologies improve indoor air quality? A literature review. *Atmospheric Environment* 45, 4329–4343. <https://doi.org/10.1016/j.atmosenv.2011.05.041>

Kim, K.-H., Szulejko, J.E., Kumar, P., Kwon, E.E., Adelodun, A.A., Reddy, P.A.K., 2017. Air ionization as a control technology for off-gas emissions of volatile organic compounds. *Environmental Pollution* 225, 729–743. <https://doi.org/10.1016/j.envpol.2017.03.026>

Liu, W., Huang, J., Lin, Y., Cai, C., Zhao, Y., Teng, Y., Mo, J., Xue, L., Liu, L., Xu, W., Guo, X., Zhang, Y., Zhang, J.J., 2020. Negative Ions Offset Cardiorespiratory Benefits of PM 2.5 Reduction from Residential Use of Negative Ion Air Purifiers. *Indoor Air* ina.12728. <https://doi.org/10.1111/ina.12728>

8. Finally, the main text could use a figure summarizing the modes of transmission and potential ways to mitigate that are discussed in the paper. However, I know that such figures are never trivial to produce, and while nice for readability, is not essential for publication.

9. The Appendix contains a lot of high quality information. The section on UVGI is particularly comprehensive. The section on musical performance/transmission is less developed, and I encourage the authors to either develop this section more clearly, or to consider cutting it to be shorter.

Review form: Referee 3

Is the manuscript an original and important contribution to its field?

Excellent

Is the paper of sufficient general interest?

Acceptable

Is the overall quality of the paper suitable?

Acceptable

Can the paper be shortened without overall detriment to the main message?

Yes

Do you think some of the material would be more appropriate as an electronic appendix?

No

Do you have any ethical concerns with this paper?

No

Recommendation?

Accept with minor revision (please list in comments)

Comments to the Author(s)

See appendix C

Decision letter (RSPA-2020-0855.R0)

07-Dec-2020

Dear Colleagues,

I am writing to inform you that the Editor has made a decision on manuscript entitled "The ventilation of buildings and other mitigating measures for COVID-19: a focus on winter 2020" which you kindly refereed for Proceedings A. Please find the authors' decision letter below.

On behalf of the Editor of Proceedings A, we thank you for your help with this article and we look forward to your input in the future.

Decision made on this manuscript: Accept with minor revision

Best wishes
Raminder Shergill
proceedingsa@royalsociety.org

07-Dec-2020

Dear Dr Burridge,

On behalf of the Editor, I am pleased to inform you that your Manuscript RSPA-2020-0855 entitled "The ventilation of buildings and other mitigating measures for COVID-19: a focus on winter 2020" has been accepted for publication subject to minor revisions in Proceedings A. Please find the referees' comments below.

The reviewer(s) have recommended publication, but also suggest some minor revisions to your manuscript. Therefore, I invite you to respond to the reviewer(s)' comments and revise your manuscript. If possible please submit the revised version of your manuscript within 7 days. If you do not think you will be able to meet this date please let me know in advance of the due date.

To revise your manuscript, log into <https://mc.manuscriptcentral.com/prsa> and enter your Author Centre, where you will find your manuscript title listed under "Manuscripts with Decisions." Under "Actions," click on "Create a Revision." Your manuscript number has been appended to denote a revision.

You will be unable to make your revisions on the originally submitted version of the manuscript. Instead, revise your manuscript and upload a new version through your Author Centre.

When submitting your revised manuscript, you will be able to respond to the comments made by the referee(s) and upload a file "Response to Referees" in Step 1: "View and Respond to Decision Letter". You can use this to document any changes you make to the original manuscript. In order to expedite the processing of the revised manuscript, please be as specific as possible in your response to the referee(s).

IMPORTANT: Your original files are available to you when you upload your revised manuscript. Please delete any redundant files before completing the submission process.

In addition to addressing all of the reviewers' and editor's comments, your revised manuscript **MUST** contain the following sections before the reference list (for any heading that does not apply to your work, please include a comment to this effect):

- Acknowledgements
- Funding statement

See <https://royalsociety.org/journals/authors/author-guidelines/> for further details.

When uploading your revised files, please make sure that you include the following as we cannot proceed without these:

- 1) A text file of the manuscript (doc, txt, rtf or tex), including the references, tables (including captions) and figure captions. Please remove any tracked changes from the text before submission. PDF files are not an accepted format for the "Main Document".
- 2) A separate electronic file of each figure (tif, eps or print-quality pdf preferred). The format should be produced directly from original creation package, or original software format.
- 3) Electronic Supplementary Material (ESM): all supplementary materials accompanying an accepted article will be treated as in their final form. Note that the Royal Society will not edit or typeset supplementary material and it will be hosted as provided. Please ensure that the supplementary material includes the paper details where possible (authors, article title, journal name). Supplementary files will be published alongside the paper on the journal website and posted on the online figshare repository (<https://figshare.com>). The heading and legend provided for each supplementary file during the submission process will be used to create the figshare page, so please ensure these are accurate and informative so that your files can be found in searches. Files on figshare will be made available approximately one week before the accompanying article so that the supplementary material can be attributed a unique DOI.

Alternatively you may upload a zip folder containing all source files for your manuscript as described above with a PDF as your "Main Document". This should be the full paper as it appears when compiled from the individual files supplied in the zip folder.

Article Funder

Please ensure you fill in the Article Funder question on page 2 to ensure the correct data is collected for FundRef (<http://www.crossref.org/fundref/>).

Media summary

Please ensure you include a short non-technical summary (up to 100 words) of the key findings/importance of your paper. This will be used for to promote your work and marketing purposes (e.g. press releases). The summary should be prepared using the following guidelines:

- *Write simple English: this is intended for the general public. Please explain any essential technical terms in a short and simple manner.
- *Describe (a) the study (b) its key findings and (c) its implications.
- *State why this work is newsworthy, be concise and do not overstate (true 'breakthroughs' are a rarity).
- *Ensure that you include valid contact details for the lead author (institutional address, email address, telephone number).

Cover images

We welcome submissions of images for possible use on the cover of Proceedings A. Images should be square in dimension and please ensure that you obtain all relevant copyright permissions before submitting the image to us. If you would like to submit an image for consideration please send your image to proceedingsa@royalsociety.org

Once again, thank you for submitting your manuscript to Proceedings A and I look forward to receiving your revision. If you have any questions at all, please do not hesitate to get in touch.

Best wishes
 Raminder Shergill
 proceedingsa@royalsociety.org
 Proceedings A

on behalf of
 Professor Chris Garrett
 Board Member
 Proceedings A

Reviewer(s)' Comments to Author:

Referee: 1

Comments to the Author(s)

An excellent overview - just missing some important additional references - and a suggestion:

- can the authors create a practical, one-page guide to what, say, a class teacher can do immediately or over a longer term - to enhance ventilation in a school - or a team leader for their office workers, e.g.

Option 1: Open the windows if available (quick, free, minimal disruption)

Option 2: If windows cannot open, work outside (quick, free, some disruption)

Option 3: If it is too cold to work outside and windows cannot open, increase the HVAC mechanical ventilation (quick, \$, minimal disruption - if possible, else see below)

Option 4: Consider periodic venting of the classroom/office if the windows can open but it is very cold outside (quick, free, some disruption)

Option 5: install CO2 monitors to optimise the timing of venting for a given occupancy in an office or classroom environment (needs installation, \$\$, some disruption when CO2 threshold exceeded indicating venting required)

Option 6: use portable HEPA-filtered air purifiers (needs ordering, \$\$, minimal disruption when in operation - some noise disturbance, drafts)

Option 7: install UVGI (needs installation, \$\$\$, minimal disruption after installation)

Option 8: upgrade HVAC system (needs serious infrastructure work, \$\$\$\$, severe disruption during installation, minimal afterwards)

The authors can think of additional features. Think of it as a one-page hand-out to various firms, companies, institutions - if you are offering your services to do this - as this may well be the case - as this virus will be with us for a while to come.

- there are few references cited for the mask section - some additional studies suggested below - to both contain exhaled aerosols as well as protecting against incoming aerosols:

<https://journals.plos.org/plospathogens/article?id=10.1371/journal.ppat.1003205>

Health and Safety Executive UK. RR619 Evaluating the protection afforded by surgical masks against influenza bioaerosols. 2008. <https://www.hse.gov.uk/research/htm/rr619.htm> (Accessed 4 Nov 2020)

Makison Booth C, Clayton M, Crook B, Gawn JM. Effectiveness of surgical masks against influenza bioaerosols. *J Hosp Infect* 2013; 84(1): 22-6.

Weber A, Willeke K, Marchioni R, Myojo T, McKay R, Donnelly J, et al. Aerosol penetration and leakage characteristics of masks used in the health care industry. *Am J Infect Control* 1993; 21(4): 167-73.

van der Sande M, Teunis P, Sabel R. Professional and home-made face masks reduce exposure to respiratory infections among the general population. *PLoS One* 2008; 3(7): e2618.

Davies A, Thompson KA, Giri K, Kafatos G, Walker J, Bennett A. Testing the efficacy of homemade masks: would they protect in an influenza pandemic? *Disaster Med Public Health Prep* 2013; 7(4): 413-8.

Also, can the authors insert a section on the 'short-range aerosol' transmission route. So this is like short-range droplets - but the transmission is mainly via aerosols produced during talking, breathing (rather than coughing, sneezing - which we don't do very often and we normally cover our mouths/noses when we do), i.e. the garlic breath phenomenon over about 1 m typical conversational distance, as we think that this is how SARS-CoV-2 is mainly transmitted now.

These additional reviews may help:

Recognition of aerosol transmission of infectious agents: a commentary

Raymond Tellier, Yuguo Li, Benjamin J. Cowling, Julian W. Tang

BMC Infect Dis. 2019; 19: 101. Published online 2019 Jan 31. doi: 10.1186/s12879-019-3707-y

Airborne spread of infectious agents in the indoor environment

Jianjian Wei, Yuguo Li

Am J Infect Control. 2016 Sep 2; 44(9): S102-S108. Published online 2016 Aug 30. doi: 10.1016/j.ajic.2016.06.003

For their section on contact/ fomite transmission - following on from the above - the UK SAGE/NERVTAG has also found that contact/fomite transmission only accounts for up to about 20% of all respiratory virus transmission - likely including SARS-COV-2 also in this mini-review that they did a while ago. Can the authors incorporate this or at least some of the studies within this review:

https://assets.publishing.service.gov.uk/government/uploads/system/uploads/attachment_data/file/897598/S0574_NERVTAG-EMG_paper_-_hand_hygiene_010720_Redacted.pdf

Referee: 2

Comments to the Author(s)

Overall, this manuscript is well-written, with comprehensive background information that is grounded in the literature combined with evidence-based recommendations for practical engineering solutions. This is a timely and useful overview. I recommend only minor changes, easily achieved by small wording changes. The one section I encourage the authors to consider cutting down or completely moving to the Appendix is Section 3.1. It contains a very different level of detail from the rest of the manuscript, and appears to be written for a different audience from the rest of the manuscript.

Specific Comments.

1. In the discussion of indoor transmission (S2.1), the authors allude to the recent research and the focus on thermal comfort and energy efficiency. What isn't clear from their wording is the fact that strides in energy efficiency have generally come at the expense of ventilation (by making buildings less 'leaky'), and thus indoor air quality. Reducing transmission of airborne pathogens requires more ventilation - the exact opposite of what energy efficiency has required. I recommend slightly rewording the text to make this dichotomy clear.

2. Section 2 includes two separate ideas: transmission (airborne versus fomite), and then some recommendations on social distancing. The literature review of these transmission sections is strong, and the authors made the (in my opinion, correct) choice to combine aerosol and droplet transmission in one section. I deeply appreciate the authors' efforts to make the combined importance of these components clear, particularly later in the text when they discuss the fact that plexiglass barriers and the like do not prevent aerosol transmission. However, my concern/comment with this section is that the authors don't provide an estimate of the relative importance of these two modes (airborne versus surface) of transmission. From my reading of the literature, the bulk of case studies (e.g. the cruise ship study and the Skagit choir study) have shown that airborne transmission dominates. This seems like a useful point to make in terms of providing useful recommendations: vast sums of money are spent on surface decontamination and chemicals, without strong evidence that this is a substantial mode of transmission. I encourage the authors to provide some wording on their / the literature's perspective on relative importance of transmission. The importance of airborne transmission is alluded to later in the section, but not considered relative to fomite transmission.

3. In the discussion of surface cleaning, I encourage the authors to note the potential dangers of exposure to certain cleaning materials (namely sodium hypochlorite), which can produce toxic byproducts - potentially dangerous to janitorial staff, particularly if improperly used.

4. In the Surface cleaning section, the authors refer to recommendations of different approaches - rightly noting that both duration of cleaning and chemical concentration are important considerations. The authors refer to the Kampf studies. However, it would be useful to note whether these testing studies are done on true SARS-CoV-2 viruses, or if they are conducted on proxies: there is recent evidence (though unpublished, I suspect - I've heard in seminars, but not seen a peer-reviewed paper) emerging that SARS-CoV-2 may not respond to disinfectants (namely hydrogen peroxide) in the same way as its proxies.

5. In section 2.3.1, the authors discuss the importance of face coverings / masks. There is substantial research showing that different masks (materials / designs) have different efficacies as a function of aerosol size, and it might be useful for the authors to allude to that fact (i.e. some masks are better than others, but any mask is probably better than nothing).

6. Section 3.1 is far more detailed than the previous sections, and while all important information, it is much harder to follow and would be a useful place to shorten to key points and move the remaining information to the manuscript's Supplemental Information for ease of reading. It also seems out of place as social distancing and ventilation are discussed in the previous section, and this part goes into more detail on that. Instead, focusing on the recommendations for Winter 2020 would be useful.

7. In section 4.2 the authors discuss UV and personal filters. I think they must comment on the downside of improper use of UV (potential generation of high ozone levels; damage to eyes/skin if improperly used/installed). In Section 4.4 the authors allude to ionizers. Again, there are potential issues of O₃ generation, as well as other byproducts:

Zhang, Y., Mo, J., Li, Y., Sundell, J., Wargocki, P., Zhang, J., Little, J.C., Corsi, R., Deng, Q., Leung, M.H.K., Fang, L., Chen, W., Li, J., Sun, Y., 2011. Can commonly-used fan-driven air cleaning technologies improve indoor air quality? A literature review. *Atmospheric Environment* 45, 4329-4343. <https://doi.org/10.1016/j.atmosenv.2011.05.041>

Kim, K.-H., Szulejko, J.E., Kumar, P., Kwon, E.E., Adelodun, A.A., Reddy, P.A.K., 2017. Air ionization as a control technology for off-gas emissions of volatile organic compounds. *Environmental Pollution* 225, 729-743. <https://doi.org/10.1016/j.envpol.2017.03.026>

Liu, W., Huang, J., Lin, Y., Cai, C., Zhao, Y., Teng, Y., Mo, J., Xue, L., Liu, L., Xu, W., Guo, X., Zhang, Y., Zhang, J.J., 2020. Negative Ions Offset Cardiorespiratory Benefits of PM 2.5 Reduction from Residential Use of Negative Ion Air Purifiers. *Indoor Air* ina.12728. <https://doi.org/10.1111/ina.12728>

8. Finally, the main text could use a figure summarizing the modes of transmission and potential ways to mitigate that are discussed in the paper. However, I know that such figures are never trivial to produce, and while nice for readability, is not essential for publication.

9. The Appendix contains a lot of high quality information. The section on UVGI is particularly comprehensive. The section on musical performance/transmission is less developed, and I encourage the authors to either develop this section more clearly, or to consider cutting it to be shorter.

Referee: 3

Comments to the Author(s)

See attached file

Board Member

Comments to Author(s):

Firstly, thank you to the authors for submitting their report to PRSA and for their public service in working on the project. The three reviews are generally positive, recommending minor revision. A fourth review has been promised but is overdue. It will be forwarded if it is received soon.

Based on my own quick reading, particularly of the sections on ventilation, I feel that significant improvements can be made but these should not be too time-consuming. Specifically:

While it may be acceptable for authors of a committee report to be responsible only for their own sections, all authors share in the responsibility for a paper, in a peer-reviewed journal, that includes them in the list of authors. I thus strongly recommend that all authors of the present paper read the whole manuscript carefully and send their recommendations for changes to the lead author.

I suspect that significant updating of the paper will be possible, and desirable, based on research published recently.

As an example of this, Peter Rhines has told me about a paper just published in JFM entitled 'Effects of ventilation on the indoor spread of COVID-19', by Rajesh Bhagat et al.

Coming now to more on ventilation, I was intrigued to learn about the surprisingly high ventilation rate required to keep indoor CO₂ levels below, say, 1,000 ppm. The derivation of this is simple enough but would be worth including as a footnote, or even in the text.

In Section 3.1, results on the quanta required for infection are quoted from the literature rather uncritically. How robust do the authors think that the values are?

The paragraphs around Table 1 make for heavy reading. Are they consistent with the contents of the table? Can they all be summarised in a single formula depending on things like volume of the occupied space, number of occupants, duration of occupation, number of infectious individuals, as well as q and Q ? Some streamlining of this section would be nice.

Related to this, in Table 1, R seems to depend linearly on q for small q , but not for large q . Why? And why not a simple linear dependence on Q ? Maybe explained in the cited paper, but a summary here would be useful.

I found Section A of the Supplementary Material to be rather weak. Several comments are really rather trivially obvious and could perhaps be omitted from a scientific review. The general recommendations are qualitative at best. Surely the analyses in the Bhagat et al. paper mentioned above can provide some guidance, in the absence of CO₂ sensors, on things like how far to open windows in typical situations?

Even qualitatively, I question some of the content. For example, it is assumed that opening a high vent in winter will lead to the inflow of fresh air. I suspect that, unless the vent is above a certain size, opening it will just lead to outflow of warm, buoyant, air that originally entered the room via infiltration or under the door from adjacent interior building space.

I appreciate the trade-off between ventilation and comfort, particularly if 'displacement ventilation' is being used with inflow at a low level. How about a compromise, with inflow at some intermediate height allowed to warm up somewhat by mixing with room air as it descends to the floor before spreading and slowly rising and being exhausted through a high vent? Some

deflection of the inflow might be required, but easy enough to achieve. Whatever the situation, quantitative guidance on advisable opening sizes would be nice.

I apologise if some of the above comments just indicate my insufficiently diligent reading of the report. I suspect that the referees also struggled with the length of the report, which makes it all the more important that all authors accept responsibility for the final product.

Author's Response to Decision Letter for (RSPA-2020-0855.R0)

See Appendix A.

RSPA-2020-0855.R1 (Revision)

Review form: Referee 1

Is the manuscript an original and important contribution to its field?

Excellent

Is the paper of sufficient general interest?

Good

Is the overall quality of the paper suitable?

Excellent

Can the paper be shortened without overall detriment to the main message?

Yes

Do you think some of the material would be more appropriate as an electronic appendix?

No

Do you have any ethical concerns with this paper?

No

Recommendation?

Accept as is

Comments to the Author(s)

The authors have responded comprehensively to the various reviewers' comments.No further comments.

Review form: Referee 2

Is the manuscript an original and important contribution to its field?

Acceptable

Is the paper of sufficient general interest?

Good

Is the overall quality of the paper suitable?

Acceptable

Can the paper be shortened without overall detriment to the main message?

Yes

Do you think some of the material would be more appropriate as an electronic appendix?

No

Do you have any ethical concerns with this paper?

No

Recommendation?

Accept as is

Comments to the Author(s)

As I noted in my review, the paper is timely, well-written and reasonably comprehensive. I appreciate that the bulk of my comments were addressed in the revisions. I will note that two of the three reviewers recommended addressing the relative importance of airborne- versus fomite-driven transmission. While I certainly won't hold up publication on this, I do encourage the authors consider adding some commentary on the current state of the knowledge on transmission.

Review form: Referee 3

Is the manuscript an original and important contribution to its field?

Excellent

Is the paper of sufficient general interest?

Good

Is the overall quality of the paper suitable?

Acceptable

Can the paper be shortened without overall detriment to the main message?

Yes

Do you think some of the material would be more appropriate as an electronic appendix?

No

Do you have any ethical concerns with this paper?

No

Recommendation?

Accept with minor revision (please list in comments)

Comments to the Author(s)

The Ventilation of Buildings and Other Mitigating Measures for COVID-19: a focus on wintertime: 2d Review

First, while largely supporting publication of the paper in my first review I (as well as other referees) advocated for adding a presentation of immediate measures, aimed at public as well as research communities, to reduce airborne transmission of COVID-19 disease. Despite occasional

changes in the text, the resubmitted version continues to be largely devoid of specific advice that can be implemented quickly (none of my recommendations were addressed specifically).

And second, the vagueness of advice to the public reflects a lack of focused, quantitative experiments in the physical science of viral transmission. One year into the pandemic we are still seeing largely qualitative science bearing on ventilation, filtration, transport and mixing of viral material, whether by contact or in airborne pathways. There are more detailed studies of ventilation, masking, air filtration in the published literature (rapidly increasing in late 2020) but few of these are well exploited in the present manuscript.

This 2d review repeats some material from the initial review, but with new insights.

1. Assessment. Generally speaking, during the pandemic advice about air quality indoors and recommended social behavior coming from public health experts is vague, qualitative, lacking detailed observations. It is very similar to that provided during the flu pandemic 100 years ago. Assessment of indoor air quality requires solid quantitative observation and experiment. Despite valuable illustration of respired air patterns in a variety of circumstances the present manuscript is largely lacking quantitative observations. Discussion here, and more widely of ventilation schemes for indoor spaces, lacks sufficient detail to implement cold-season passive ventilation, filtration and heating for comfort. The complexity of indoor airflow and mixing makes it difficult to argue for or against mixing-ventilation (because threads of highly concentrated viral aerosol are spread by mixing, and may or may not be diluted). Airborne transmission of disease by viral aerosol hinges on the relationship between viral load and severity of infection¹, which remains uncertain.

Assessment of net ventilation rate for moderate sized dwellings, offices, care homes, restaurants, prisons, school classrooms needs guidance. The manuscript spends nearly a full page outlining difficulties in measuring net ventilation of an interior space (Sec. 3.1) yet in fact the net ventilation can readily be measured with sensors for passive tracers carried by the air:

(i) Ventilation time-series using CO₂ sourced with dry ice. With one or a few CO₂ sensors, time-series of CO₂ concentration and its variance can be recorded following injection of an initial 5000 ppm concentration (for example) sublimated from dry ice. The CO₂ is readily mixed to near uniformity with a fan, and subsequently is measured as it declines. This is readily done for a variety of active and passive ventilation schemes, and can in be quick and quantitatively accurate. It complements point-source release of CO₂ illustrated for occupied hallways in Sec. 2.5. Both forms of CO₂ release experiment can be used to assess air change ventilation in all manner of occupied spaces. Using 4 CO₂ sensors we have recorded time-series for both mean concentration and variability (notably vertical structure) through the ventilation period with a single elevated exhaust fan. This assessment of room air-change time-scale (or its inverse, air changes per hour) can then contribute when sufficient real data for the contact-traced infection rate in occupied rooms becomes available.

CO₂ of course is a swiss-army-knife for ventilation, recording respired breath (absent gas cookers and outdoor sources); being itself a health hazard in poorly ventilated indoor spaces; and CO₂ is a sensitive tracer of net ventilation, which when diluted moves accurately with the air.

Adverse health impacts of CO₂ laden air have themselves had extensive experimentation², going far beyond the manuscript's citation (Secs. 1 and 3.2) of the 1000 ppm CO₂ threshold for safe air (which turns out to be inaccurate if cognitive ability is the metric). Sec. 3.2 could well point to the need for opening bedroom windows at night to avoid really dangerous expired CO₂ levels (and potentially, infections).

(ii) Ventilation time series using fine particle aerosol measurement with laser particle scattering (e.g., Purple Air, the widely deployed US network of dual laser particle sensors). Particles are seeded by a brief episode of stove-top cooking, which populates an interior space with a frightening concentration of 0.1 - 10 micron particles. These sensors are more accurate and less expensive than typical CO₂ sensors. PM_{2.5} aerosol measurements also provide independent

data on infiltration of outdoor air into a building, and might be used along with CO₂ measurements to characterize both respired air volume and its (possibly infective) aerosol burden.

The manuscript describes in detail models of viral transmission (Sec. 3.1), with observed CO₂ concentration providing evidence of respired air volume. However, 'superspreaders' account for a significant, perhaps dominant fraction of infections. These require contact tracing that reach both both forward in time (with reproductive number R) and backward in time to locate viral sources (made vivid by the 'friends paradox' and the book *The Rules of Contagion* by Kucharski). This suggests backing off from the manuscript's long discussion and quantitative recommendations based on modeling of both airborne transmission rates and epidemiology of infections (Sec. 3.1 and Burridge et al. 2020), about which they say "...superspreading events - we choose not to dwell on such cases but note that should we have done so then the risks reported herein would be dramatically increased." ... because the viral load from a superspreader is far from uniform among inhabitants of the space. Genomic reconstruction of the huge network of infections seeded by a February 26-27 business conference in Boston³ shows superspreading in action, and underlines the need for similarly sophisticated analysis of physics of airborne transmission. Yet CO₂ is nevertheless a promising indicator of expired, potentially infective, air, and has provided particularly vivid pre-pandemic studies of indoor air quality. Reproducing a figure based on the school-room study of Perez et al. (Sensors, 2018) would make that point.

2. Remediation. Both referee 1 and I urged the inclusion of a brief account of procedures the public could embrace for avoiding COVID-19 infection, possibly to be followed by more expansive publication and website activity. The authors responded that relevant publications by their group are available separately. Yet the two references (CIBSE Guide B: Ventilation and Ductwork, 2020 and MusicMark Ventilation of Teaching Spaces, 2020) seem inadequate for widespread advice. I may well be missing some better outreach publications, but the current surge in infections makes it obvious that the general population is unable to protect itself despite measures available to do so. Governmental agencies like WHO and CDC in the US have many relevant advisory documents on the web, yet they often lack detail and lag far behind the science.

Most of the world's people are at risk of Covid-19 illness. Simple measures to isolate them from the virus are available and should be a part of any comprehensive publication of the science of the pandemic. These are not normal times, where presentation of underlying science and applied practice are separated and deferred. Of course Proc.Roy.Soc.A papers are not read by the general public but investigative journalists would respond to sharply focused strategic advice. The manuscript already provides timely discussion of masking, fomites and disinfectants, droplet dynamics, UV treatment, qualitative patterns of aerosol spreading, quantitative model study of infection inferred from CO₂ concentration of respired air.

One example to recommend regards face masking: coverage in the text is long and comprehensive, and I see your response to reviewer 2 on this issue. But instead of a full page simply referencing many advocates of masking why not include a figure⁴ showing effectiveness of various masking materials, which can protect the wearer as well as providing 'source control' to protect others. This could amplify your message and inspire mask construction at home or commercially, with great effect. It might also reduce the selfish resistance to wearing masks found widely among those unconcerned with the public good. More detailed data on pathways (nose, mouth, larynx, lungs) for aerosol inhalation with and without masks is given by Xi et al. (Phys. Fluids October 2020).

Exhaust fans are another example of great importance. The key goal is steady upward displacement ventilation, carrying virus away from face level, using active, elevated exhaust at a high window. The manuscript strangely describes only passive ventilation. Forced displacement ventilation is described for hospitals by Bhagat & Linden (Roy. Soc. Open Science, 2020) but is not mentioned in the present paper. Although windows do not normally extend up to the ceiling, it is not difficult to cut a small opening in an exterior wall. Failing that, fans combined with ducts

can simulate an elevated exhaust port (accepting the head loss in curved ducts). Relatively inexpensive exhaust fans are capable of a wide range of flow rates, from ~ 100 cfm to several thousand cfm (cubic feet per minute), readily enabling rapid room air changes per hour. Attention to outdoor air input at floor level also is needed, and ducted fans may assist there as well.

Extensive discussion is given in the manuscript to support opening of windows to improve passive ventilation, with qualitative suggestions on how this should be done (Secs. 3.2, A.1, A.2). When an elevated window is opened during the cold season, does the air flow inward or outward? Of course it can do either or both, depending on interior thermal stratification, outside temperature, winds. Indeed, Baghat et al. 2020 give fine discussion of these issues. As the manuscript well describes (Sec. 2.4), rapid purging of air during frequent breaks can improve school classroom air quality, but this is most rapidly done with active exhaust fans.

Portable air purifiers are another example of remediation (Sec. 4.1). As well as highly engineered commercial units, inexpensive home-built purifiers combining a HEPA filter and box window-fan have been widely tested, and can effectively remove fine aerosol particles bearing virus. Typical airflow rates range from 100 to 400 cfm. This provides a strategy for the cold season where installed HVAC heating cannot be retrofitted with HEPA or MERV-16 level filters. The current text acknowledges portable air cleaners yet with ambivalent recommendation.

Outdoor classrooms, established a century ago during the Spanish flu pandemic, are able to keep children and teachers safe. Even in cold climates this seemingly extreme strategy is working, though not widespread. Dressed warmly, classes in partially enclosed large tents are currently open in our town, where the public schools are locked down, following a brief period of in-person schooling given up.

The pandemic may be with us for some time. Cold-season ventilation using heat-recovery ventilation (HRV) could reduce the conflict between keeping warm and keeping ventilated indoors. Capture efficiencies of 70% to 80% of the temperature difference between warm outgoing and cold incoming airstreams are possible. While commercial units for entire buildings are costly, heat-exchanger ventilation units can be constructed for smaller spaces without great cost.

1 Liu, Yan, Wan, Xiang, Le, Liu, Peiris, Poon and Zhang: Viral dynamics in mild and severe cases of COVID-19, *Lancet Infectious Disease*, vol. 20, 656-657, 2020 : "The mean viral load of severe cases was around 60 times higher than that of mild cases, suggesting that higher viral loads might be associated with severe clinical outcomes."

2 Allen, JG: Association of cognitive function scores with carbon dioxide, ventilation, and volatile organic compound exposures in office workers: a controlled exposure of green and conventional office environments. *Environmental Health Perspectives*, 124, 2016

3 Lemieux et al. Phylogenetic analysis of SARS-CoV-2 in Boston highlights the impact of superspreading events. *Science* 20 Dec. 2020.

4 Figure 4 in Pan, Harb, Leng & Marr: Inward and outward effectiveness of cloth masks, a surgical mask and a face shield. *MedRxiv* 2020. doi:<https://doi.org/10.1101/2020.11.18.20233353>

Decision letter (RSPA-2020-0855.R1)

15-Jan-2021

Dear Dr Burridge,

On behalf of the Editor, I am pleased to inform you that your Manuscript RSPA-2020-0855.R1 entitled "The ventilation of buildings and other mitigating measures for COVID-19: a focus on wintertime" has been accepted for publication subject to minor revisions in Proceedings A. Please find the referees' comments below.

The reviewer(s) have recommended publication, but also suggest some minor revisions to your manuscript. Therefore, I invite you to respond to the reviewer(s)' comments and revise your manuscript. Please note that we have a strict upper limit of 28 pages for each paper. Please endeavour to incorporate any revisions while keeping the paper within journal limits.

It is a condition of publication that you submit the revised version of your manuscript within 7 days. If you do not think you will be able to meet this date please let me know in advance of the due date.

To revise your manuscript, log into <https://mc.manuscriptcentral.com/prsa> and enter your Author Centre, where you will find your manuscript title listed under "Manuscripts with Decisions." Under "Actions," click on "Create a Revision." Your manuscript number has been appended to denote a revision.

You will be unable to make your revisions on the originally submitted version of the manuscript. Instead, revise your manuscript and upload a new version through your Author Centre.

When submitting your revised manuscript, you will be able to respond to the comments made by the referee(s) and upload a file "Response to Referees" in Step 1: "View and Respond to Decision Letter". You can use this to document any changes you make to the original manuscript. In order to expedite the processing of the revised manuscript, please be as specific as possible in your response to the referee(s).

IMPORTANT: Your original files are available to you when you upload your revised manuscript. Please delete any redundant files before completing the submission process.

When uploading your revised files, please make sure that you include the following as we cannot proceed without these:

- 1) A text file of the manuscript (doc, txt, rtf or tex), including the references, tables (including captions) and figure captions. Please remove any tracked changes from the text before submission. PDF files are not an accepted format for the "Main Document".
- 2) A separate electronic file of each figure (tif, eps or print-quality pdf preferred). The format should be produced directly from original creation package, or original software format.
- 3) Electronic Supplementary Material (ESM): all supplementary materials accompanying an accepted article will be treated as in their final form. Note that the Royal Society will not edit or typeset supplementary material and it will be hosted as provided. Please ensure that the supplementary material includes the paper details where possible (authors, article title, journal name). Supplementary files will be published alongside the paper on the journal website and posted on the online figshare repository (<https://figshare.com>). The heading and legend provided for each supplementary file during the submission process will be used to create the figshare page, so please ensure these are accurate and informative so that your files can be found

in searches. Files on figshare will be made available approximately one week before the accompanying article so that the supplementary material can be attributed a unique DOI.

Alternatively you may upload a zip folder containing all source files for your manuscript as described above with a PDF as your "Main Document". This should be the full paper as it appears when compiled from the individual files supplied in the zip folder.

Article Funder

Please ensure you fill in the Article Funder question on page 2 to ensure the correct data is collected for FundRef (<http://www.crossref.org/fundref/>).

Media summary

Please ensure you include a short non-technical summary (up to 100 words) of the key findings/importance of your paper. This will be used for to promote your work and marketing purposes (e.g. press releases). The summary should be prepared using the following guidelines:

- *Write simple English: this is intended for the general public. Please explain any essential technical terms in a short and simple manner.
- *Describe (a) the study (b) its key findings and (c) its implications.
- *State why this work is newsworthy, be concise and do not overstate (true 'breakthroughs' are a rarity).
- *Ensure that you include valid contact details for the lead author (institutional address, email address, telephone number).

Cover images

We welcome submissions of images for possible use on the cover of Proceedings A. Images should be square in dimension and please ensure that you obtain all relevant copyright permissions before submitting the image to us. If you would like to submit an image for consideration please send your image to proceedingsa@royalsociety.org

Once again, thank you for submitting your manuscript to Proceedings A and I look forward to receiving your revision. If you have any questions at all, please do not hesitate to get in touch.

Best wishes
 Raminder Shergill
proceedingsa@royalsociety.org
 Proceedings A

on behalf of
 Professor Chris Garrett
 Board Member
 Proceedings A

Reviewer(s)' Comments to Author:

Referee: 2

Comments to the Author(s)

As I noted in my review, the paper is timely, well-written and reasonably comprehensive. I appreciate that the bulk of my comments were addressed in the revisions. I will note that two of the three reviewers recommended addressing the relative importance of airborne- versus fomite-driven transmission. While I certainly won't hold up publication on this, I do encourage the authors consider adding some commentary on the current state of the knowledge on transmission.

Referee: 3

Comments to the Author(s)

The Ventilation of Buildings and Other Mitigating Measures for COVID-19: a focus on wintertime: 2d Review

First, while largely supporting publication of the paper in my first review I (as well as other referees) advocated for adding a presentation of immediate measures, aimed at public as well as research communities, to reduce airborne transmission of COVID-19 disease. Despite occasional changes in the text, the resubmitted version continues to be largely devoid of specific advice that can be implemented quickly (none of my recommendations were addressed specifically).

And second, the vagueness of advice to the public reflects a lack of focused, quantitative experiments in the physical science of viral transmission. One year into the pandemic we are still seeing largely qualitative science bearing on ventilation, filtration, transport and mixing of viral material, whether by contact or in airborne pathways. There are more detailed studies of ventilation, masking, air filtration in the published literature (rapidly increasing in late 2020) but few of these are well exploited in the present manuscript.

This 2d review repeats some material from the initial review, but with new insights.

1. Assessment. Generally speaking, during the pandemic advice about air quality indoors and recommended social behavior coming from public health experts is vague, qualitative, lacking detailed observations. It is very similar to that provided during the flu pandemic 100 years ago. Assessment of indoor air quality requires solid quantitative observation and experiment. Despite valuable illustration of respired air patterns in a variety of circumstances the present manuscript is largely lacking quantitative observations. Discussion here, and more widely of ventilation schemes for indoor spaces, lacks sufficient detail to implement cold-season passive ventilation, filtration and heating for comfort. The complexity of indoor airflow and mixing makes it difficult to argue for or against mixing-ventilation (because threads of highly concentrated viral aerosol are spread by mixing, and may or may not be diluted). Airborne transmission of disease by viral aerosol hinges on the relationship between viral load and severity of infection¹, which remains uncertain.

Assessment of net ventilation rate for moderate sized dwellings, offices, care homes, restaurants, prisons, school classrooms needs guidance. The manuscript spends nearly a full page outlining difficulties in measuring net ventilation of an interior space (Sec. 3.1) yet in fact the net ventilation can readily be measured with sensors for passive tracers carried by the air:

(i) Ventilation time-series using CO₂ sourced with dry ice. With one or a few CO₂ sensors, time-series of CO₂ concentration and its variance can be recorded following injection of an initial 5000 ppm concentration (for example) sublimated from dry ice. The CO₂ is readily mixed to near uniformity with a fan, and subsequently is measured as it declines. This is readily done for a variety of active and passive ventilation schemes, and can in be quick and quantitatively accurate. It complements point-source release of CO₂ illustrated for occupied hallways in Sec. 2.5. Both forms of CO₂ release experiment can be used to assess air change ventilation in all manner of occupied spaces. Using 4 CO₂ sensors we have recorded time-series for both mean concentration and variability (notably vertical structure) through the ventilation period with a single elevated exhaust fan. This assessment of room air-change time-scale (or its inverse, air changes per hour) can then contribute when sufficient real data for the contact-traced infection rate in occupied rooms becomes available.

CO₂ of course is a swiss-army-knife for ventilation, recording respired breath (absent gas cookers and outdoor sources); being itself a health hazard in poorly ventilated indoor spaces; and CO₂ is a sensitive tracer of net ventilation, which when diluted moves accurately with the air.

Adverse health impacts of CO₂ laden air have themselves had extensive experimentation², going far beyond the manuscript's citation (Secs. 1 and 3.2) of the 1000 ppm CO₂ threshold for

safe air (which turns out to be inaccurate if cognitive ability is the metric). Sec. 3.2 could well point to the need for opening bedroom windows at night to avoid really dangerous expired CO₂ levels (and potentially, infections).

(ii) Ventilation time series using fine particle aerosol measurement with laser particle scattering (e.g., Purple Air, the widely deployed US network of dual laser particle sensors). Particles are seeded by a brief episode of stove-top cooking, which populates an interior space with a frightening concentration of 0.1 - 10 micron particles. These sensors are more accurate and less expensive than typical CO₂ sensors. PM_{2.5} aerosol measurements also provide independent data on infiltration of outdoor air into a building, and might be used along with CO₂ measurements to characterize both respired air volume and its (possibly infective) aerosol burden.

The manuscript describes in detail models of viral transmission (Sec. 3.1), with observed CO₂ concentration providing evidence of respired air volume. However, 'superspreaders' account for a significant, perhaps dominant fraction of infections. These require contact tracing that reach both both forward in time (with reproductive number R) and backward in time to locate viral sources (made vivid by the 'friends paradox' and the book *The Rules of Contagion* by Kucharski). This suggests backing off from the manuscript's long discussion and quantitative recommendations based on modeling of both airborne transmission rates and epidemiology of infections (Sec. 3.1 and Burrige et al. 2020), about which they say "...superspreading events - we choose not to dwell on such cases but note that should we have done so then the risks reported herein would be dramatically increased." ... because the viral load from a superspreader is far from uniform among inhabitants of the space. Genomic reconstruction of the huge network of infections seeded by a February 26-27 business conference in Boston³ shows superspreading in action, and underlines the need for similarly sophisticated analysis of physics of airborne transmission. Yet CO₂ is nevertheless a promising indicator of expired, potentially infective, air, and has provided particularly vivid pre-pandemic studies of indoor air quality. Reproducing a figure based on the school-room study of Perez et al. (Sensors, 2018) would make that point.

2. Remediation. Both referee 1 and I urged the inclusion of a brief account of procedures the public could embrace for avoiding COVID-19 infection, possibly to be followed by more expansive publication and website activity. The authors responded that relevant publications by their group are available separately. Yet the two references (CIBSE Guide B: Ventilation and Ductwork, 2020 and MusicMark Ventilation of Teaching Spaces, 2020) seem inadequate for widespread advice. I may well be missing some better outreach publications, but the current surge in infections makes it obvious that the general population is unable to protect itself despite measures available to do so. Governmental agencies like WHO and CDC in the US have many relevant advisory documents on the web, yet they often lack detail and lag far behind the science.

Most of the world's people are at risk of Covid-19 illness. Simple measures to isolate them from the virus are available and should be a part of any comprehensive publication of the science of the pandemic. These are not normal times, where presentation of underlying science and applied practice are separated and deferred. Of course Proc.Roy.Soc.A papers are not read by the general public but investigative journalists would respond to sharply focused strategic advice. The manuscript already provides timely discussion of masking, fomites and disinfectants, droplet dynamics, UV treatment, qualitative patterns of aerosol spreading, quantitative model study of infection inferred from CO₂ concentration of respired air.

One example to recommend regards face masking: coverage in the text is long and comprehensive, and I see your response to reviewer 2 on this issue. But instead of a full page simply referencing many advocates of masking why not include a figure⁴ showing effectiveness of various masking materials, which can protect the wearer as well as providing 'source control' to protect others. This could amplify your message and inspire mask construction at home or commercially, with great effect. It might also reduce the selfish resistance to wearing masks found widely among those unconcerned with the public good. More detailed data on pathways

(nose, mouth, larynx, lungs) for aerosol inhalation with and without masks is given by Xi et al. (Phys. Fluids October 2020).

Exhaust fans are another example of great importance. The key goal is steady upward displacement ventilation, carrying virus away from face level, using active, elevated exhaust at a high window. The manuscript strangely describes only passive ventilation. Forced displacement ventilation is described for hospitals by Bhagat & Linden (Roy. Soc. Open Science, 2020) but is not mentioned in the present paper. Although windows do not normally extend up to the ceiling, it is not difficult to cut a small opening in an exterior wall. Failing that, fans combined with ducts can simulate an elevated exhaust port (accepting the head loss in curved ducts).

Relatively inexpensive exhaust fans are capable of a wide range of flow rates, from ~ 100 cfm to several thousand cfm (cubic feet per minute), readily enabling rapid room air changes per hour. Attention to outdoor air input at floor level also is needed, and ducted fans may assist there as well.

Extensive discussion is given in the manuscript to support opening of windows to improve passive ventilation, with qualitative suggestions on how this should be done (Secs. 3.2, A.1, A.2). When an elevated window is opened during the cold season, does the air flow inward or outward? Of course it can do either or both, depending on interior thermal stratification, outside temperature, winds. Indeed, Baghat et al. 2020 give fine discussion of these issues. As the manuscript well describes (Sec. 2.4), rapid purging of air during frequent breaks can improve school classroom air quality, but this is most rapidly done with active exhaust fans.

Portable air purifiers are another example of remediation (Sec. 4.1). As well as highly engineered commercial units, inexpensive home-built purifiers combining a HEPA filter and box window-fan have been widely tested, and can effectively remove fine aerosol particles bearing virus. Typical airflow rates range from 100 to 400 cfm. This provides a strategy for the cold season where installed HVAC heating cannot be retrofitted with HEPA or MERV-16 level filters. The current text acknowledges portable air cleaners yet with ambivalent recommendation.

Outdoor classrooms, established a century ago during the Spanish flu pandemic, are able to keep children and teachers safe. Even in cold climates this seemingly extreme strategy is working, though not widespread. Dressed warmly, classes in partially enclosed large tents are currently open in our town, where the public schools are locked down, following a brief period of in-person schooling given up.

The pandemic may be with us for some time. Cold-season ventilation using heat-recovery ventilation (HRV) could reduce the conflict between keeping warm and keeping ventilated indoors. Capture efficiencies of 70% to 80% of the temperature difference between warm outgoing and cold incoming airstreams are possible. While commercial units for entire buildings are costly, heat-exchanger ventilation units can be constructed for smaller spaces without great cost.

1 Liu, Yan, Wan, Xiang, Le, Liu, Peiris, Poon and Zhang: Viral dynamics in mild and severe cases of COVID-19, *Lancet Infectious Disease*, vol. 20, 656-657, 2020 : "The mean viral load of severe cases was around 60 times higher than that of mild cases, suggesting that higher viral loads might be associated with severe clinical outcomes."

2 Allen, JG: Association of cognitive function scores with carbon dioxide, ventilation, and volatile organic compound exposures in office workers: a controlled exposure of green and conventional office environments. *Environmental Health Perspectives*, 124, 2016

3 Lemieux et al. Phylogenetic analysis of SARS-CoV-2 in Boston highlights the impact of superspreading events. *Science* 20 Dec. 2020.

4 Figure 4 in Pan, Harb, Leng & Marr: Inward and outward effectiveness of cloth masks, a surgical mask and a face shield. MedRxiv 2020. doi:<https://doi.org/10.1101/2020.11.18.20233353>

Referee: 1

Comments to the Author(s)

The authors have responded comprehensively to the various reviewers' comments.No further comments.

Board Member

Comments to Author(s):

See attached

Author's Response to Decision Letter for (RSPA-2020-0855.R1)

See Appendix B.

Decision letter (RSPA-2020-0855.R2)

10-Feb-2021

Dear Dr Burridge

I am pleased to inform you that your manuscript entitled "The ventilation of buildings and other mitigating measures for COVID-19: a focus on wintertime" has been accepted for publication in Proceedings A.

With respect to earlier comments on the use of the Wells-Riley formula and lack of consideration of alternative dose-response models, we recommend that you do at least include a reference to the 2010 review paper 'Review and comparison between the Wells-Riley and dose-response approaches to risk assessment of infectious respiratory diseases' by G. N. Sze To and C. Y. H. Chao, *Indoor Air* 20, 2-16. You might also consider commenting on the impact of new variants on your conclusions. These could be added when returning the proof corrections.

Our Production Office will be in contact with you in due course. You can expect to receive a proof of your article soon. Please contact the office to let us know if you are likely to be away from e-mail in the near future. If you do not notify us and comments are not received within 5 days of sending the proof, we may publish the paper as it stands.

COVID-19 rapid publication process: We are taking steps to expedite the publication of research relevant to the pandemic. If you wish, you can opt to have your paper published as soon as it is ready, rather than waiting for it to be published on the scheduled Wednesday.

This means your paper will not be included in the weekly media round-up which the Society sends to journalists ahead of publication. However, it will appear in the COVID-19 Publishing Collection which journalists will be directed to each week

(<https://royalsocietypublishing.org/topic/special-collections/novel-coronavirus-outbreak>)

If you wish to have your paper published immediately please notify

proca_proofs@royalsociety.org and press@royalsociety.org

The Royal Society has signed a Wellcome statement on the subject of research findings and data relevant to the coronavirus (COVID-19) outbreak. We are one of several signatories to this

statement and our collective aim is to ensure that the relevant research and data are shared rapidly and openly in order to inform the worldwide public health response and to help save lives. We are therefore making papers related to COVID-19 open access free of charge.

Under the terms of our licence to publish you may post the author generated postprint (ie. your accepted version not the final typeset version) of your manuscript at any time and this can be made freely available. Postprints can be deposited on a personal or institutional website, or a recognised server/repository. Please note however, that the reporting of postprints is subject to a media embargo, and that the status the manuscript should be made clear. Upon publication of the definitive version on the publisher's site, full details and a link should be added.

You can cite the article in advance of publication using its DOI. The DOI will take the form: 10.1098/rspa.XXXX.YYYY, where XXXX and YYYY are the last 8 digits of your manuscript number (eg. if your manuscript number is RSPA-2017-1234 the DOI would be 10.1098/rspa.2017.1234).

For tips on promoting your accepted paper see our blog post:
<https://royalsociety.org/blog/2020/07/promoting-your-latest-paper-and-tracking-your-results/>

On behalf of the Editor of Proceedings A, we look forward to your continued contributions to the Journal.

Sincerely,
Raminder Shergill
proceedingsa@royalsociety.org

on behalf of
Dr Chris Garrett
Board Member
Proceedings A

Appendix A - Author's response to decision letter for RSPA-2020-0855.R0

Response to reviewers and board members concerning the decision on RSPA-2020-0855 - Proceedings A

Original text in black with response in red.

Reviewer(s)' Comments to Author:

Referee: 1

Comments to the Author(s)

An excellent overview - just missing some important additional references - and a suggestion:
We thank the reviewer for their constructive comments and suggestions, and for the useful references provided.

- can the authors create a practical, one-page guide to what, say, a class teacher can do immediately or over a longer term - to enhance ventilation in a school - or a team leader for their office workers, e.g.

Option 1: Open the windows if available (quick, free, minimal disruption)

Option 2: If windows cannot open , work outside (quick, free, some disruption)

Option 3: If it is too cold to work outside and windows cannot open, increase the HVAC mechanical ventilation (quick, \$, minimal disruption - if possible, else see below)

Option 4: Consider periodic venting of the classroom/office if the windows can open but it is very cold outside (quick, free, some disruption)

Option 5: install CO2 monitors to optimise the timing of venting for a given occupancy in an office or classroom environment (needs installation, \$\$, some disruption when CO2 threshold exceeded indicating venting required)

Option 6: use portable HEPA-filtered air purifiers (needs ordering, \$\$, minimal disruption when in operation - some noise disturbance, drafts)

Option 7: install UVGI (needs installation, \$\$\$, minimal disruption after installation)

Option 8: upgrade HVAC system (needs serious infrastructure work, \$\$\$\$, severe disruption during installation, minimal afterwards)

The authors can think of additional features. Think of it as a one-page hand-out to various firms, companies, institutions - if you are offering your services to do this - as this may well be the case - as this virus will be with us for a while to come.

This is an excellent suggestion as to some much-needed advice. However, we are not convinced that our article is the appropriate place for publication of such advice. Careful consideration needs to be given as to which circumstances to cover in order to ensure appropriate advice and details are provided. For example advice is available which is relatively focused, e.g.

<https://www.musicmark.org.uk/wp-content/uploads/Questions-about-Ventilation.pdf>, a number of our authors are very much involved in issuing broader level advice: CIBSE COVID-19 Ventilation Guidance (dbsservices.co.uk), and others still are working with the Department for Education to ensure the right tools are available for schools to ensure compliance with guidance/advice.

- there are few references cited for the mask section - some additional studies suggested below - to both contain exhaled aerosols as well as protecting against incoming aerosols:

See below for our response and details of certain changes/additions as a result of this helpful review.

<https://journals.plos.org/plospathogens/article?id=10.1371/journal.ppat.1003205>

Health and Safety Executive UK. RR619 Evaluating the protection afforded by surgical masks against influenza bioaerosols. 2008. <https://www.hse.gov.uk/research/htm/rr619.htm> (Accessed 4 Nov 2020)

Makison Booth C, Clayton M, Crook B, Gawn JM. Effectiveness of surgical masks against influenza bioaerosols. *J Hosp Infect* 2013; 84(1): 22-6.

Weber A, Willeke K, Marchioni R, Myojo T, McKay R, Donnelly J, et al. Aerosol penetration and leakage characteristics of masks used in the health care industry. *Am J Infect Control* 1993; 21(4): 167-73.

van der Sande M, Teunis P, Sabel R. Professional and home-made face masks reduce exposure to respiratory infections among the general population. *PLoS One* 2008; 3(7): e2618.

Davies A, Thompson KA, Giri K, Kafatos G, Walker J, Bennett A. Testing the efficacy of homemade masks: would they protect in an influenza pandemic? *Disaster Med Public Health Prep* 2013; 7(4): 413-8.

Also, can the authors insert a section on the 'short-range aerosol' transmission route. So this is like short-range droplets - but the transmission is mainly via aerosols produced during talking, breathing (rather than coughing, sneezing - which we don't do very often and we normally cover our mouths/noses when we do), i.e. the garlic breath phenomenon over about 1 m typical conversational distance, as we think that this is how SARS-CoV-2 is mainly transmitted now.

We thank the referee for the question and pointing us towards the perspective and the review paper. In the revised manuscript, we have split the first paragraph of Section 2.1.1 in two so that we have one paragraph explaining the distinction between aerosol and droplet transmission, and a second on the mechanisms of these transmission routes. Furthermore, we have put greater emphasis on the short-range nature of the droplet transmission route.

We have not commented on the relative importance of droplet or aerosol transmission as the evidence suggest that this is still an open question.

These additional reviews may help:

Recognition of aerosol transmission of infectious agents: a commentary

Raymond Tellier, Yuguo Li, Benjamin J. Cowling, Julian W. Tang

BMC Infect Dis. 2019; 19: 101. Published online 2019 Jan 31. doi: 10.1186/s12879-019-3707-y

Airborne spread of infectious agents in the indoor environment

Jianjian Wei, Yuguo Li

Am J Infect Control. 2016 Sep 2; 44(9): S102–S108. Published online 2016 Aug 30. doi: 10.1016/j.ajic.2016.06.003

For their section on contact/ fomite transmission - following on from the above - the UK SAGE/NERVTAG has also found that contact/fomite transmission only accounts for up to about 20% of all respiratory virus transmission - likely including SARS-COV-2 also in this mini-review that they did a while ago. Can the authors incorporate this or at least some of the studies within this review:

We have considered the referee's guidance carefully and we thank them very much for their input.

- Our priority of the guidance within the article was in the mitigation of emission of aerosols / droplets / viruses provided by face coverings and masks. This is (was) the case especially given that it is only high-quality masks (FFP2/N95 or greater) that will provide adequate safe protection for the wearer, and if fitted correctly. Also, that such masks are (were) in short supply and should be prioritised to healthcare. However, we have added a reference from the above list (and several others which are more up to date), which reviews masks with respect to emissions and infections (see below).
- The 'few' references that were given were to represent a snapshot of face mask / coverings articles at the time of writing. The intention was to ensure they were all relevant and up to date with respect to COVID. However, the author acknowledges there are excellent older articles (e.g., van der Sande) and more recent articles, which have been published that could help explain the narrative. These few of these have been added (which are relevant and more up to date) – see below.

We have added to the paper the following text

The physics of particle capture by the materials of face masks is complex, it also recognised these mechanisms will be at play in face coverings. Beyond those normally associated with filtration, they reduce the forward momentum of the exhaled fluid, effectively limiting the spread of air and aerosol from the wearer. Face coverings and masks keep the fluid close to the body plume lowering the chances of direct exposure of potential contagion-laden droplets, and by preventing the release of the non-gaseous constituents of the expiratory fluids into the environment (Wei & Li, 2016; Bhagat et al., 2020; K'ahler& Hain, 2020).

The study of efficacy of face coverings and masks in regulating emissions during expiratory activities was conducted by Asadi et al. (2020) they noted "both surgical masks and unvented KN95 respirators, even without fit-testing, reduce the outward particle emission rates by 90% and 74% on average during speaking and coughing, respectively, compared to wearing no mask, corroborating their effectiveness at reducing outward emission." Their results imply cloth face covering and other masks would "reduce emission of virus-laden aerosols and droplets associated with expiratory activities." Including many of the particles emitted in the aerosol range (<5µm), and inertial impaction will be more prevalent as particle sizes increased and is likely to lead to fewer droplet size particles being emitted (>5µm).

A similar experiment also looked at the efficacy of masks, coverings and face shields and their ability to reduce transmission against simulated cough-generated, small aerosol particles from 0 to 7µm. Lindsley et al. (2020) results show N95 respirators are 99% effective at blocking transmission, surgical masks 59%, 3-ply cotton cloth face masks 51% and a polyester neck gaiter 47% and folded double 60%, whereas a face shield blocked only 2% of the cough aerosol. The masks, coverings and face shield became progressively more efficient at blocking the cough aerosol, as the aerosol particle size fraction increased from <0.6µm to 7µm. They stated "These results suggest that cloth face coverings would be effective as source control devices against the large respiratory aerosols that are thought to play an important role in SARS-CoV-2 transmission"

Face masks and coverings may also protect the wearer, to different degrees of effectiveness. To quote van der Sande et al. (2008) "Any type of general mask use is likely to decrease viral exposure and infection risk on a population level, in spite of imperfect fit and imperfect adherence, personal respirators providing most protection." They found that FFP2 (N95) masks protected the wearer significantly better than surgical masks or a simple cloth. Also, similarly outward protection (emissions) was correlated with protection with FFP2 masks, the best of those tested.

Referee: 2

Comments to the Author(s)

Overall, this manuscript is well-written, with comprehensive background information that is grounded in the literature combined with evidence-based recommendations for practical engineering solutions. This is a timely and useful overview. I recommend only minor changes, easily achieved by small wording changes. The one section I encourage the authors to consider cutting down or completely moving to the Appendix is Section 3.1. It contains a very different level of detail from the rest of the manuscript, and appears to be written for a different audience from the rest of the manuscript.

We thank the reviewer for their positive and constructive comments. We respond to each point in turn below.

Specific Comments.

1. In the discussion of indoor transmission (S2.1), the authors allude to the recent research and the focus on thermal comfort and energy efficiency. What isn't clear from their wording is the fact that strides in energy efficiency have generally come at the expense of ventilation (by making buildings less 'leaky'), and thus indoor air quality. Reducing transmission of airborne pathogens requires more ventilation - the exact opposite of what energy efficiency has required. I recommend slightly rewording the text to make this dichotomy clear.

Thanks for the comment, we have now modified this paragraph, which now reads:

Traditionally, building ventilation has been studied in the context of thermal comfort and in the last few decades energy efficiency. The focus on energy efficiency, often imposed by changes in construction standards, in combination with space restrictions arising from higher population density, have led to more tightly constructed, and less spacious buildings. These tightly constructed structures can, without careful design, maintenance and operation, result in inadequate ventilation provision. However, there has been a timely shift of focus and, in addition to energy efficiency and thermal comfort, indoor air quality (and implicitly the removal of any indoor airborne pollutants produced by the occupants) has become a core focus \citep{sloan2020prioritising}.

2. Section 2 includes two separate ideas: transmission (airborne versus fomite), and then some recommendations on social distancing. The literature review of these transmission sections is strong, and the authors made the (in my opinion, correct) choice to combine aerosol and droplet transmission in one section. I deeply appreciate the authors' efforts to make the combined importance of these components clear, particularly later in the text when they discuss the fact that plexiglass barriers and the like do not prevent aerosol transmission. However, my concern/comment with this section is that the authors don't provide an estimate of the relative importance of these two modes (airborne versus surface) of transmission. From my reading of the literature, the bulk of case studies (e.g. the cruise ship study and the Skagit choir study) have shown that airborne transmission dominates. This seems like a useful point to make in terms of providing useful recommendations: vast sums of money are spent on surface decontamination and chemicals, without strong evidence that this is a substantial mode of transmission. I encourage the authors to provide some wording on their / the literature's perspective on relative importance of transmission. The importance of airborne transmission is alluded to later in the section, but not considered relative to fomite transmission.

We appreciate this thoughtful comment and would dearly love to comment on the relative importance of the transmission route from a position of evidenced based knowledge. It is true to say that a number of highly investigated outbreak events are suggestive as to the transmission routes relative importance. However, whilst these events do document cases that give rise to a surprisingly high number of infections – the total number of people infected in these investigated cases is inconsequential compared the total infection globally. Moreover, the environmental conditions and/or behaviours in each of these cases is far from typical. As such, it is far from clear that reporting the relative importance of the transmission routes based on these data is representative for the pandemic as a whole.

3. In the discussion of surface cleaning, I encourage the authors to note the potential dangers of exposure to certain cleaning materials (namely sodium hypochlorite), which can produce toxic byproducts - potentially dangerous to janitorial staff, particularly if improperly used.

We thank the referee for reminding the authors of this point. We have added in an additional sentence advising readers to consult guidance on the appropriate use of cleaning materials, with particular reference to 2017 and 2020 documents by the Health and Safety Executive.

4. In the Surface cleaning section, the authors refer to recommendations of different approaches - rightly noting that both duration of cleaning and chemical concentration are important considerations. The authors refer to the Kampf studies. However, it would be useful to note whether these testing studies are done on true SARS-CoV-2 viruses, or if they are conducted on proxies: there is recent evidence (though unpublished, I suspect - I've heard in seminars, but not seen a peer-reviewed paper) emerging that SARS-CoV-2 may not respond to disinfectants (namely hydrogen peroxide) in the same way as its proxies.

We thank the referee for suggesting some additional clarity on the report. In general, we have gone through the relevant sections (particularly where further details are provided in the Appendix) to specify more clearly where certain references are applicable to general coronavirus or SARS-CoV-2. The referee highlights some uncertainty about the effectiveness of hydrogen peroxide; however, we have not been able to find a reference for this. We have added in an additional paragraph in the Appendix with a citation to a study on the effectiveness of hydrogen peroxide. From Sec. B.5.2:

Note that some of the substances in this list and in \cite{Kampf2020} have not been experimentally proven to be effective against SARS-CoV-2, although they have been proven for other strains of coronaviruses. Recently, \cite{Gerlach2020} has confirmed that alcohols, specifically 70% ethanol and 70% isopropanol, are very effective against Sars-CoV-2 on surfaces such as stainless steel, plastic, glass, cardboard and PVC. In addition to these, \cite{Gerlach2020} confirm that 0.1% hydrogen peroxide and 0.1% sodium laureth sulphate (SLS), generally found in soaps and household cleaning fluids, can completely inactivate the virus on the same surfaces within 60 s. SLS can also successfully inactivate the virus on cotton fabric, suggesting that reusable cotton masks can be disinfected by washing with soap or fluid containing this substance. Note that the rapid inactivation of the virus using SLS shown in this study conflicts with the findings in \cite{Chin2020} where hand soap solution was used to inactivate the virus. The concentration of SLS used in the latter study is unknown. \cite{Gerlach2020} also use a lower concentration of hydrogen peroxide, contrary to the recommendations made by \cite{Kampf2020} on using this substance.

5. In section 2.3.1, the authors discuss the importance of face coverings / masks. There is substantial research showing that different masks (materials / designs) have different efficacies as a function of aerosol size, and it might be useful for the authors to allude to that fact (i.e. some masks are better than others, but any mask is probably better than nothing).

- We thank the reviewer for the comment, it is much appreciated. We consider that although the subject of masks to prevent the wearer from becoming infected is discussed briefly. This section was intended to discuss mask use to prevent transmission. The substantial research showing that different masks (materials / designs) have different efficacies as a function of aerosol size is more relevant to prevention of infection, for example, FFP2 and n95 masks. However, there has been substantial research on varied materials for masks since the original section was written, we will review what is written and allude to any new data available. However, we have now included both van de Sande (2008) any mask better than no mask & Asadi (2020) & Lindsley et al. (2020), these cover different material, masks and the different sized particles captured and some mechanisms – and are included among other articles in the revision of this section.

We have added (and this is subject to final review) -

The physics of particle capture by the materials of face masks is complex, it also recognised these mechanisms will be at play in face coverings. Beyond those normally associated with filtration, they reduce the forward momentum of the exhaled fluid, effectively limiting the spread of air and aerosol from the wearer. Face coverings and masks keep the fluid close to the body plume lowering the chances of direct exposure of potential contagion-laden droplets, and by preventing the release of the non-gaseous constituents of the expiratory fluids into the environment (Wei & Li, 2016; Bhagat et al., 2020; K'ahler & Hain, 2020).

The study of efficacy of face coverings and masks in regulating emissions during expiratory activities was conducted by Asadi et al. (2020) they noted “both surgical masks and unvented KN95 respirators, even without fit-testing, reduce the outward particle emission rates by 90% and 74% on average during speaking and coughing, respectively, compared to wearing no mask, corroborating their effectiveness at reducing outward emission.” Their results imply cloth face covering and other masks would “reduce emission of virus-laden aerosols and droplets associated with expiratory activities.” Including many of the particles emitted in the aerosol range (<5µm), and inertial impaction will be more prevalent as particle sizes increased and is likely to lead to fewer droplet size particles being emitted (>5µm).

A similar experiment also looked at the efficacy of masks, coverings and face shields and their ability to reduce transmission against simulated cough-generated, small aerosol particles from 0 to 7µm. Lindsley et al. (2020) results show N95 respirators are 99% effective at blocking transmission, surgical masks 59%, 3-ply cotton cloth face masks 51% and a polyester neck gaiter 47% and folded double 60%, whereas a face shield blocked only 2% of the cough aerosol. The masks, coverings and face shield became progressively more efficient at blocking the cough aerosol, as the aerosol particle size fraction increased from <0.6µm to 7µm. They stated “These results suggest that cloth face coverings would be effective as source control devices against the large respiratory aerosols that are thought to play an important role in SARS-CoV-2 transmission”

Face masks and coverings may also protect the wearer, to different degrees of effectiveness. To quote van der Sande et al. (2008) “Any type of general mask use is likely to decrease viral exposure and infection risk on a population level, in spite of imperfect fit and imperfect adherence, personal respirators providing most protection.” They found that FFP2 (N95) masks protected the wearer significantly better than surgical masks or a simple cloth. Also, similarly outward protection (emissions) was correlated with protection with FFP2 masks, the best of those tested.

6. Section 3.1 is far more detailed than the previous sections, and while all important information, it is much harder to follow and would be a useful place to shorten to key points and move the remaining information to the manuscript's Supplemental Information for ease of reading. It also seems out of place as social distancing and ventilation are discussed in the previous section, and this part goes into more detail on that. Instead, focusing on the recommendations for Winter 2020 would be useful.

We have given this point much consideration. As the reviewer points out ventilation is a focus area of the paper and so it is natural that this section contains a greater level of detail than some others. Moreover, ventilation is not a silver bullet and so it is necessary to tie in comments to other measures, e.g. social distancing, from here and elsewhere. We were not keen to put more details on ventilation in the Supplemental Information but we have sought to better highlight key points both within section 3.1 and crucially in the executive summary.

7. In section 4.2 the authors discuss UV and personal filters. I think they must comment on the downside of improper use of UV (potential generation of high ozone levels; damage to eyes/skin if improperly used/installed). In Section 4.4 the authors allude to ionizers. Again, there are potential issues of O₃ generation, as well as other byproducts:

Zhang, Y., Mo, J., Li, Y., Sundell, J., Wargocki, P., Zhang, J., Little, J.C., Corsi, R., Deng, Q., Leung, M.H.K., Fang, L., Chen, W., Li, J., Sun, Y., 2011. Can commonly-used fan-driven air cleaning technologies improve indoor air quality? A literature review. *Atmospheric Environment* 45, 4329–4343. <https://doi.org/10.1016/j.atmosenv.2011.05.041>

Kim, K.-H., Szulejko, J.E., Kumar, P., Kwon, E.E., Adelodun, A.A., Reddy, P.A.K., 2017. Air ionization as a control technology for off-gas emissions of volatile organic compounds. *Environmental Pollution* 225, 729–743. <https://doi.org/10.1016/j.envpol.2017.03.026>

Liu, W., Huang, J., Lin, Y., Cai, C., Zhao, Y., Teng, Y., Mo, J., Xue, L., Liu, L., Xu, W., Guo, X., Zhang, Y., Zhang, J.J., 2020. Negative Ions Offset Cardiorespiratory Benefits of PM 2.5 Reduction from Residential Use of Negative Ion Air Purifiers. *Indoor Air* ina.12728. <https://doi.org/10.1111/ina.12728>

The reviewer is right to highlight the potential harmful effects of the improper use of UV light. A sentence has now been added to address this in section 4.1 on the use of air cleaners. While the potential risks of using UV light was mentioned in section 4.6 on upper room UV, this has now been expanded in order to emphasise the potential dangers of improper installation. The possible adverse effect on air quality through the production of ozone by ionizers was mentioned in section 4.4, however an additional comment has been added, making use of the reference provided by the reviewer to a very recent paper which is highly relevant.

8. Finally, the main text could use a figure summarizing the modes of transmission and potential ways to mitigate that are discussed in the paper. However, I know that such figures are never trivial to produce, and while nice for readability, is not essential for publication.

We have given this suggestion great consideration. We feel that both in the popular press and the academic literature, to which we refer, there is a wealth of diagrams illustrating the modes of transmission. Adding to this a thorough and robust illustration of the various mitigation strategies is extremely challenging and one which we struggle to find appropriate resources to address. We hope the reviewer is understanding of our circumstances.

9. The Appendix contains a lot of high quality information. The section on UVGI is particularly comprehensive. The section on musical performance/transmission is less developed, and I encourage the authors to either develop this section more clearly, or to consider cutting it to be shorter.

Response to reviewers concerning Appendix D (musical performance aspects).

The purpose of this appendix was to reference useful and usable information, for the benefit of laity, and also as a starting point for planning more penetrating research. In addition, it is probably important to point out that during this period of uncertainty, feelings in amateur musical groups have been running very high, and it has been difficult for the various support organisations to contain and direct behaviours to ensure safety. For these organisations, having scientific papers to cite, involving scientists in communications exercises, and scientists responding to questions on social media has been an important part of maintaining some control.

I accept that your comments about being “less developed” are quite correct, but I have also been trying to capture the mood of the musicians I have been engaging with, and the appendix includes issues that are important to those musicians. For example, the problems of playing wearing masks and/or bell covers: this mainly affects students, who cannot attend tuition without them, but the efficacy is unknown and the compromise to their playing is significant. This deserves more research. Likewise, the issue of sanitising of shared instruments is important both to schools and to instrument sales. Then there is an issue of specious beliefs that need to be quashed, although this is hard without solid evidence. Examples of this include the assumption that all respiratory droplets are condensed within the instrument and therefore brass instruments are safe – this is not true, and I have proved it to my own satisfaction, but it hardly merits an academic paper. Also in this category is the idea that non-blown instruments are safe – but they are still played by breathing humans, and similar issues apply to them as to travellers moving up and down a train carriage, and all the other concerns about ventilation effectiveness. On balance, I have condensed the description of these issues slightly, but kept them all included.

In the short time since the paper was first submitted there have been more guidance publications from support organisations, so a few more references have been added to point to those. This current paper, in pre-print form, has been circulated and cited by some of those organisations. The scientific studies of aerosol production are still underway: the Colorado group has issued a second update (citation added), but there are no additional journal papers to cite on this. I have added a bit about the mental health and economic benefits of music with a couple of references.

All in all, this work is ongoing, and one would hope that a review paper such as this would prompt other researchers to tackle the knowledge gaps. As an appendix, this is “additional material” and I hope that more and better additional material can be made available as time progresses.

Referee: 3

Comments to the Author(s)

See attached file

Response to referee 3 who identified himself as Peter Rhines:

Dear Peter

Thank you for taking the time and trouble to referee our paper. Given our earlier emails I appreciate and agree with your viewpoint of the importance of Web-based communication. However, as you say there is a lot of unrefereed literature on this subject and we, therefore, think it is important to have a paper like the one we have written in the peer reviewed literature.

You suggest that we include visual guidance like the figures you kindly put in your report. We have considered this suggestion carefully and think that it is better in terms of space and impact to refer to papers such as Bhagat et al. 2020 JFM, 903 which shows measurements of CO₂ variations in an office, rather than reproduce other researchers figures. We also feel that the schematics you show are not entirely appropriate for this paper – we are after all writing for readers of Proc. Roy. Soc A.

The many other points you raise in your interesting and informative view are great food for thought and future action.

With best wishes

Paul

On behalf of all the authors.

Board Member

Comments to Author(s):

Firstly, thank you to the authors for submitting their report to PRSA and for their public service in working on the project. The three reviews are generally positive, recommending minor revision. A fourth review has been promised but is overdue. It will be forwarded if it is received soon.

We are extremely grateful for the positive and constructive comment of both yourself and the reviewers. We have made a number of changes throughout the manuscript, with additions to the text highlighted within the 'marked up' submission, and we have responded to each point raised by yourself and the reviewers. We do hope that you find our responses and amendments appropriate and that we can move swiftly towards publication so that the article might provide a useful resource this winter.

Based on my own quick reading, particularly of the sections on ventilation, I feel that significant improvements can be made but these should not be too time-consuming. Specifically:

1. While it may be acceptable for authors of a committee report to be responsible only for their own sections, all authors share in the responsibility for a paper, in a peer-reviewed journal, that includes them in the list of authors. I thus strongly recommend that all authors of the present paper read the whole manuscript carefully and send their recommendations for changes to the lead author.

We appreciate this point being raised. Throughout the discussion, which led to this paper being written, we continually sought to form a consensus between the group. The writing of the text was delegated to individuals, based on expertise, only for efficient production. Throughout the entire process, all authors have been accepting of responsibility not only for the text written by themselves but for all the text within the document. The editing process then sought to ensure a consistent/complimentary tone throughout the document. We only hope that we have succeeded in this regard.

2. I suspect that significant updating of the paper will be possible, and desirable, based on research published recently.

All authors have incorporated any recently published research relevant to COVID-19 or the study thereof that they are aware of.

3. As an example of this, Peter Rhines has told me about a paper just published in JFM entitled 'Effects of ventilation on the indoor spread of COVID-19', by Rajesh Bhagat et al.

Taking this article as an example, a subset of the authors were involved in both publications. Appropriate references to this article have now been made in a number of places.

4. Coming now to more on ventilation, I was intrigued to learn about the surprisingly high ventilation rate required to keep indoor CO₂ levels below, say, 1,000 ppm. The derivation of this is simple enough but would be worth including as a footnote, or even in the text.

This calculation has been added to the text.

5. In Section 3.1, results on the quanta required for infection are quoted from the literature rather uncritically. How robust do the authors think that the values are?

Yes, this is a very good point to have made. It was implicit that uncertainties abound from the variety of quanta rates taken within and across the studies reported. It was also explicit in the statement “However, just as for most other airborne diseases, there is wide variation in the values relevant for an infectious individual depending on a) their activity level, b) the viral load in their sputum and c) the ratio between one infectious quantum and the amount of infectious RNA/ml.”.

However, we agree this is not sufficient and we have amended the existing text to “Just as for most other airborne diseases, there is wide variation in the values relevant for an infectious individual depending on a) their activity level, b) the viral load in their sputum and c) the ratio between one infectious quantum and the amount of infectious RNA/ml. However, with such a novel disease uncertainties are compounded by our relative lack of experience and data concerning COVID-19.”, and further added the text “Not unsurprisingly, there is still great uncertainty regarding the nature of transmission and infection for the disease COVID-19. Difficulties in robustly parameterising and modelling infection via the airborne route does not escape these uncertainties. For example, the results discussed above take quanta generation rates that differ by at least two orders of magnitude. Efforts to reduce these uncertainties are ongoing; however, whilst the absolute numbers vary with such modelling choices the predominant conclusions that fewer people in well ventilated spaces for shorter durations will reduce transmission by the airborne route do not change.”.

6. The paragraphs around Table 1 make for heavy reading. Are they consistent with the contents of the table? Can they all be summarised in a single formula depending on things like volume of the occupied space, number of occupants, duration of occupation, number of infectious individuals, as well as q and Q ? Some streamlining of this section would be nice.

We have modified the text around Table 1 and more clearly aligned values discussed in the text (i.e. taken out any rounding). We have considered the inclusion of a simple formula carefully however to do so we would have to report a steady state which the literature does not (these studies allow for variation in the risk a quanta transiently build up due to people coming and going). We hope you agree that the new text is more streamlined.

7. Related to this, in Table 1, R seems to depend linearly on q for small q , but not for large q . Why? And why not a simple linear dependence on Q ? Maybe explained in the cited paper, but a summary here would be useful.

The infection risk depends on the exponential of an integral over the duration. In ‘low-risk’ scenarios events are well approximated as linear but reporting where this approximation ceases to be reasonable requires some non-rigorous/arbitrary choice which we prefer to avoid, not least to avoid the risk it becomes misleading.

8. I found Section A of the Supplementary Material to be rather weak. Several comments are really rather trivially obvious and could perhaps be omitted from a scientific review. The general recommendations are qualitative at best. Surely the analyses in the Bhagat et al. paper mentioned above can provide some guidance, in the absence of CO₂ sensors, on things like how far to open windows in typical situations?

The focus of this section was not ventilation experts but for laity; for example, members of Public Health England who may have considerable expertise relevant to COVID-19 but little experience in the provision of building ventilation. The primary function was to counter the perceived notion of binary state of windows e.g. fully open or fully closed. CO₂ sensors are a

much more practical and simple way to assess the ventilation provision in a well occupied space, such as a classroom. Asking occupants to undertake orifice flow equation calculations (which require effective area – a value dependent upon the opening characteristics of the window, and not easily obtained – see Sharpe, P., *et al.* (2020), <https://doi.org/10.1016/j.enbuild.2020.110556>) would be far more challenging than utilising CO2 as a proxy – which is suggested throughout the document.

9. Even qualitatively, I question some of the content. For example, it is assumed that opening a high vent in winter will lead to the inflow of fresh air. I suspect that, unless the vent is above a certain size, opening it will just lead to outflow of warm, buoyant, air that originally entered the room via infiltration or under the door from adjacent interior building space.

We appreciate your point. However, bi-directional flow will occur through any single opening, irrespective of it's location within the façade. Outside air will enter at the low point of the vent with warmer indoor air exhausting through the top portion of the vent.

10. I appreciate the trade-off between ventilation and comfort, particularly if 'displacement ventilation' is being used with inflow at a low level. How about a compromise, with inflow at some intermediate height allowed to warm up somewhat by mixing with room air as it descends to the floor before spreading and slowly rising and being exhausted through a high vent? Some deflection of the inflow might be required, but easy enough to achieve. Whatever the situation, quantitative guidance on advisable opening sizes would be nice.

You are correct to highlight that multiple flow configurations are potentially possible which is a reason that we are not always more specific in some of our text. In the context of this section of the report, we feel it prudent to simply consider airflow options with the available openings already present within the building fabric and avoid postulating air flow paths that may arise if changes were made.

Appendix B - Author's response to decision letter for RSPA-2020-0855.R1

Editorial comments on RSPA-2020-0855.R1.

I appreciate the desire to publish this report as soon as possible, and hope that the two rounds of peer review have been useful. The referees, and I, are clearly aware of the importance and urgency of the problem and commend the authors for all the hard work that they have undertaken. However, I think that the authors' responses to the first set of reviews were not completely satisfactory and that significant further changes are advisable before the paper is published in Proc. Roy. Soc. A.

As this will take time, I recommend that the report be issued first under the auspices of RAMP, perhaps along with a press release describing the main findings and recommendations. Further changes could be made at the authors' discretion before this is done, though these changes might not be as extensive as are desirable for the journal. A public release could possibly include a statement that a version of the report will be published later in a Royal Society journal.

Dear Chris,

We appreciate your comments and thank you for all your input. We respond to each of your points below.

Yours sincerely, Henry Burridge on behalf of his co-authors.

The second reviews from the referees recommend publication, though comments from referees 2 and 3 should be seriously taken into account. As an editor, I found the response to the first review by Peter Rhines to be inadequate, even dismissive. But here I will focus on my own earlier comments. In particular, I still find Section 3.1, including Table 1, to be confusing and incomplete. A reader who doesn't have a situation identical to that described in Table 1 would find it hard to take away any message more substantial than the obvious points that more quanta are bad and more ventilation is good.

We appreciate these points. We have written a fuller response to Peter Rhines – we certainly did not intend to be dismissive, members of RAMP have had extensive correspondence with Peter regarding the pandemic, we hope our latest response is satisfactory. We have worked on the text in order to clarify aspects around table 1 – you are correct that there is no guidance that can be universally issued to all spaces, hence our response of highlighting research which covers key indoor spaces including offices, schools, shops and restaurants.

This lack of generality is particularly unfortunate in that the nice 2020 paper by Burridge et al. only needs to be taken a little further to provide a formula that is reasonably accurate in realistic situations and easily applied. To be specific, as a first step it is surely worth considering the steady state situation with constant f , q , and n , and retaining only the first term in the expansion of the exponential in their equation 8 (as is reasonable if the risk is kept low). Then n is the same as N and close enough to $N-1$ to cancel the ratio. R_A , or S in the current paper, is just $f q T$ (writing T instead of T_A). Now $f=(C-C_0)/C_a=p/Q$, where C_a is the excess CO2 concentration and p is the individual breathing rate, which the authors take as 8L/minute.

The values given by this simplification can be compared with the values in Table 1 that allow for other considerations, such as the time-dependent build-up of contaminants. In particular, for the second column, which corresponds to the realistic situation that has $C=900$ ppm, the values are (0.16, 0.53, 2.7) instead of (0.13, 0.42, 2.1). A little higher, but not far off. The agreement is even better for the

third column, with $C=650$ ppm, though less so in both columns for the fourth and fifth row, perhaps not surprisingly as the next terms in the expansion of the exponential may come into play, though most of the difference may come from the time-dependence. This could be checked easily. The differences are larger in the first column, but here the time dependence is more important as Q is less. For the last two rows the next terms in the expansion of the exponential are probably important, but these cases are irrelevant as they give a high transmission rate and would never be considered. The first column has $C=1670$ ppm, I think, which may be unhealthy anyway, as the authors say on page 41. (It would be worth mentioning the Allen paper cited by Peter Rhines, with evidence for cognitive decline even at values of C less than 1,000 ppm.)

We have now included reference to a couple of papers on the wider effects of CO₂. We have added a short section on simple linear approximations to the airborne infection risk under Wells-Riley based modelling.

This suggests that things like the volume of the office are not important, except in determining small corrections to the basic formula in time-dependent cases, and that good guidance can be provided from the formula $S=fqT$, where $f=(C-C_0)/C_a$ could be determined from measured values of C , the carbon dioxide concentration. If desired, the formula could be presented graphically with, say, q on the ordinate and fT (with $f=p/Q$ or $(C-C_0)/C_a$) on the abscissa. It clearly shows the simple relevance, as expected, of all of f , q , and T , but quantitatively and with uncertainty in q the critical factor.

It's interesting that the result doesn't depend on the number of people in the room, but this is solely because the ventilation rate, for fixed f , is proportional to the number of occupants. In a typical situation, f would drop with fewer occupants. The formula would, of course, need to be scaled up if more than one occupant is infectious. It's recognised that the model assumes well-mixed conditions, or that averaging is equivalent to such an assumption.

This is an interesting suggestion but we feel this explores the modelling in a manner beyond the scope of this paper. We hope that the spirit of this comment is addressed by our inclusion of the discussion of linear approximations to the risk.

There's nothing special in the above as it is a very minor use of the Burridge et al. paper, and may not be entirely correct. It does need caveats, of course, but I urge the authors to consider including it, or something like it. It would also be useful to point out that the time scale for the indoor concentrations to asymptotically approach the steady state for a previously unoccupied and uncontaminated room is just $V/(NQ)$, or about an hour for the middle column and conditions of Table 1.

Further, the results of Buonanno cited on page 18 could be organised with, for each situation, the values of $S=fqT$, the time scale $V/(NQ)$ and the equilibrium CO₂ concentration C presented, and the value of S compared with that found by Buonanno. The values of S this way seem accurate for the cases described on lines 21-32 of page 18 of the present submission, but I haven't gone further. As it stands, this section does not really qualify as 'evidence synthesis'.

Even before this use of the Wells-Riley formula, I think it would be useful for the readers of the journal to see the derivation of the formula. For a start, it might even be worth mentioning where fqT comes from in that (as is in the literature), if A is the concentration of quanta in the room, then $ANQ=q$ in a steady state with N people and the intake rate of an individual is $pA=pq/(NQ)$ or fqT/N over a time T . Then with $N-1$ close enough to N , $S=fqT$. (I suppose the numbers I had for fqT before

for your Table 1 should be multiplied by 39/40.)

But here S really represents the total exposure. The W-R formula assumes that f_q/N represents the probability of infection per unit time. Thus if the time T is divided into m equal intervals, the chance of not being infected in one of the intervals is $1-f_qT/(Nm)$. So the chance of not being infected in m intervals is $[1-f_qT/(Nm)]^m$, which tends to $\exp(-f_qT/N)$ as m tends to infinity. Thus the chance of being infected is the W-R formula $1-\exp(-f_qT/N)$. All presumably well-known to the authors and in some of the literature, but perhaps worth including before moving on to use of the formula, both in full and in its linearised form (which I contend is adequate, given all the other uncertainties).

This also draws attention to the formula relying on the probabilistic assumption being made, as would be appropriate if an infectious dose could be obtained from a single breath, or a few breaths. But is it really applicable to uniformly dispersed aerosols, perhaps with a threshold below which there is no infection? In that case, the office is safe if f_qT/N is less than 1. Certainly not something that one would want to rely on for all sorts of reasons such as the well mixed assumption, but I do wonder whether the risk is being overestimated in some situations. I'm reminded of the Paracelsus dictum:

https://en.wikipedia.org/wiki/The_dose_makes_the_poison. It would at least help guide future research if this dichotomy were mentioned.

In summary, it seems to me that the W-R formula is being applied without adequate explanation or justification, and that the application is not even as transparent as it could easily be by explaining and applying a simple and probably adequate linearisation. I would have welcomed the kind of things that I am now calling for, rather than having to slow-wittedly figure it out myself.

We have now included presentation of the Wells-Riley equation in a manner that an interested individual could explore the discrepancies between a crude linearisation and the results presented in the papers cited.

On another topic, I don't agree with the authors' response to my earlier point 9. I've stayed in enough stuffy hotel rooms to know that a single opening may well just suck out of the room the air that has infiltrated from other parts of the building. The role of the wind in determining pressure differences and flows is also important. Even in an ideal situation where two-way flow occurs, this will be limited by a critical internal Froude number (earlier papers by Linden and others?). I appreciate that having cold air tempered by mixing with warm interior air is good for comfort, but surely it is much more effective to have inflow and outflow through different windows so that opening more than one high window, if possible, might be worth mentioning.

Thank you for this comment and you are quite correct that if the aim is to maximise the airflow then separating the inflow and outflow is a wise strategy. Moreover, you highlight the complexities in predicting the precise flow that arise – something that we emphasise in our section on ventilation but remains a research challenge far beyond the scope of this paper. In wintertime we feel the key is promoting a balance between ventilation and comfort. We have tried to emphasise this throughout our paper and the specific text which you refer to now reads: "A single set of high- and low-level openings: In wintertime it may be preferable to open the high-level vents first, providing this results in adequate supply of outdoor air. For relatively well-sealed rooms an exchange flow can then be expected across the high-level vents and the turbulent plume of cooler outdoor air entering will entrain warm room air as it falls under gravity, tempering the

air before it enters the occupied zone (Turner, 1973). A helpful draught plume calculator is available in the BB101 calculation tools, which enables this effect to be measured (UK Government, 2018). For less well-sealed rooms in wintertime, opening the high-level vents first is likely to promote a displacement ventilation flow which can be an efficient strategy for trying to reduce airborne infection (Bhagat & Linden, 2020). When the opening of the high-level vents is alone insufficient to provide adequate ventilation then low-level openings should be exploited to increase the airflow. A safe means of opening and closing high level vents should be supplied in workplaces (HM Government, 1992b, regulation 15).

Multiple openable windows and/or vents: Where a room has multiple openable vents, it may be possible to deliver the adequate ventilation provision through opening of just a single vent. However, it is usually possible to create a more comfortable indoor environment, with respect to draughts, if the airflow is achieved through opening all the vents by a smaller amount than that necessary in the scenario of a single set of openings as described above. If there are openable vents at both high and low level, then the principle of opening as many high-level vents should initially be considered (see above).”.

On another issue, I made the point earlier that it is the job of all the authors to read the paper carefully. I'm not convinced that this has happened. For example, there are many places where numerical results are presented to many more significant figures than is justified. Surely someone knows better than to do this! When there are so many authors, it is inappropriate to rely on referees or editors to pick up on details like this.

We have made every effort to ensure that all authors have re-read the paper on numerous occasions and that they do accept responsibility for all that is included within.

One other point: I find it confusing to have 'airborne' refer to aerosols, given that droplets are also airborne initially. Wouldn't it be better just to say 'aerosol' throughout?

We have decided to stick with the WHO definition of the 'airborne transmission' which refers to far-field infection by matter which is airborne but at source might have been droplet or aerosol dependent on how one distinguishes between the two and the influence of environmental conditions.

Finally, much of the paper contains simple qualitative statements that are really rather platitudinous (see my previous comment 8). Possibly acceptable for a report to be released to the public, but hardly necessary or appropriate for a scientific article. Considerable tightening of the paper is advisable for the final journal version.

We have tried hard to reduce qualitative statements but where insufficient evidence is available to make quantitative comment in the context of COVID-19 we believe that qualitative statements are better than none.

Referee: 2

Comments to the Author(s)

As I noted in my review, the paper is timely, well-written and reasonably comprehensive. I appreciate that the bulk of my comments were addressed in the revisions. I will note that two of the three reviewers recommended addressing the relative importance of airborne- versus fomite-driven transmission. While I certainly won't hold up publication on this, I do encourage the authors consider adding some commentary on the current state of the knowledge on transmission.

We thank the reviewer for their supportive comments. Regarding the relative importance of airborne- versus fomite-driven transmission we still do not regard there to be sufficient evidence to pass comment explicitly. However, the predominant focus of the document is on ventilation (and other strategies) aimed at reducing airborne transmission, we hope this focus implicitly evidences on which route we consider greater efforts to be best spent.

Response to Peter Rhines

Referee 3 (Peter Rhines): First, while largely supporting publication of the paper in my first review I (as well as other referees) advocated for adding a presentation of immediate measures, aimed at public as well as research communities, to reduce airborne transmission of COVID-19 disease. Despite occasional changes in the text, the resubmitted version continues to be largely devoid of specific advice that can be implemented quickly (none of my recommendations were addressed specifically).

And second, the vagueness of advice to the public reflects a lack of focused, quantitative experiments in the physical science of viral transmission. One year into the pandemic we are still seeing largely qualitative science bearing on ventilation, filtration, transport and mixing of viral material, whether by contact or in airborne pathways. There are more detailed studies of ventilation, masking, air filtration in the published literature (rapidly increasing in late 2020) but few of these are well exploited in the present manuscript.

Dear Peter

First let us say how much we appreciate the work that you have put into helping us improve our paper and its impact - you have given us much food for thought. We acknowledge that we have not included specific advice that can be implemented quickly. This is not the purpose of the paper and we do not think Proc. Roy. Soc. A is a venue for such advice. As you say there remains a lack of focused quantitative experiments and many aspects of viral transmission, infectious load, etc. are still very poorly understood. We have tried to summarise the current state of knowledge and, as scientists, have been reluctant to advocate responses in the face of these uncertainties. We respond to your specific comments below.

Thanking you again for your help with our paper - and your insights on Covid-19.

Best wishes

Paul Linden: on behalf of the authors

This 2nd review repeats some material from the initial review, but with new insights.

1. Assessment. Generally speaking, during the pandemic advice about air quality indoors and recommended social behavior coming from public health experts is vague, qualitative, lacking detailed observations. It is very similar to that provided during the flu pandemic 100 years ago. Assessment of indoor air quality requires solid quantitative observation and experiment. Despite valuable illustration of respired air patterns in a variety of circumstances the present manuscript is largely lacking quantitative observations. Discussion here, and more widely of ventilation schemes for indoor spaces, lacks sufficient detail to implement cold-season passive ventilation, filtration and heating for comfort. The complexity of indoor airflow and mixing makes it difficult to argue for or against mixing-ventilation (because threads of highly concentrated viral aerosol are spread by mixing, and may or may not be diluted). Airborne transmission of disease by viral aerosol hinges on the relationship between viral load and severity of infection¹, which remains uncertain.

Your comment illustrates why it is hard to offer implementable advice. Buildings differ widely as do climatic conditions, and what may be optimal in some cases will be poor in others. As you say the importance of airborne transmission is still unclear, although evidence suggests it can be quite

important.

Assessment of net ventilation rate for moderate sized dwellings, offices, care homes, restaurants, prisons, school classrooms needs guidance. The manuscript spends nearly a full page outlining difficulties in measuring net ventilation of an interior space (Sec. 3.1) yet in fact the net ventilation can readily be measured with sensors for passive tracers carried by the air:

(i) Ventilation time-series using CO₂ sourced with dry ice. With one or a few CO₂ sensors, time-series of CO₂ concentration and its variance can be recorded following injection of an initial 5000 ppm concentration (for example) sublimated from dry ice. The CO₂ is readily mixed to near uniformity with a fan, and subsequently is measured as it declines. This is readily done for a variety of active and passive ventilation schemes, and can in be quick and quantitatively accurate. It complements point-source release of CO₂ illustrated for occupied hallways in Sec. 2.5. Both forms of CO₂ release experiment can be used to assess air change ventilation in all manner of occupied spaces. Using 4 CO₂ sensors we have recorded time-series for both mean concentration and variability (notably vertical structure) through the ventilation period with a single elevated exhaust fan. This assessment of room air-change time-scale (or its inverse, air changes per hour) can then contribute when sufficient real data for the contact-traced infection rate in occupied rooms becomes available.

We agree estimates for ventilation from CO₂ decay are possible, but contest the assertion that it is easy. For a scientist with experience in lab techniques maybe, but even then it can be challenging as these rates vary depending on openings, weather conditions in naturally ventilated or mixed-mode spaces. Even in mechanically ventilated spaces the CO₂ needs to remain well-mixed for the data to be readily interpreted.

CO₂ of course is a swiss-army-knife for ventilation, recording respired breath (absent gas cokers and outdoor sources); being itself a health hazard in poorly ventilated indoor spaces; and CO₂ is a sensitive tracer of net ventilation, which when diluted moves accurately with the air.

Adverse health impacts of CO₂ laden air have themselves had extensive experimentation², going far beyond the manuscript's citation (Secs. 1 and 3.2) of the 1000 ppm CO₂ threshold for safe air (which turns out to be inaccurate if cognitive ability is the metric). Sec. 3.2 could well point to the need for opening bedroom windows at night to avoid really dangerous expired CO₂ levels (and potentially, infections).

We agree and have added a statement to that effect in section 3.2.

(ii) Ventilation time series using fine particle aerosol measurement with laser particle scattering (e.g., Purple Air, the widely deployed US network of dual laser particle sensors). Particles are seeded by a brief episode of stove-top cooking, which populates an interior space with a frightening concentration of 0.1 - 10 micron particles. These sensors are more accurate and less expensive than typical CO₂ sensors. PM_{2.5} aerosol measurements also provide independent data on infiltration of outdoor air into a building, and might be used along with CO₂ measurements to characterize both respired air volume and its (possibly infective) aerosol burden.

It is not obvious how PM_{2.5} measurements correlate with rebreathed air. It is also the case that particulate matter is generated indoors as well as infiltrating from outdoors. Carbon dioxide levels are usually a better indicator of the ingress of outside air.

The manuscript describes in detail models of viral transmission (Sec. 3.1), with observed CO₂ concentration providing evidence of respired air volume. However, 'superspreaders' account for a significant, perhaps dominant fraction of infections. These require contact tracing that reach both

both forward in time (with reproductive number R) and backward in time to locate viral sources (made vivid by the 'friends paradox' and the book The Rules of Contagion by Kucharski). This suggests backing off from the manuscript's long discussion and quantitative recommendations based on modeling of both airborne transmission rates and epidemiology of infections (Sec. 3.1 and Burrige et al. 2020), about which they say "...superspreading events - we choose not to dwell on such cases but note that should we have done so then the risks reported herein would be dramatically increased." ... because the viral load from a superspreader is far from uniform among inhabitants of the space. Genomic reconstruction of the huge network of infections seeded by a February 26-27 business conference in Boston³ shows superspreading in action, and underlines the need for similarly sophisticated analysis of physics of airborne transmission. Yet CO₂ is nevertheless a promising indicator of expired, potentially infective, air, and has provided particularly vivid pre-pandemic studies of indoor air quality. Reproducing a figure based on the school-room study of Perez et al. (Sensors, 2018) would make that point.

We are not in a position to discuss super spreading events, as there is huge uncertainty about their causes. We agree that CO₂ is a good indicator of indoor air quality, and have now included a reference to Perez et al. (2018). Thanks for bring that paper to our attention.

2. Remediation. Both referee 1 and I urged the inclusion of a brief account of procedures the public could embrace for avoiding COVID-19 infection, possibly to be followed by more expansive publication and website activity. The authors responded that relevant publications by their group are available separately. Yet the two references (CIBSE Guide B: Ventilation and Ductwork, 2020 and MusicMark Ventilation of Teaching Spaces, 2020) seem inadequate for widespread advice. I may well be missing some better outreach publications, but the current surge in infections makes it obvious that the general population is unable to protect itself despite measures available to do so. Governmental agencies like WHO and CDC in the US have many relevant advisory documents on the web, yet they often lack detail and lag far behind the science.

Most of the world's people are at risk of Covid-19 illness. Simple measures to isolate them from the virus are available and should be a part of any comprehensive publication of the science of the pandemic. These are not normal times, where presentation of underlying science and applied practice are separated and deferred. Of course Proc.Roy.Soc.A papers are not read by the general public but investigative journalists would respond to sharply focused strategic advice. The manuscript already provides timely discussion of masking, fomites and disinfectants, droplet dynamics, UV treatment, qualitative patterns of aerosol spreading, quantitative model study of infection inferred from CO₂ concentration of respired air.

One example to recommend regards face masking: coverage in the text is long and comprehensive, and I see your response to reviewer 2 on this issue. But instead of a full page simply referencing many advocates of masking why not include a figure⁴ showing effectiveness of various masking materials, which can protect the wearer as well as providing 'source control' to protect others. This could amplify your message and inspire mask construction at home or commercially, with great effect. It might also reduce the selfish resistance to wearing masks found widely among those unconcerned with the public good. More detailed data on pathways (nose, mouth, larynx, lungs) for aerosol inhalation with and without masks is given by Xi et al. (Phys. Fluids October 2020).

Exhaust fans are another example of great importance. The key goal is steady upward displacement ventilation, carrying virus away from face level, using active, elevated exhaust at a high window. The manuscript strangely describes only passive ventilation. Forced displacement ventilation is described for hospitals by Bhagat & Linden (Roy. Soc. Open Science, 2020) but is not mentioned in the present paper. Although windows do not normally extend up to the ceiling, it is not difficult to cut a

small opening in an exterior wall. Failing that, fans combined with ducts can simulate an elevated exhaust port (accepting the head loss in curved ducts). Relatively inexpensive exhaust fans are capable of a wide range of flow rates, from ~ 100 cfm to several thousand cfm (cubic feet per minute), readily enabling rapid room air changes per hour. Attention to outdoor air input at floor level also is needed, and ducted fans may assist there as well.

Extensive discussion is given in the manuscript to support opening of windows to improve passive ventilation, with qualitative suggestions on how this should be done (Secs. 3.2, A.1, A.2). When an elevated window is opened during the cold season, does the air flow inward or outward? Of course it can do either or both, depending on interior thermal stratification, outside temperature, winds. Indeed, Baghat et al. 2020 give fine discussion of these issues. As the manuscript well describes (Sec. 2.4), rapid purging of air during frequent breaks can improve school classroom air quality, but this is most rapidly done with active exhaust fans.

Portable air purifiers are another example of remediation (Sec. 4.1). As well as highly engineered commercial units, inexpensive home-built purifiers combining a HEPA filter and box window-fan have been widely tested, and can effectively remove fine aerosol particles bearing virus. Typical airflow rates range from 100 to 400 cfm. This provides a strategy for the cold season where installed HVAC heating cannot be retrofitted with HEPA or MERV-16 level filters. The current text acknowledges portable air cleaners yet with ambivalent recommendation.

Outdoor classrooms, established a century ago during the Spanish flu pandemic, are able to keep children and teachers safe. Even in cold climates this seemingly extreme strategy is working, though not widespread. Dressed warmly, classes in partially enclosed large tents are currently open in our town, where the public schools are locked down, following a brief period of in-person schooling given up.

The pandemic may be with us for some time. Cold-season ventilation using heat-recovery ventilation (HRV) could reduce the conflict between keeping warm and keeping ventilated indoors. Capture efficiencies of 70% to 80% of the temperature difference between warm outgoing and cold incoming airstreams are possible. While commercial units for entire buildings are costly, heat-exchanger ventilation units can be constructed for smaller spaces without great cost.

We agree that mitigation measures are important but feel that making specific recommendations as described above do not sit well in the paper. To take the example of masks, we are not qualified to evaluate the effectiveness of different materials, and to simply quote another study without being able to evaluate with any authority whether the conclusions of that study are valid seems questionable to say the least. Similarly, there is significant controversy around air purifiers and filters, and again we have not addressed these issues nor are we qualified to do so.

We understand the need to take active measures and in different capacities and through different avenues we are doing that as best we can.

Appendix C - referee 3's attachment for RSPA-2020-0855.R0

Referee's report: The ventilation of buildings and other mitigating measures for COVID-19: a focus on winter 2020, task 7 of the Royal Society RAMP project

This has been a difficult paper to review, owing to the complexity of indoor air flow, coupled with the many uncertainties about airborne viral transport and infection. The current (Nov 29, 2020) surging COVID-19 infection rate makes it vital to suggest immediate remediation strategy, in enough detail to be implemented widely and quickly without much expertise or training. This must be obvious to the authors, and I can only wonder whether parallel documents focusing on implementation are in the works, or are already somewhere in the vast literature (much of it as yet unrefereed). There are related assessment and remediation efforts throughout the world, which need constant attention. The natural medium for timely, adaptive communication is of course the online Web. Many governmental and NGO web sites exist; tending to repeat the short-list of familiar qualitative and often incomplete advice. Directing the present paper into a comprehensive, frequently updated website will be most useful.

My background is research in fluid mechanics at Cambridge Univ., MIT, Univ. of Washington) on the large scale of atmospheres and oceans. I have been working to understand air quality (AQ) research, for both physical indoor airflow and epidemiological/virological understanding of infection pathways. This is primarily an effort to advise schools struggling to reopen, in a small town in the northwestern US, and more generally to experiment with and promote new techniques for AQ improvement.

Organizing airflow and mixing to make indoor air safe hinges on understanding the viral load carried by airborne particles and droplets, and on how the intensity of infective illness depends on that viral load. Dilution of contaminated air is evidently crucial, made obvious by the relative safe AQ found outdoors. But even this intuitive result (that diluted, less concentrated viral particle clouds produce milder illness, or none at all)¹ is not known in detail for COVID-19. Quantitative circulation and mixing strategies thus rest on uncertain ground. Guidance in this paper makes impressive use of clear physical dynamics of airflow and mixing coupled with epidemiological ideas which are relatively schematic and untested. Both point to the need for extensive quantitative data collection in parallel with deployment of ventilation and air filtering strategies.

Modelling the infective potential of air is at an early stage, yet begins to give strategic advice to bring aerosol particle densities, viral load on those particles, duration of exposure and likelihood of transmission all down to safe levels. Modelling originating with Rudnick et al. 2003, seeks to estimate the reproductive index R , and has advanced currently (Burrige et al. 2020 and others), beginning to provide useful advice in the context of offices and school-rooms. Concentration of infections in homes, bars, restaurants, nursing homes, hospitals and, particularly, among often crowded low-income families begs for their consideration, needing immediate help. Despite many months of attempts to reduce transmission in care- (nursing-) homes, they continue to account for ~40% of US infections.

Model analysis central to this paper runs into difficulty due to super-spreaders (as remarked by Burrige *et al.* 2020): crude estimates suggest roughly 80% of infections are caused by contact

with 20% of infected people. Contact tracing forward in time then needs to look backwards to find the source of an infection rather than focusing only on the destination of subsequent infections. Super-spreading events have been numerous, and are increasingly well documented².

Innovative use of CO₂ observations of interior air to predict communication of virus are important part of the strategy, yet managing CO₂ sensors, accumulating data and coupling it to infective transmission modelling are all complex tasks, with challenges for up-scaling to measure many parallel sites. Experience can be gained with aggressive experimental deployments under a variety of ventilation conditions, also taking into account sources of CO₂ other than human breath. Infrared NDIR CO₂ sensors tend to require frequent calibration, have slow response, and less accuracy than sensors for fine aerosol particle concentration, temperature, and humidity which can provide some of the same information on ventilation and mixing.

Rapid implementation of ventilation strategies requires much detail, and this needs to inform a wide audience; not only experts. While the appendices give detail in subproblems like UV treatment, infection via virus on surfaces, corresponding detail on airflow (ventilation and interior recirculation) is less complete. Appendix A gives general guidance, yet is just the beginning of a cold-season strategy to balance the dual needs for fresh outdoor air ventilation and interior recirculation by installed HVAC heating and filtered portable air purifiers.

In view of the need for immediate deployment of ventilation/filtering techniques across the whole range of indoor spaces, I suggest only minor changes in this paper, yet immediate follow-up and Web-based communication. My experience in a rural town of ~5,000 inhabitants, near a major city, is that public health officials, medical workers and school administrators are overwhelmed with their task, particularly due to failure of national public health guidance in US (and seemingly so in UK). It is a time to energize and train volunteers, many of whom are isolated at home: calling heavily on the retired and, in the case of schools, on young families distraught and emotionally disturbed by months of isolation and loss of employment. In schools, teaching the science of air in its broadest sense can accompany student/teacher projects on ventilation. This group's guidance and energy could circumvent the inertia of government, in proving its worth to public health administrators

To summarize more specifically: I suggest considering these measures meriting more attention, briefly in the present paper and extensively in future. Also appended below are a few figures. The manuscript lacks visual guidance of this kind and I recommend including these or others like them.

- o While honing the CO₂ technique for inferring ventilation and danger of infection, explore more readily measured tracers of ventilation and filtration. CO₂ sensors are expensive, sometimes unreliable, and require frequent calibration. Multiple tracers provide both redundancy and a variety of source distributions for exploration of ventilation and outdoor air infiltration (though only CO₂ senses expired breath volume)³.

- Less expensive and more accurate PM_{2.5} laser fine-particle sensors mounted indoors readily measure air exchange using a seeded aerosol as an indoor

tracer: for example simply residual aerosol from kitchen cooking (Figure 1 at end). CO2 sensors also provide similar information by recording its decay rate in absence of sources⁴.

This gives another measurement for exploring ventilation strategies. Ambient smoke or other pollutants outdoors, measured by a second PM2.5 sensor outdoors, act as seed tracers for infiltration, which is surprisingly rapid even with unventilated, tightly constructed dwellings. These complementary air transports, outward and inward, would be the same for a box-model of ventilation but real indoor spaces may reveal inefficient air pathways.

- Temperature and humidity sensors (incorporated in CO2 sensors) provide independent estimates of ventilation/infiltration rates, and to some extent expired air through correlation with CO2.
 - Infrared thermometers with built-in laser pointers can assess temperature stratification, HVAC affected hot spots, and stagnant zones in a room, very rapidly.
 - Finally, document non-human-respired sources of CO2 from gas cookers, propane stoves, infiltration of outdoor pollutants and photosynthetic cycles.
- o Expand this sensor data to include mapping indoor spaces in space and time, summarizing beneficial effects of purging (e.g. rapid ventilation during frequent school class breaks), recording variance statistics as well as 'typical diurnal cycles' through a wide range of externals: weather, doors/windows/HVAC performance, ventilation/filtration configuration, purging of air at night and on weekends.
 - o As virological data becomes available, focus on viral load measured as weighted aerosol particle size spectrum, to enhance understanding of the distribution among particle sizes and hence the effect of filtration on AQ⁵.
 - o Recognize that outdoor air quality may be sub-standard, from industrial and automotive pollution, HVAC exhaust, and wild-fires in western US, all requiring a shift of strategy to optimize indoor AQ; explore the complement to exhaust ventilation study: infiltration of outdoor air in a variety of regimes, using tracers of outdoor air like PM2.5 fine particle density⁴.
 - o For schools particularly, consider *outdoor* classrooms (shared time with indoor rooms), with safety assessed by ventilation/infiltration measurements using CO2 assessment of ventilation rate and recycling of exhaled breath⁶.
 - o Encourage massive construction of inexpensive portable air purifiers with HEPA filters. Develop clear protocols for filter replacement, for MERV-rated and HEPA rated filters.
 - o Encourage massive mask construction at home using recent assessment of filtration efficiency of fabrics, for example the surprising superiority of vacuum-cleaner bags (§2.3.2 and⁷), and the need to improve edge sealing of the masks.
 - o Encourage immediate deployment of exhaust fans with advice for ductwork minimizing length and curvature in cases where high windows are absent (carefully recognizing Reynolds number-dependent energy loss in curvy ducts, and the choice between AC fans and pulse modulated DC duct blowers less likely to stall

- by flow resistance). Both steadily running low power (typically 300 cfm) exhaust fans and inexpensive high-power 'attic fans' (typically 1000 cfm and greater) for purging indoor spaces ...in a few minutes... during brief unoccupied periods, tuning these breaks to minimize rebreathing of expired air. Attend to inflow pathways for the outdoor air, possibly incorporating active duct fans.
- o Increase observation or experimentation under real-world conditions (for infections as well as ventilation rates) updating advice on optimal frequency of air changes per hour, building confidence in physical/epidemiological models.
 - o Attempt to give clearer advice for monitoring CO2 and other tracers, to optimize in the cold season the difficult balance between
 - ventilation by exhaust fans;
 - heating by both installed HVAC units;
 - tuning the outdoor ventilation vs. indoor air recirculation by existing HVAC (with upgraded filters); and,
 - deployment of portable HEPA air purifiers.
 - o Assess and adopt innovations as they occur:
 - cell-phone tracking apps for forward and retrospective contact tracing, revealing close proximity to an infected person; particularly important for super-spreader events⁸
 - HRV and ERV 'heat-recovery' and 'energy-recovery' ventilation systems that exchange indoor air for outdoor air while retaining most of the heat (HRV) and moisture (ERV) of the outgoing air. Commercial systems have low flow rates (~ 100 cfm) and are very expensive; home-built heat exchangers can be constructed at low cost⁹, Figs. 4,5.
 - coordinate one- or two-stage pooled viral testing strategy for well-defined work-, social- and family groups. This can guide optimal definition of small work- or class groups in schools, offices and homes in collaboration with CO2 measurement¹⁰.
 - o Web-based group communication is an essential component of COVID-19 remediation. Educating for ventilation techniques and safe organization of indoor spaces needs to be rapidly promulgated. This online focus could lead to widespread volunteer activity related to improving AQ. And it could present other routes to alleviating the isolation of young people under COVID-19, reconnecting them with Nature and science.

To summarize, the paper is a useful and wide-ranging step toward mitigation of infectious indoor air during COVID-19. It needs to point to immediate measures for the people at risk, who often are relatively unfamiliar with pathways of infection and with heating, ventilation and filtration hardware. In order not to delay publication I suggest brief amendments, followed later by expansive website material and other forms of rapid communication, at the authors' discretion. Quantitative measurement should be a high priority, pursued immediately: we have had nearly a year of vague prescriptions for social distancing, masking, ventilation with a great lack of clear experiments, while infections and mortality surge and surge again.

Peter Rhines, University of Washington rhines@uw.edu
<http://www.ocean.washington.edu/research/gfd>

¹ Heneghan, C. et al 2020, SARS-CoV-2 viral load and the severity of COVID-19

Centre for Evidence Based Medicine

<https://www.cebm.net/covid-19/sars-cov-2-viral-load-and-the-severity-of-covid-19/>

² Adam D.C. *et al.* 2020. Clustering and superspreading potential of SARS-CoV-2 infections in Hong Kong

<https://www.nature.com/articles/s41591-020-1092-0> Super spreading and contact tracing

<https://www.theatlantic.com/health/archive/2020/09/k-overlooked-variable-driving-pandemic/616548/>

³ Measurement of CO₂ concentration to infer expired air volume is a remarkable tool. It would be helpful to list the percentage of expired air in a room, relative to 'fresh' outdoor air, at a range of CO₂ concentrations.. 1.5% at 1000 ppm etc. and to study variability of CO₂ concentration in expired breath (owing to age, robust metabolism, etc.). This could readily be done with cold-start experiments with initially uncontaminated air and known inhabitants.

CO₂ data could be used for steady-state and time-dependent assessments, both averaging by moving the sensors, and looking locally, close to face-level of occupants. Although slow response of NDIR sensors is a problem.

Interpretation still presents challenges. For data at a single location within an occupied room, how is one to know whether a peak in CO₂ concentration originated from a single, possibly infected person, or through lesser volumes of expired air from many people? Is there anything in the data that would identify a violent cough or sneeze (perhaps PM_{2.5} particle spectra for a range of particle sizes might do this)?

Earlier work before the pandemic with CO₂ (the revealing papers of Perez *et al* 2018; Muscatello, N. *et al.*, *J.Indoor Air*, 2015) sought to establish adverse health effects of high-CO₂ air. These continue to be of interest, and not irrelevant to pursuit of unhealthy COVID-19 air (Figs. 2,3).

Non-human sources of CO₂ anomalies include gas cookers, propane stoves, wood smoke, industrial/vehicular pollution and photosynthesis.

⁴ Barn, P. *et al.* 2008. Infiltration of forest fire and residential wood smoke: an evaluation of air cleaner effectiveness.

<https://www.nature.com/articles/7500640#ref-CR18>

⁵ Determining the viral load of drifting aerosols is a task in its infancy, as is its implications for infective potential. One hopes for progress soon.

[https://www.thelancet.com/journals/laninf/article/PIIS1473-3099\(20\)30232-2/fulltext:](https://www.thelancet.com/journals/laninf/article/PIIS1473-3099(20)30232-2/fulltext)

"The mean viral load of severe cases was around 60 times higher than that of mild cases, suggesting that higher viral loads might be associated with severe clinical outcomes."

The idea is that the immune system can eliminate a few Coronavirus particles, yet a high density of viral particles initiates a 'cytokine storm' where the immune system attacks itself. Yet this has been contradicted in more recent research:

Kox, M. *et al.* 2020, *JAMA*. 2020;324(15):1565-1567. doi:10.1001/jama.2020.17052

⁶ There is a long history of outdoor classrooms in the US, especially during the tuberculosis (TB) epidemic and Spanish Flu epidemic peaking in 1918. The cure for TB was to isolate and sleep outdoors for many months.

<https://www.nytimes.com/2020/07/17/nyregion/coronavirus-nyc-schools-reopening-outdoors.html>

<https://www.theguardian.com/teacher-network/outdoor-learning>

Sunfield Farm School is a Waldorf School in Port Townsend, Washington, emphasizing outdoor activity and blended indoor/outdoor classes in a farm setting.

<http://sunfieldfarm.org/>

While, as COVID-19 infections surge, public schools throughout Washington have returned to online teaching following a brief period of in-person classes during the fall of 2020, Sunfield Farm hopes to remain fully open with its blend of outdoor classes, farm work and indoor classes. We have begun to monitor CO₂ there, and hope to install overhead exhaust fans to support this effort.

Outdoors, tenting open at one end is set up close to indoor classrooms, whose windows and doors are generally left open. Children are heavily dressed, and frequent breaks scheduled with walking and running in the 50-acre rural space (with farm animals). Outdoor schools attract outdoor instructors who would otherwise be unemployed during COVID-19.

⁷ Testing of mask cloth for optimal fine particle filtration. Pan, Ham, Leng & Marr, MedRxiv preprint, Nov. 2020

<https://www.medrxiv.org/content/10.1101/2020.11.18.20233353v1>

⁸ Cell phone apps for contact tracing.

<https://www.nhsx.nhs.uk/covid-19-response/nhs-covid-19-app/>

<https://www.cnbc.com/2020/09/01/apple-google-will-build-coronavirus-contact-tracing-software-right-into-your-phone.html>

⁹ Heat exchangers coupled with balanced ventilation/infiltration systems (HRV, heat recovery ventilation) can introduce fresh outdoor air while retaining much of the heat anomaly in the exhaust branch. Indoor humidity control is also possible (ERV, energy recovery ventilation). Commercial units are very expensive (\$1000 - \$2000 plus cost of insulated ductwork), but promising experiments suggest that inexpensive coaxial ducts can work as well. Yet development is needed. Schematic figures 2a, 2b show commercial units, which can be tied to HVAC systems.

¹⁰ Mallapaty, S., July 2020. The mathematical strategy that could transform Coronavirus testing, Nature 583.

<http://acdc2007.free.fr/nature20720.pdf>

US CDC, October 2020. Interim Guidance for Use of Pooling Procedures in SARS-CoV-2 Diagnostic, Screening, and Surveillance Testing

<https://www.cdc.gov/coronavirus/2019-ncov/lab/pooling-procedures.ht>

Fig.1 Passive ventilation clears into air contaminated by kitchen cooking. Two-day record of PM2.5 EPA fine particle index indoors, shows two pulses of cooking induced contamination. The particle density decay with time is largely due to exchange with outdoor air (with no open doors or windows, no forced ventilation). Some deposition onto hard surfaces may also be occurring. The PurpleAir crowd-funded network of dual laser particle sensors has tested well for accuracy, precision and coverage: www.purpleair.com.

Figure 6. Transient evaluation of temperature, humidity and CO₂ concentration in room R2.17 in the period between 19.02.2018 and 19.03.2018. Due to overheating the thermal comfort was only medium throughout the week. The CO₂ levels have been recorded to be above 2000 ppm almost every day at some point.

Figs. 2,3 From Perez *et al.* 2018. Inclusion of figures like this, with explanation, in the manuscript could improve its impact.

Figure 7. (a) Exemplary dynamics of the evolution of the CO₂ level on three days of the week. Because doors and windows are closed after school finished CO₂ can accumulate. (b) The accumulation leads to a constant increase in the background level of CO₂, which makes it harder to maintain good CO₂ levels the older the week gets.

Figs. 4,5. Schematic HRV heat recovery ventilation system, tied into resident HVAC.

Appendix D - board member's attachment for RSPA-2020-0855.R1

Editorial comments on RSPA-2020-0855.R1.

I appreciate the desire to publish this report as soon as possible, and hope that the two rounds of peer review have been useful. The referees, and I, are clearly aware of the importance and urgency of the problem and commend the authors for all the hard work that they have undertaken. However, I think that the authors' responses to the first set of reviews were not completely satisfactory and that significant further changes are advisable before the paper is published in Proc. Roy. Soc. A.

As this will take time, I recommend that the report be issued first under the auspices of RAMP, perhaps along with a press release describing the main findings and recommendations. Further changes could be made at the authors' discretion before this is done, though these changes might not be as extensive as are desirable for the journal. A public release could possibly include a statement that a version of the report will be published later in a Royal Society journal.

The second reviews from the referees recommend publication, though comments from referees 2 and 3 should be seriously taken into account. As an editor, I found the response to the first review by Peter Rhines to be inadequate, even dismissive. But here I will focus on my own earlier comments. In particular, I still find Section 3.1, including Table 1, to be confusing and incomplete. A reader who doesn't have a situation identical to that described in Table 1 would find it hard to take away any message more substantial than the obvious points that more quanta are bad and more ventilation is good.

This lack of generality is particularly unfortunate in that the nice 2020 paper by Burridge et al. only needs to be taken a little further to provide a formula that is reasonably accurate in realistic situations and easily applied. To be specific, as a first step it is surely worth considering the steady state situation with constant f , q , and n , and retaining only the first term in the expansion of the exponential in their equation 8 (as is reasonable if the risk is kept low). Then n is the same as N and close enough to $N-1$ to cancel the ratio. R_A , or S in the current paper, is just fqT (writing T instead of T_A). Now $f=(C-C_0)/C_a=p/Q$, where C_a is the excess CO₂ concentration and p is the individual breathing rate, which the authors take as 8L/minute.

The values given by this simplification can be compared with the values in Table 1 that allow for other considerations, such as the time-dependent build-up of contaminants. In particular, for the second column, which corresponds to the realistic situation that has $C=900$ ppm, the values are (0.16, 0.53, 2.7) instead of (0.13, 0.42, 2.1). A little higher, but not far off. The agreement is even better for the third column, with $C=650$ ppm, though less so in both columns for the fourth and fifth row, perhaps not surprisingly as the next terms in the expansion of the exponential may come into play, though most of the difference may come from the time-dependence. This could be checked easily. The differences are larger in the

first column, but here the time dependence is more important as Q is less. For the last two rows the next terms in the expansion of the exponential are probably important, but these cases are irrelevant as they give a high transmission rate and would never be considered. The first column has $C=1670$ ppm, I think, which may be unhealthy anyway, as the authors say on page 41. (It would be worth mentioning the Allen paper cited by Peter Rhines, with evidence for cognitive decline even at values of C less than 1,000 ppm.)

This suggests that things like the volume of the office are not important, except in determining small corrections to the basic formula in time-dependent cases, and that good guidance can be provided from the formula $S=fqT$, where $f=(C-C_0)/C_a$ could be determined from measured values of C , the carbon dioxide concentration. If desired, the formula could be presented graphically with, say, q on the ordinate and fT (with $f=p/Q$ or $(C-C_0)/C_a$) on the abscissa. It clearly shows the simple relevance, as expected, of all of f , q , and T , but quantitatively and with uncertainty in q the critical factor.

It's interesting that the result doesn't depend on the number of people in the room, but this is solely because the ventilation rate, for fixed f , is proportional to the number of occupants. In a typical situation, f would drop with fewer occupants. The formula would, of course, need to be scaled up if more than one occupant is infectious. It's recognised that the model assumes well-mixed conditions, or that averaging is equivalent to such an assumption.

There's nothing special in the above as it is a very minor use of the Burrige et al. paper, and may not be entirely correct. It does need caveats, of course, but I urge the authors to consider including it, or something like it. It would also be useful to point out that the time scale for the indoor concentrations to asymptotically approach the steady state for a previously unoccupied and uncontaminated room is just $V/(NQ)$ or about an hour for the middle column and conditions of Table 1.

Further, the results of Buonanno cited on page 18 could be organised with, for each situation, the values of $S=fqT$, the time scale $V/(NQ)$ and the equilibrium CO2 concentration C presented, and the value of S compared with that found by Buonanno. The values of S this way seem accurate for the cases described on lines 21-32 of page 18 of the present submission, but I haven't gone further. As it stands, this section does not really qualify as 'evidence synthesis'.

Even before this use of the Wells-Riley formula, I think it would be useful for the readers of the journal to see the derivation of the formula. For a start, it might even be worth mentioning where fqT comes from in that (as is in the literature), if A is the concentration of quanta in the room, then $ANQ=q$ in a steady state with N people and the intake rate of an individual is $pA=pq/(NQ)$ or fqT/N over a time T . Then with $N-1$ close enough to N , $S=fqT$. (I suppose the numbers I had for fqT before for your Table 1 should be multiplied by 39/40.)

But here S really represents the total exposure. The W-R formula assumes that fq/N represents the probability of infection per unit time. Thus if the time T is divided into m equal intervals, the chance of not being infected in one of the intervals is $1-fqT/(Nm)$. So the chance of not being infected in m intervals is $[1-fqT/(Nm)]^m$, which tends to $\exp(-fqT/N)$ as m tends to infinity. Thus the chance of being infected is the W-R formula $1-\exp(-fqT/N)$. All presumably well-known to the authors and in some of the literature, but perhaps worth including before moving on to use of the formula, both in full and in its linearised form (which I contend is adequate, given all the other uncertainties).

This also draws attention to the formula relying on the probabilistic assumption being made, as would be appropriate if an infectious dose could be obtained from a single breath, or a few breaths. But is it really applicable to uniformly dispersed aerosols, perhaps with a threshold below which there is no infection? In that case, the office is safe if fqT/N is less than 1. Certainly not something that one would want to rely on for all sorts of reasons such as the well-mixed assumption, but I do wonder whether the risk is being overestimated in some situations. I'm reminded of the Paracelsus dictum: https://en.wikipedia.org/wiki/The_dose_makes_the_poison. It would at least help guide future research if this dichotomy were mentioned.

In summary, it seems to me that the W-R formula is being applied without adequate explanation or justification, and that the application is not even as transparent as it could easily be by explaining and applying a simple and probably adequate linearisation. I would have welcomed the kind of things that I am now calling for, rather than having to slow-wittedly figure it out myself.

On another topic, I don't agree with the authors' response to my earlier point 9. I've stayed in enough stuffy hotel rooms to know that a single opening may well just suck out of the room the air that has infiltrated from other parts of the building. The role of the wind in determining pressure differences and flows is also important. Even in an ideal situation where two-way flow occurs, this will be limited by a critical internal Froude number (earlier papers by Linden and others?). I appreciate that having cold air tempered by mixing with warm interior air is good for comfort, but surely it is much more effective to have inflow and outflow through different windows so that opening more than one high window, if possible, might be worth mentioning. .

On another issue, I made the point earlier that it is the job of all the authors to read the paper carefully. I'm not convinced that this has happened. For example, there are many places where numerical results are presented to many more significant figures than is justified. Surely someone knows better than to do this! When there are so many authors, it is inappropriate to rely on referees or editors to pick up on details like this.

One other point: I find it confusing to have 'airborne' refer to aerosols, given that droplets are also airborne initially. Wouldn't it be better just to say 'aerosol' throughout?

Finally, much of the paper contains simple qualitative statements that are really rather platitudinous (see my previous comment 8). Possibly acceptable for a report to be released to the public, but hardly necessary or appropriate for a scientific article. Considerable tightening of the paper is advisable for the final journal version.